# Juxtaposition of heterozygous and homozygous regions causes reciprocal crossover remodelling via interference during Arabidopsis meiosis

Piotr A Ziolkowski[1,2], Luke E Berchowitz[3,4], Christophe Lambing[1], Nataliya E Yelina[1], Xiaohui Zhao[1], Krystyna A Kelly[1], Kyuha Choi[1], Liliana Ziolkowska[1], Viviana June[1], Eugenio Sanchez-Moran[5], Chris Franklin[5], Gregory P Copenhaver[3,4], Ian R Henderson[1]*

[1]Department of Plant Sciences, University of Cambridge, Cambridge, United Kingdom; [2]Department of Biotechnology, Adam Mickiewicz University, Poznań, Poland; [3]Department of Biology and the Carolina Center for Genome Sciences, University of North Carolina at Chapel Hill, Chapel Hill, United States; [4]Lineberger Comprehensive Cancer Center, University of North Carolina School of Medicine, Chapel Hill, United States; [5]School of Biosciences, University of Birmingham, Birmingham, United Kingdom

**Abstract** During meiosis homologous chromosomes undergo crossover recombination. Sequence differences between homologs can locally inhibit crossovers. Despite this, nucleotide diversity and population-scaled recombination are positively correlated in eukaryote genomes. To investigate interactions between heterozygosity and recombination we crossed *Arabidopsis* lines carrying fluorescent crossover reporters to 32 diverse accessions and observed hybrids with significantly higher and lower crossovers than homozygotes. Using recombinant populations derived from these crosses we observed that heterozygous regions increase crossovers when juxtaposed with homozygous regions, which reciprocally decrease. Total crossovers measured by chiasmata were unchanged when heterozygosity was varied, consistent with homeostatic control. We tested the effects of heterozygosity in mutants where the balance of interfering and non-interfering crossover repair is altered. Crossover remodeling at homozygosity-heterozygosity junctions requires interference, and non-interfering repair is inefficient in heterozygous regions. As a consequence, heterozygous regions show stronger crossover interference. Our findings reveal how varying homolog polymorphism patterns can shape meiotic recombination.

*For correspondence: irh25@cam.ac.uk

Competing interests: The authors declare that no competing interests exist.

## Introduction

Sexual reproduction via meiosis is highly conserved within eukaryotes and allows recombination of genetic variation within populations (*Barton and Charlesworth, 1998*). During meiosis homologous chromosomes pair and undergo crossover recombination, which together with independent chromosome segregation and gamete fusion increases genetic diversity between progeny (*Barton and Charlesworth, 1998*; *Villeneuve and Hillers, 2001*). Meiotic crossovers form via the repair of DNA double-strand breaks (DSBs) generated by the SPO11 endonuclease (*Bergerat et al., 1997*; *Keeney et al., 1997*). Nucleolytic resection of DSBs generates 3′ single-stranded DNA (ssDNA), which is bound by the RAD51 and DMC1 recombinases (*Bishop et al., 1992*; *Shinohara et al., 1992*). The resulting nucleoprotein filament then invades a homologous chromatid to form a heteroduplex

**eLife digest** The genomes of plants and animals consist of several long DNA molecules that are called chromosomes. Most organisms carry two copies of each chromosome: one inherited from each parent. This means that an individual has two copies of each gene. Some of these gene copies may be identical (known as 'homozygous'), but other gene copies will have sequence differences (or be 'heterozygous').

The sex cells (eggs and sperm) that pass half of each parent's genes on to its offspring are made in a process called meiosis. Before the pairs of each chromosome are separated to make two new sex cells, sections of genetic material can be swapped between a chromosome-pair to produce chromosomes with unique combinations of genetic material.

The 'crossover' events that cause the genetic material to be swapped are less likely to happen in sections of chromosomes that contain heterozygous genes. However, in a whole population of organisms, the exchange of genetic material between pairs of chromosomes tends to be higher when there are more genetic differences present.

Here, Ziolkowski et al. sought to understand these two seemingly contradictory phenomena by studying crossover events during meiosis in a plant known as *Arabidopsis.* The plants were genetically modified to carry fluorescent proteins that mark when and where crossovers occur. Ziolkowski et al. cross-bred these plants with 32 other varieties of *Arabidopsis*. The experiments show that some of these 'hybrid' plants had higher numbers of crossover events than plants produced from two genetically identical parents, but other hybrid plants had lower numbers of crossovers.

Ziolkowski et al. found that crossovers are more common between heterozygous regions that are close to homozygous regions on the same chromosome. The boundaries between these identical and non-identical regions are important for determining where crossovers take place. The experiments also show that the heterozygous regions have higher levels of interference—where one crossover event prevents other crossover events from happening nearby on the chromosome. In future, using chromosomes with varying patterns of heterozygosity may shed light on how this interference works.

intermediate (*Hunter and Kleckner, 2001*). The invading ssDNA 3′-ends undergo DNA synthesis using the homologous duplex as a template and after second-end capture forms double Holliday junctions (dHJs) (*Szostak et al., 1983*; *Schwacha and Kleckner, 1995*). The dHJs can then be resolved as crossovers, which are cytologically evident as chiasmata (*Page and Hawley, 2003*; *Janssens et al., 2012*). Chiasmata hold chromosomes together and ensure that homologous pairs segregate to opposite cell poles, so that gametes inherit a balanced chromosome number (*Page and Hawley, 2003*).

Crossover numbers are under tight control, with many eukaryote species experiencing 1–2 per chromosome, despite large variation in genome size (*Villeneuve and Hillers, 2001*; *Smukowski and Noor, 2011*; *Henderson, 2012*; *Mercier et al., 2014*). In *Arabidopsis* ~200 DSBs form per meiosis and proceed to form strand invasion intermediates, of which ~10 are repaired as crossovers, with the excess being repaired as non-crossovers, or via intersister recombination (*Giraut et al., 2011*; *Ferdous et al., 2012*; *Lu et al., 2012*; *Sun et al., 2012*; *Yang et al., 2012*; *Drouaud et al., 2013*; *Wijnker et al., 2013*; *Qi et al., 2014*). 80–85% of wild type crossovers are dependent on the ZMM pathway (*MSH4, MSH5, MER3, HEI10, ZIP4, SHOC1, PTD*) and show interference, that is, they are spaced further apart than expected at random (*Copenhaver et al., 2002*; *Higgins et al., 2004*, *2008a*; *Chen et al., 2005*; *Mercier et al., 2005*; *Chelysheva et al., 2007*, *2010*, *2012*; *Macaisne et al., 2008*). The remaining minority of crossovers are non-interfering and require *MUS81* (*Berchowitz et al., 2007*; *Higgins et al., 2008b*). However, as chiasmata are still observed in *msh4 mus81* double mutants, additional crossover pathways must exist (*Higgins et al., 2008b*). The majority of interhomolog strand invasion intermediates are dissolved by the FANCM helicase, which acts with the MHF1 and MHF2 co-factors (*Crismani et al., 2012*; *Knoll et al., 2012*; *Girard et al., 2014*). Mutations in *FANCM*, *MHF1* and *MHF2* cause dramatic increases in non-interfering crossovers (*Crismani et al., 2012*; *Knoll et al., 2012*; *Girard et al., 2014*). It is presently unclear whether non-interfering crossovers occurring in *fancm* are generated by the same pathway as in wild type, as a direct test of *MUS81* dependence is precluded by *fancm mus81* lethality (*Crismani et al., 2012*;

*Knoll et al., 2012*). Both crossovers and non-crossovers can be accompanied by gene conversion events, which in the case of non-crossovers form via the synthesis-dependent strand annealing pathway (*Allers and Lichten, 2001*; *McMahill et al., 2007*; *Lu et al., 2012*; *Sun et al., 2012*; *Yang et al., 2012*; *Drouaud et al., 2013*; *Wijnker et al., 2013*; *Qi et al., 2014*).

Meiotic recombination is sensitive to DNA polymorphism between homologous chromosomes, that is, heterozygosity. For example, insertion-deletion (indel) and single nucleotide polymorphisms (SNPs) suppress crossovers at the scale of hotspots (kb) in fungi, plants and mammals (*Dooner, 1986*; *Borts and Haber, 1987*; *Jeffreys and Neumann, 2005*; *Baudat and de Massy, 2007*; *Cole et al., 2010*). This is thought to occur due to heteroduplex base-pair mismatches inhibiting recombination, following interhomolog strand invasion. Large scale chromosome rearrangements, such as inversions or translocations, also suppress crossovers (*Schwander et al., 2014*; *Thompson and Jiggins, 2014*). Despite the inhibitory effects of polymorphism on crossovers, nucleotide diversity and population-scaled recombination estimates are positively correlated in many plant and animal genomes (*Begun and Aquadro, 1992*; *Hellmann et al., 2003*; *Spencer et al., 2006*; *Gore et al., 2009*; *Paape et al., 2012*; *Cutter and Payseur, 2013*). For example, linkage disequilibrium-based crossover estimates and sequence diversity ($\pi$) are positively correlated in *Arabidopsis* at varying physical scales (*Figure 1A* and *Table 1*) (*Cao et al., 2011*; *Choi et al., 2013*). Multiple processes contribute to these relationships. For example, positive or negative directional selection can reduce diversity at linked sites, with a greater effect in regions of low recombination, known as hitchhiking and background selection (*Hill and Robertson, 1966*; *Hudson and Kaplan, 1995*; *Nordborg et al., 1996*; *Smith and Haigh, 2007*; *Cutter and Payseur, 2013*; *Campos et al., 2014*). These phenomena will cause regions of low recombination under selection to have low diversity, consistent with data in *Drosophila* (*Aguade et al., 1989*; *Begun and Aquadro, 1992*; *Wiehe and Stephan, 1993*; *Campos et al., 2014*). Recombination may also be mutagenic and increase diversity, for example via mismatch repair enzymes showing a mutational bias for A:T > G:C transversions (*Duret and Galtier, 2009*; *Webster and Hurst, 2012*; *Glémin et al., 2014*).

Here we use natural variation in *Arabidopsis* to directly investigate the influence of heterozygosity on meiotic recombination. Extensive evidence exists for *cis* and *trans* modification of crossover frequency by plant genetic variation (*Barth et al., 2001*; *Yao and Schnable, 2005*; *Yandeau-Nelson et al., 2006*; *Esch et al., 2007*; *McMullen et al., 2009*; *López et al., 2012*; *Salomé et al., 2012*; *Bauer et al., 2013*). We define *trans* modifiers as loci encoding diffusible molecules that control recombination on other chromosomes, and elsewhere on the same chromosome, as exemplified by mammalian *PRDM9* (*Baudat et al., 2010*; *Berg et al., 2010*; *Myers et al., 2010*; *Parvanov et al., 2010*; *Fledel-Alon et al., 2011*; *Sandor et al., 2012*; *Kong et al., 2013*). We define *cis* modification as variation that influences recombination only on the same chromosome, for example, the inhibitory effects of high SNP density, inversions and translocations (*Dooner, 1986*; *Borts and Haber, 1987*; *Jeffreys and Neumann, 2005*; *Baudat and de Massy, 2007*; *Cole et al., 2010*; *Schwander et al., 2014*; *Thompson and Jiggins, 2014*). Regional patterns of chromatin and epigenetic information can also cause significant *cis* effects, for example loss of either H2A.Z deposition or DNA methylation alters crossover frequency in *Arabidopsis* (*Colomé-Tatché et al., 2012*; *Melamed-Bessudo and Levy, 2012*; *Mirouze et al., 2012*; *Yelina et al., 2012*; *Choi et al., 2013*).

In this study we crossed *Arabidopsis* lines carrying fluorescent crossover reporters generated in a common background (Col-0) to 32 diverse accessions. We observed extensive variation in F$_1$ hybrid recombination rates, with both significantly higher and lower crossovers than homozygous backgrounds. We further analysed Col × Ct F$_2$ recombinant populations using three independent crossover reporter intervals (*420*, *CEN3* and *I2f*). We did not detect *trans* modifiers in these crosses, but observed a novel *cis* modification effect caused by heterozygosity. Specifically, juxtaposition of heterozygous and homozygous regions is associated with increased crossover frequency in the heterozygous region and a reciprocal decrease in the homozygous region. To investigate this phenomenon mechanistically we repeated analysis in mutants where the balance of interfering and non-interfering crossover repair is altered (*fancm*, *zip4* and *fancm zip4*). This analysis demonstrates that remodelling of crossovers across heterozygosity/homozygosity junctions is dependent on interference. We also show that the non-interfering repair is less efficient in heterozygous regions. As a consequence, interference measurements are stronger in heterozygous regions. Our findings show how varying polymorphism patterns can differentially influence meiotic recombination along chromosomes.

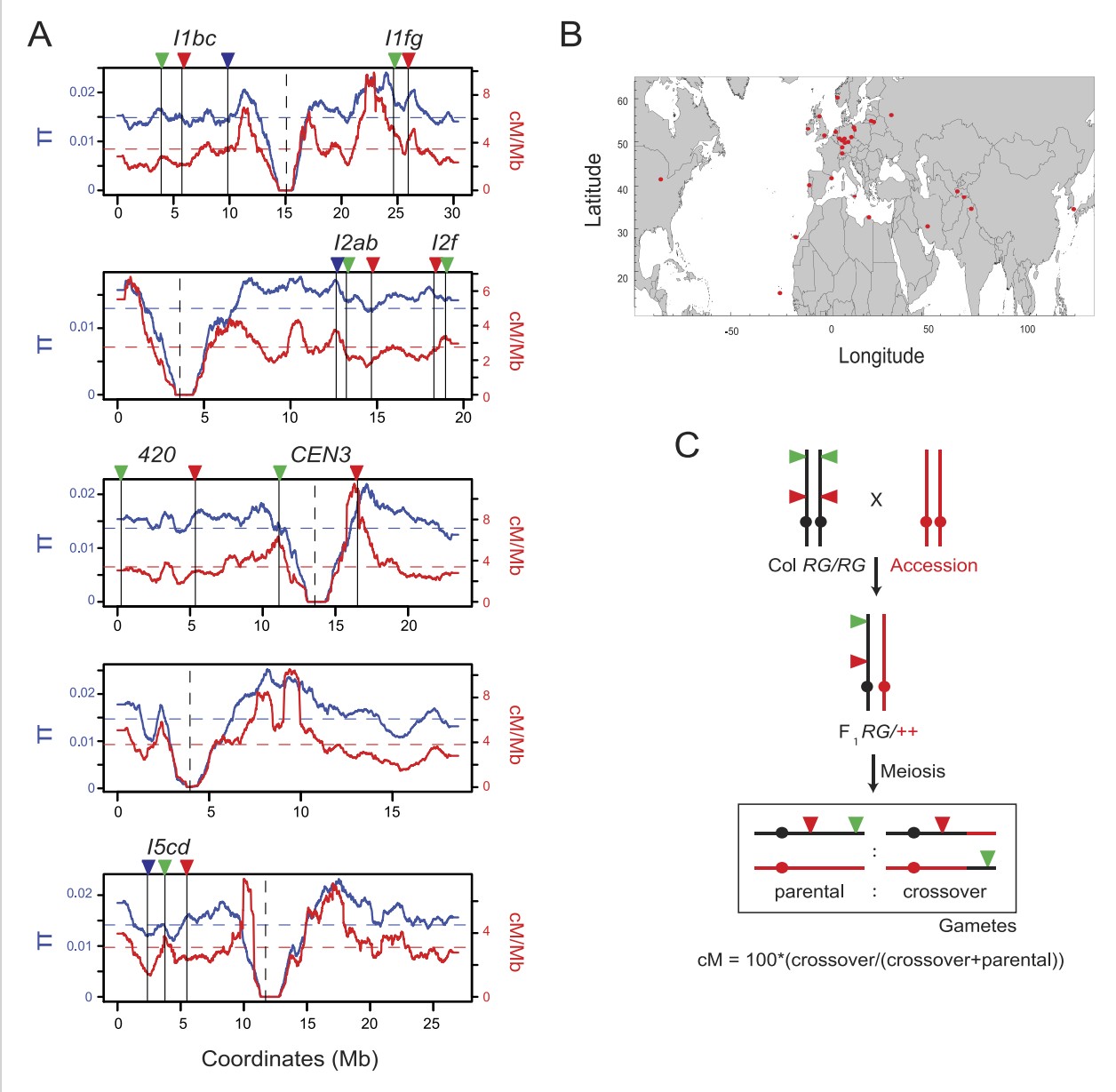

**Figure 1**. Testing for crossover modification by Arabidopsis natural variation. (**A**) Historical crossover frequency (red, cM/Mb) and sequence diversity (π, blue) along the physical length of the *Arabidopsis thaliana* chromosomes (Mb) (*Cao et al., 2011*; *Choi et al., 2013*). Mean values are indicated by horizontal dotted lines and centromeres by vertical dotted lines. The fluorescent crossover intervals analysed are indicated by solid vertical lines and coloured triangles. (**B**) Map showing the geographical origin of the *Arabidopsis* accessions studied, indicated by red points. (**C**) Genetic diagram illustrating the experimental approach with a single chromosome shown for simplicity. Fluorescent crossover reporters (triangles) were generated in the Col background (black) and crossed to accessions of interest (red) to generate $F_1$ heterozygotes. Following meiosis the proportion of parental:crossover gametes from $F_1$ heterozygotes was analysed to measure genetic distance (cM) between the fluorescent protein encoding transgenes.

## Results

### Heterozygosity extensively modifies crossover frequency in *Arabidopsis*

To test the effect of heterozygosity on meiotic recombination we crossed transgenic *Arabidopsis* with fluorescent crossover reporters generated in the Col-0 background to 32 diverse accessions that represent global genetic diversity within this species (*Figure 1*, *Tables 2, 3*) (*Melamed-Bessudo et al., 2005*; *Berchowitz and Copenhaver, 2008*; *Yelina et al., 2013*). The 5 intervals tested (*I1b*, *I1fg*, *I2f*,

**Table 1.** Correlations between historical recombination and sequence diversity at varying physical scales

| Scale ($\pi$) | Chr1 | Chr2 | Chr3 | Chr4 | Chr5 |
|---|---|---|---|---|---|
| 5 kb | 0.521 | 0.301 | 0.545 | 0.575 | 0.541 |
| 10 kb | 0.556 | 0.305 | 0.565 | 0.602 | 0.562 |
| 50 kb | 0.657 | 0.381 | 0.579 | 0.692 | 0.619 |
| 100 kb | 0.699 | 0.563 | 0.601 | 0.744 | 0.646 |
| 500 kb | 0.741 | 0.528 | 0.615 | 0.841 | 0.653 |
| 1 Mb | 0.639 | 0.504 | 0.683 | 0.846 | 0.624 |
| Scale ($\theta$) | Chr1 | Chr2 | Chr3 | Chr4 | Chr5 |
| 5 kb | 0.537 | 0.298 | 0.557 | 0.585 | 0.553 |
| 10 kb | 0.569 | 0.303 | 0.576 | 0.610 | 0.572 |
| 50 kb | 0.662 | 0.382 | 0.592 | 0.699 | 0.623 |
| 100 kb | 0.710 | 0.573 | 0.617 | 0.752 | 0.650 |
| 500 kb | 0.754 | 0.534 | 0.635 | 0.844 | 0.655 |
| 1 Mb | 0.647 | 0.504 | 0.697 | 0.849 | 0.635 |

Spearman's rank correlation between historical crossover frequency estimates from LDhat and sequence diversity ($\theta$ and $\pi$) at varying physical scales (*Cao et al., 2011*; *Choi et al., 2013*). Adjacent windows of the indicated physical size were used for correlations.

*420* and *CEN3*) range from 0.67–5.40 megabases (Mb), represent 11.5% of the genome (14.34 Mb) in total and are located in sub-telomeric, interstitial and centromeric regions (*Figure 1A* and *Table 2*). The intervals vary in experimental recombination rate, with the centromeric interval *CEN3* being the lowest (2.11 cM/Mb) and the sub-telomeric interval *I2f* being the highest (13.02 cM/Mb) (*Table 2*). As *Arabidopsis* male meiosis shows elevated sub-telomeric recombination, this likely contributes to the high *I2f* crossover frequency, which is measured in pollen (*Giraut et al., 2011*). Low recombination in *CEN3* is also expected, as the centromeric regions are heterochromatic and known to show suppressed crossover frequency (*Figure 1A*) (*Copenhaver et al., 1999*; *Giraut et al., 2011*; *Salomé et al., 2012*; *Yelina et al., 2012*). To asses relative heterozygosity levels we analysed pairwise sequence differences relative to Col-0 using the 19 genomes dataset, which was generated from a subset of the accessions used in our crosses (*Gan et al., 2011*). *CEN3* shows the highest heterozygosity levels, followed by the interstitial and sub-telomeric intervals (*Table 2*). Therefore, the regions analysed represent diverse chromosomal environments with varying levels of recombination and inter-accession sequence polymorphism.

The crossover reporter systems utilize fluorescent proteins encoded by linked, heterozygous transgenes that are expressed from the pollen-specific *LAT52*, or seed-specific *NapA* promoters (*Melamed-Bessudo et al., 2005*; *Francis et al., 2006*; *Yelina et al., 2013*). Fluorescent measurements of gametes or progeny are used to asses segregation of the transgenes through meiosis and thereby measure crossover rates (*Melamed-Bessudo et al., 2005*; *Berchowitz and Copenhaver, 2008*; *Yelina et al., 2013*). Previously, we developed flow cytometry protocols to increase scoring-throughput using fluorescent pollen, allowing up to 80,000 gametes to be scored per individual plant (*Yelina et al., 2012*, *2013*). To increase throughput when measuring fluorescent seed we adapted CellProfiler image analysis software, allowing us to rapidly score ~2000 seed per individual (*Figure 2A–F*) (*Carpenter et al., 2006*). This method gives recombination measurements not significantly different from manually collected data (*Figure 2F*, *Figure 2—source data 1*) (generalized linear model (GLM), hereafter GLM, p = 0.373). To test for significant differences between recombinant and non-recombinant counts using replicate groups we used a GLM assuming a binomial count distribution. Replicate heterozygous $F_1$ individuals were analysed for each cross and 13,264,943 gametes were scored in total, to provide an extensive survey of the influence of polymorphism heterozygosity on crossover frequency (*Figure 3* and *Table 3*).

We observed substantial variation in crossovers between $F_1$ crosses, although the interstitial intervals varied less than those in sub-telomeric and centromeric locations (*Figure 3A–E*, *Figure 3—source data 1–5*). $F_1$ heterozygotes showed both significantly higher and lower total recombination compared to Col homozygotes (*Figure 3* and *Table 3*) (GLM with 113° of freedom p < 2.0 × 10$^{-16}$). $F_1$ genetic distances and polymorphism levels within the intervals were poorly correlated, consistent with previous observations (*Table 4*) (*Barth et al., 2001*; *Gan et al., 2011*; *Salomé et al., 2012*). This weak correlation may be partially explained by unknown structural rearrangements. For example, the Shahdara (Sha) accession has a sub-telomeric inversion (3–5.1 Mb) on chromosome 3 relative to Col (*Loudet et al., 2002*; *Simon et al., 2008*; *Salomé et al., 2012*), and Col/Sha $F_1$s show consistently low crossovers in *420*, which overlaps the inversion (*Figure 3D* and *Table 3*). Hence the contribution of unknown structural polymorphisms to variation in recombination rates could be significant. Further evidence of the complex effect of polymorphism is evident from the

**Table 2**. Fluorescent crossover reporter intervals

| Interval | Chr | Method | T-DNA 1 | T-DNA 2 | Mb | Location | cM/Mb (Col-0) | cM/Mb ($F_1$) | Heterozygosity |
|----------|-----|--------|---------|---------|-----|----------|---------------|---------------|----------------|
| *I1b* | 1 | Pollen | 3,905,441-YFP | 5,755,618-dsRed2 | 1.85 | Interstitial | 4.25 | 4.05 | 1.93 (3.16) |
| *I1c* | 1 | Pollen | 5,755,618-dsRed2 | 9,850,022-CFP | 4.09 | Interstitial | 4.55 | N/A | 2.80 (3.16) |
| *I1fg* | 1 | Pollen | 24,645,163-YFP | 25,956,590-dsRed2 | 1.31 | Interstitial | 6.20 | 6.02 | 2.52 (3.16) |
| *I2a* | 2 | Pollen | 12,640,092-CFP | 13,226,013-YFP | 0.59 | Interstitial | 5.19 | N/A | 2.33 (3.30) |
| *I2b* | 2 | Pollen | 13,226,013-YFP | 14,675,407-dsRed2 | 1.45 | Interstitial | 3.09 | N/A | 1.53 (3.30) |
| *I2f* | 2 | Pollen | 18,286,716-dsRed2 | 18,957,093-YFP | 0.67 | Sub-telomeric | 13.02 | 17.41 | 1.43 (3.30) |
| *420* | 3 | Seed | 256,516-GFP | 5,361,637-dsRed2 | 5.11 | Sub-telomeric | 3.70 | 2.93 | 1.19 (3.37) |
| *CEN3* | 3 | Pollen | 11,115,724-YFP | 16,520,560-dsRed2 | 5.40 | Centromeric | 2.11 | 2.38 | 6.69 (3.37) |
| *I3b* | 3 | Pollen | 498,916-CFP | 3,126,994-YFP | 2.63 | Sub-telomeric | 5.99 | N/A | 1.11 (3.37) |
| *I3c* | 3 | Pollen | 3,126,994-YFP | 4,319,513-dsRed2 | 1.19 | Sub-telomeric | 4.01 | N/A | 1.64 (3.37) |
| *I5c* | 5 | Pollen | 2,372,623-CFP | 3,760,756-YFP | 1.39 | Interstitial | 4.01 | N/A | 1.01 (3.27) |
| *I5d* | 5 | Pollen | 3,760,756-YFP | 5,497,513-dsRed2 | 1.74 | Interstitial | 3.20 | N/A | 1.56 (3.27) |

The interval name is listed together with chromosome, method of scoring and location of the flanking T-DNAs together with the fluorescent proteins they encode. Interval cM/Mb values from Col-0 homozygous are listed (Col-0), in addition to the mean cM/Mb observed across all $F_1$ crosses ($F_1$). Heterozygosity values were calculated using pairwise comparison of polymorphism data from the 19 genomes project to the Col reference (**Gan et al., 2011**), and the mean value for the interval shown, in addition to the mean chromosome value in parentheses.

*CEN3* interval, which spans the repetitive and structurally diverse centromeric region of chromosome 3 (**Figure 1A**) (**Copenhaver et al., 1999**; **Clark et al., 2007**; **Ito et al., 2007**; **Cao et al., 2011**; **Gan et al., 2011**; **Horton et al., 2012**; **Long et al., 2013**), and showed high variability in $F_1$ crossover frequency (**Figure 3E** and **Table 3**). Unexpectedly, some of the most diverged crosses, for example two accessions from Atlantic islands Cvi-0 and Can-0, showed highest *CEN3* crossovers (**Figure 3E** and **Table 3**) (**Ito et al., 2007**). 10 of 26 $F_1$s showed significantly higher summed crossover frequency compared with Col homozygotes, consistent with previous reports that recombination can increase in heterozygous backgrounds in *Arabidopsis* (**Barth et al., 2001**) (**Figure 3F** and **Table 3**). Both *cis* and *trans* modification of crossovers by genetic variation has been observed in plants (**Barth et al., 2001**; **Yao and Schnable, 2005**; **Yandeau-Nelson et al., 2006**; **Esch et al., 2007**; **McMullen et al., 2009**; **López et al., 2012**; **Salomé et al., 2012**; **Bauer et al., 2013**). Therefore, the variation in $F_1$ crossover frequency observed here is likely caused by complex interactions between *cis* and *trans* modifying effects.

## Modification of crossover frequency by juxtaposition of heterozygosity and homozygosity

To investigate the extent of *cis* and *trans* modification of crossover frequency by heterozygosity we generated a *420* Col × Ct recombinant $F_2$ population (n = 139) (**Figure 4A**). We selected $F_2$ individuals that were heterozygous for linked T-DNAs expressing red and green fluorescent proteins and Col/Ct heterozygous within *420*, but genetically mosaic elsewhere in the genome (**Figure 4A,E**). The *420/++* Col/Ct $F_2$ population showed significantly greater variation in recombination rates than Col/Col homozygotes (F-test p = 0.0129) (**Figure 4D**, **Figure 4—source data 1**). We genotyped 51 Col/Ct markers throughout the genome and tested for their association with *420* crossover frequency using QTL analysis. We detected no association using markers on chromosomes 1, 2, 4 or 5 (**Figure 4B**). However, on chromosome 3 itself homozygosity (Col/Col or Ct/Ct) outside of *420* was associated with high recombination (FDR corrected chi-square test p = $3.29 \times 10^{-31}$) (**Figure 4B,E–F** and **Table 5**). For each marker we used the heterozygous and homozygous counts in the hottest quartile vs the coldest quartile to construct 2 × 2 contingency tables and performed chi-square tests, followed by FDR correction for multiple testing (**Table 5**).

To test an additional chromosome for the effect of heterozygosity/homozygosity juxtaposition we measured recombination in an *I2f* Col × Ct $F_2$ population (n = 78) (**Figure 4G–I**). The *I2f* interval is 0.67 Mb and located sub-telomerically on the long arm of chromosome 2 (**Figure 1A** and **Table 2**). The *I2f/*

**Table 3.** Genetic distance in $F_1$ heterozygotes

| Accession | Location | I1b | I1fg | I2f | 420 | CEN3 | Total | P |
|---|---|---|---|---|---|---|---|---|
| Tsu-0 | Tsushima, Japan | 6.6 | 6.3 | 6.9 | 14.5 | 9.4 | 43.7 | $<2.00 \times 10^{-16}$ |
| Hi-0 | Hilversum, Netherlands | 6.8 | 6.9 | 6.9 | 13.6 | 9.6 | 43.8 | $<2.00 \times 10^{-16}$ |
| Wil-2 | Vilnius, Lithuania | 6.1 | 6.9 | 6.1 | 15.9 | 10.1 | 45.0 | $<2.00 \times 10^{-16}$ |
| Kn-0 | Kaunas, Lithuania | 7.4 | 6.6 | 8.0 | 15.5 | 8.7 | 46.2 | $<2.00 \times 10^{-16}$ |
| Ler-0 | Gorzow, Poland | 6.6 | 8.2 | 7.6 | 12.3 | 11.9 | 46.6 | $<2.00 \times 10^{-16}$ |
| Ws-0 | Vassilyevichy, Belarus | 6.7 | 7.7 | 10.2 | 13.0 | 9.0 | 46.6 | $<2.00 \times 10^{-16}$ |
| No-0 | Nossen, Germany | 7.4 | 7.9 | 6.7 | 14.1 | 11.4 | 47.4 | $<2.00 \times 10^{-16}$ |
| Wu-0 | Wurzburg, Germany | 7.6 | 6.3 | 9.5 | 14.0 | 11.4 | 48.8 | $<2.00 \times 10^{-16}$ |
| Zu-0 | Zurich, Switzerland | 7.5 | 7.1 | 13.4 | 12.2 | 9.9 | 50.1 | 0.0438 |
| Po-0 | Poppelsdorf, Germany | 7.2 | 7.9 | 9.1 | 15.8 | 10.9 | 51.0 | 0.000484 |
| Ct-1 | Catania, Italy | 7.8 | 8.7 | 7.2 | 15.9 | 12.1 | 51.7 | $9.27 \times 10^{-08}$ |
| Oy-0 | Oystese, Norway | 7.7 | 8.4 | 8.5 | 15.7 | 12.5 | 52.8 | 0.969 |
| Rsch-4 | Rschew, Russia | 7.9 | 6.8 | 10.7 | 15.2 | 12.4 | 53.1 | 0.505 |
| Col-0 | Columbia, USA | 8.0 | 8.2 | 8.8 | 18.0 | 11.5 | 54.5 | – |
| Sf-2 | San Feliu, Spain | 8.2 | 8.8 | 7.4 | 18.6 | 12.3 | 55.3 | 0.724 |
| Kas | Kashmir, India | 6.9 | 8.6 | 13.2 | 13.8 | 13.3 | 55.8 | $<2.00 \times 10^{-16}$ |
| Kond | Pugus, Tajikistan | 7.1 | 8.1 | 15.8 | 13.7 | 11.4 | 56.2 | $<2.00 \times 10^{-16}$ |
| Edi-0 | Edinburgh, Scotland | 8.0 | 8.0 | 13.4 | 13.3 | 13.6 | 56.3 | $<2.00 \times 10^{-16}$ |
| Bay-0 | Bayreuth, Germany | 8.6 | 8.3 | 11.3 | 18.6 | 11.5 | 58.3 | $<2.00 \times 10^{-16}$ |
| Mt-0 | Martuba, Libya | 9.6 | 7.8 | 13.2 | 20.6 | 9.6 | 60.8 | $<2.00 \times 10^{-16}$ |
| Sha | Pamiro-Alaya, Tajikistan | 7.8 | 7.5 | 20.0 | 7.0 | 18.6 | 60.9 | 0.0012 |
| C24 | Columbia, USA | 8.8 | 8.5 | 18.5 | 12.1 | 14.1 | 61.9 | $<2.00 \times 10^{-16}$ |
| Bur-0 | Burren, Ireland | 6.7 | 9.1 | 21.9 | 14.7 | 17.8 | 70.2 | $<2.00 \times 10^{-16}$ |
| Cvi-0 | Cape Verde Islands | 9.1 | 10.0 | 11.3 | 12.6 | 27.6 | 70.7 | $<2.00 \times 10^{-16}$ |
| Can-0 | Las Palmas, Canary Isles | 7.8 | 8.5 | 22.1 | 12.4 | 31.4 | 82.2 | $<2.00 \times 10^{-16}$ |
| Co | Coimbra, Portugal | – | – | – | 11.1 | 13.8 | – | – |
| Nw-0 | Neuweilnau, Germany | – | – | – | 14.7 | 14.4 | – | – |
| Mh-0 | Szczecin, Poland | – | – | – | 14.9 | 10.1 | – | – |
| Wl-0 | Wildbad, Germany | – | – | – | 17.0 | 9.5 | – | – |
| Bu-0 | Burghaun, Germany | – | – | – | 28.9 | 8.8 | – | – |
| CIBC5 | Ascot, United Kingdom | – | – | – | 13.2 | 11.3 | – | – |
| RRS7 | North Liberty, USA | – | – | – | 17.2 | 11.7 | – | – |
| | $F_1$ cM mean | 7.6 | 7.9 | 11.5 | 15.0 | 12.9 | 54.8 | |
| | cM StDev | 0.8 | 0.9 | 4.8 | 3.6 | 4.9 | 9.3 | |

The accessions crossed to are listed with their geographic location. Genetic distance (cM) data is shown for the five fluorescent intervals, in addition to a summed total. Also shown are the mean and standard deviation for all $F_1$s. A generalized linear model (GLM) was used to test for significant differences between total recombinant vs non-recombinant counts between replicate groups of Col-0 homozygotes and $F_1$ heterozygotes. Tests were performed for genotypes where data from all five tested intervals had been collected.

++ Col/Ct $F_2$ population also showed significantly greater variation in recombination rates than Col/Col homozygotes (F-test, p = 0.04) (*Figure 4G*, *Figure 4—source data 2*). We performed QTL analysis for Col/Ct markers on chromosomes 2 and 3 and again observed a significant effect on the same chromosome and no *trans* effect from chromosome 3. An identical trend to that seen for *420* was

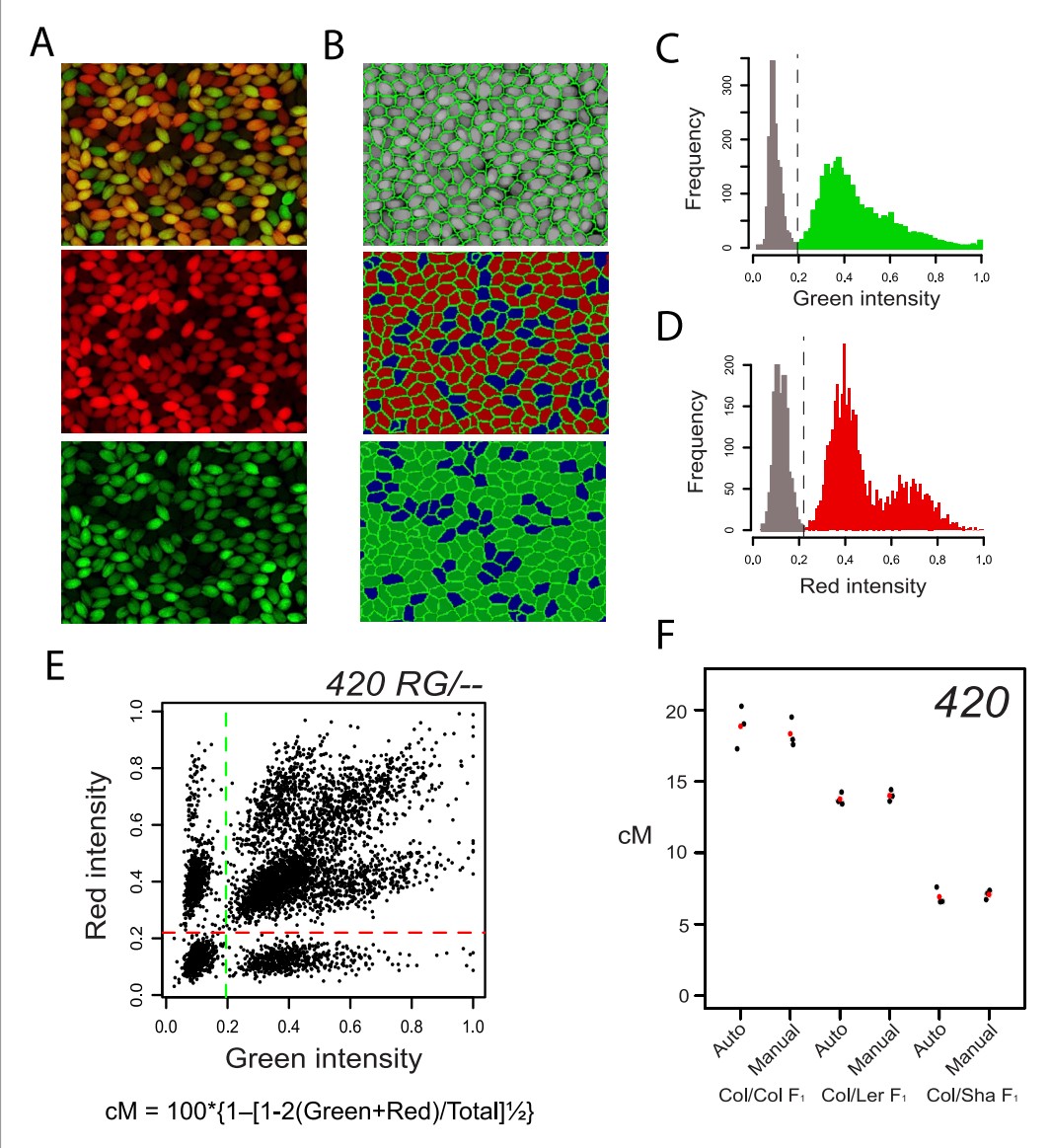

**Figure 2**. High-throughput measurement of crossover frequency using image analysis of fluorescent seed. (**A**) Combined red and green, red alone and green alone fluorescent micrographs of seed from a self-fertilized *420/++* plant. (**B**) CellProfiler output showing identification of seed objects by green lines and scoring of red and green fluorescence shown by shading. Blue shading shows an absence of colour. (**C**–**D**) Histograms of seed object fluorescence intensities, with coloured and non-coloured seed divided by vertical dotted lines. (**E**) Plot of seed object red vs green fluorescence intensities, with each point representing an individual seed. The red and green dashed lines show the colour vs non-colour divisions indicated in (**C**–**D**). The formula used for cM calculation is printed below. (**F**) *420* cM measurements from replicate plants of the indicated genotypes (Col/Col $F_1$, Col/Ler $F_1$, Col/Sha $F_1$) are shown by black dots with mean values indicated by red dots. Data generated by automatic and manual scoring are plotted alongside one another. Measurements made by the different methods were not significantly different as tested using generalized linear model (GLM). See *Figure 2—source data 1*.

The following source data and figure supplement are available for figure 2:

**Source data 1**. *420* crossover frequency measured via manual or automated scoring of seed fluorescence.

**Figure supplement 1**. Distinguishing *420 RFP-GFP/++* vs *RFP-+/+-GFP* recombinant individuals.

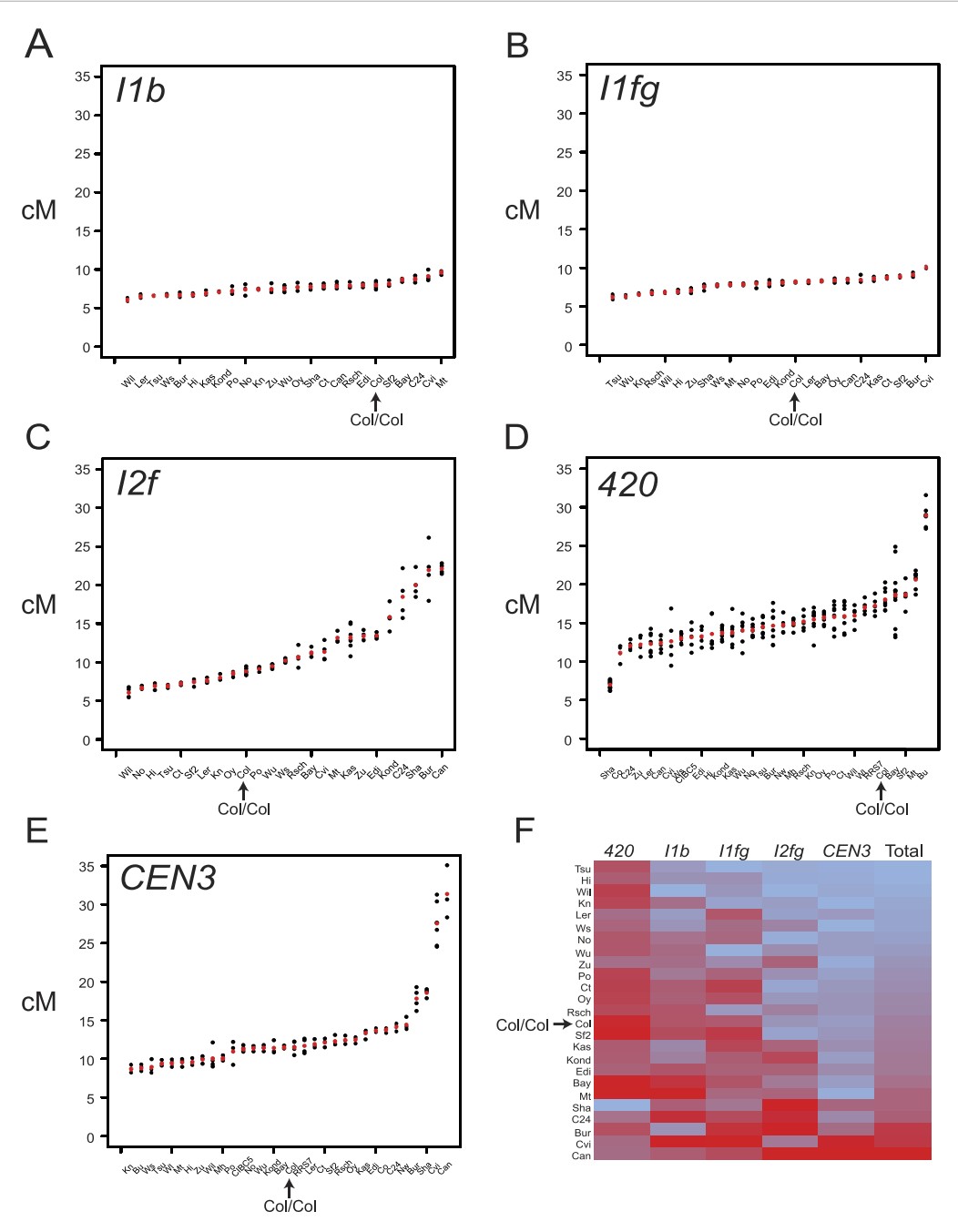

**Figure 3**. Variation in $F_1$ hybrid crossover frequency. (**A–E**) Genetic distance (cM) measurements for fluorescent crossover intervals *I1b*, *I1fg*, *I2f*, *420* and *CEN3* with individual replicates (black dots) and mean values (red dots) for the crosses labelled on the x-axis. See *Figure 3—source data 1–5*. (**F**) Heatmap summarising crossover frequency data for $F_1$ crosses with data from all five intervals. Accessions are listed as rows and fluorescent intervals listed as columns. The heatmap is ordered according to ascending 'Total' cM (red = highest, blue = lowest), which is the sum of the individual interval genetic distances. GLM testing for significant differences between total recombinant vs non-recombinant counts between replicate groups of Col-0 homozygotes and $F_1$ heterozygotes was performed, for genotypes where data from all five tested intervals were collected (*Table 3*). Col/Col homozygous data are labelled and highlighted with an arrow in each plot.

*Figure 3. continued on next page*

*Figure 3. Continued*
The following source data are available for figure 3:
**Source data 1**. *I1b* F$_1$ flow cytometry count data.
**Source data 2**. *I1b* F$_1$ flow cytometry count data.
**Source data 3**. *I1b* F$_1$ flow cytometry count data.
**Source data 4**. *I1b* F$_1$ flow cytometry count data.
**Source data 5**. *CEN3* F$_1$ flow cytometry count data.

observed, with the highest recombination F$_2$ quartile showing significantly greater marker homozygosity (both Col/Col and Ct/Ct) outside *I2f* on chromosome 2 (FDR corrected chi-square test p = 1.44 × 10$^{-10}$) (*Figure 4C,G–I* and *Table 6*). The most distal marker showing a significant difference between hot and cold quartiles was of comparable megabase distance for *420* (10.60 Mb) and *I2f* (10.12 Mb).

To test whether the effect of heterozygosity/homozygosity juxtaposition is dependent on chromosomal location we measured crossovers in a *CEN3* Col × Ct F$_2$ population (n = 121) (*Figures 4A* and *5C*, *Figure 4—figure supplement 1*, *Figure 4—source data 3*). As for *420* and *I2f*, *CEN3* F$_2$ recombination rates were significantly more variable than Col/Col homozygotes (F-test p = 0.01268) (*Figure 4A*, *Figure 4—figure supplement 1*). We genotyped 9 Col/Ct markers on chromosome 3 and observed that 5 markers in proximity to *CEN3* were significantly more homozygous in the hottest compared to the coldest F$_2$ quartile (FDR corrected chi-square test p = 1.14 × 10$^{-07}$) (*Figure 4D–F*, *Figure 4—figure supplement 1* and *Table 7*). The physical extent of the effect was less (2.62 Mb) on the long arm of chromosome 3 for *CEN3* than observed for *420* and *I2f*, potentially due to heterozygosity effects acting independently from both arms across the centromere. Together this shows that juxtaposition of heterozygous and homozygous regions in various chromosomal locations can modify local crossover frequency.

## Juxtaposed heterozygous and homozygous regions show reciprocal changes in crossover frequency

We reasoned that if heterozygous regions increase recombination when juxtaposed with homozygous regions, then the linked homozygous regions may show compensatory decreases, due to crossover interference (*Copenhaver et al., 2002*; *Zhang et al., 2014a*). To test this idea we constructed a three-colour pollen FTL interval termed *I3bc* that overlaps the *420* seed interval on chromosome 3 (*Figure 5* and *Table 2*). Three-colour FTL configurations allow simultaneous measurement of crossover frequency in adjacent intervals and measurement of crossover interference (*Berchowitz and Copenhaver, 2008*; *Yelina et al., 2013*) (*Figure 5—figure supplement 1*). To calculate interference, the observed double crossover (DCO) classes (N$_{-Y-}$ + N$_{B-R}$) are compared to the number expected in the absence of interference: (*I3b* cM/100) × (*I3c* cM/100) × N$_{total}$ (*Figure 5A*). The Coefficient of Coincidence (CoC) is calculated by dividing Observed DCOs by Expected DCOs, and interference strength calculated as 1-CoC (*Figure 5A*).

*I3bc* wild type genetic distance was greater than that measured from *420* self-fertilization data, as expected due to increases observed in sub-telomeric regions in male meiosis (*Table 2—Figure 5—source data 1*) (*Giraut et al., 2011*). *I3b* crossover frequency was also higher than *I3c*, consistent with a telomeric gradient in male crossover frequency (*Figure 5B* and *Table 2*) (*Giraut et al., 2011*). We compared crossovers in plants that were entirely Col homozygous (HOM-HOM) vs plants that were Col/Ct heterozygous within *I3b*, but Col/Col homozygous in *I3c* and for the rest of chromosome 3 (HET-HOM) (*Figure 5A*). Dense genotyping markers were used to confirm the location of homozygous and heterozygous regions (*Figure 5A*). We observed that *I3b* crossovers significantly increased in HET-HOM compared to HOM-HOM plants, and there was a reciprocal decrease in *I3c* crossovers (*Figure 5B*, *Figure 5—source data 2*) (both GLM p < 2.0 × 10$^{-16}$). Together this is consistent with reciprocal crossover changes in juxtaposed heterozygous and homozygous regions being driven by crossover interference.

**Table 4**. F$_1$ heterozygosity levels relative to Col-0

| Accession | Chr 1 | I1b | I1fg | Chr 2 | I2f | Chr 3 | 420 | CEN3 | Chr 4 | Chr 5 |
|---|---|---|---|---|---|---|---|---|---|---|
| Bur-0 | 3.35 | 1.86 | 3.62 | 3.60 | 1.51 | 3.58 | 1.57 | 6.20 | 3.89 | 3.16 |
| Can-0 | 3.75 | 2.99 | 3.51 | 3.92 | 0.92 | 3.98 | 1.02 | 8.27 | 5.34 | 4.24 |
| Ct-1 | 2.62 | 1.67 | 2.29 | 2.61 | 1.85 | 3.35 | 0.96 | 6.91 | 3.23 | 3.36 |
| Edi-0 | 3.30 | 1.91 | 3.64 | 3.26 | 0.91 | 3.05 | 1.34 | 5.48 | 3.42 | 3.81 |
| Hi-0 | 2.43 | 1.59 | 1.87 | 1.80 | 1.50 | 2.58 | 1.07 | 4.62 | 2.69 | 2.46 |
| Kn-0 | 3.15 | 1.78 | 2.85 | 3.35 | 2.18 | 3.58 | 1.49 | 6.69 | 3.76 | 3.40 |
| Ler-0 | 3.10 | 1.61 | 2.66 | 3.62 | 2.24 | 3.43 | 1.13 | 7.39 | 3.87 | 3.53 |
| Mt-0 | 3.02 | 1.77 | 1.16 | 3.49 | 1.57 | 3.17 | 1.07 | 5.70 | 3.95 | 2.71 |
| No-0 | 3.25 | 2.28 | 2.71 | 3.36 | 1.27 | 3.52 | 1.21 | 7.14 | 3.51 | 3.56 |
| Oy-0 | 3.48 | 1.68 | 2.10 | 3.05 | 0.58 | 2.94 | 1.23 | 6.16 | 2.95 | 2.72 |
| Po-0 | 2.45 | 1.78 | 1.19 | 2.36 | 0.67 | 2.87 | 0.79 | 5.99 | 2.53 | 2.59 |
| Rsch-4 | 2.94 | 1.84 | 1.17 | 3.36 | 1.22 | 3.09 | 1.05 | 5.37 | 3.89 | 3.22 |
| Sf-2 | 3.61 | 1.94 | 4.24 | 3.54 | 2.06 | 3.74 | 1.30 | 8.24 | 3.81 | 3.58 |
| Tsu-0 | 3.37 | 1.68 | 2.36 | 3.69 | 1.39 | 3.98 | 1.14 | 8.78 | 3.69 | 3.05 |
| Wil-2 | 3.56 | 1.99 | 2.45 | 3.77 | 2.11 | 3.81 | 1.56 | 7.55 | 4.44 | 3.34 |
| Ws-0 | 3.25 | 1.93 | 3.54 | 3.68 | 1.58 | 3.30 | 1.30 | 6.65 | 3.70 | 3.41 |
| Wu-0 | 3.13 | 2.53 | 1.95 | 3.14 | 0.67 | 3.50 | 1.22 | 7.41 | 3.36 | 3.15 |
| Zu-0 | 3.10 | 1.85 | 2.02 | 3.83 | 1.43 | 3.19 | 0.96 | 5.84 | 3.38 | 3.64 |
| Mean | 3.16 | 1.93 | 2.52 | 3.30 | 1.43 | 3.37 | 1.19 | 6.69 | 3.63 | 3.27 |
| Correlation (cM) | – | 0.13 (p = 0.61) | 0.47 (p = 0.05) | – | −0.29 (p = 0.23) | – | 0.06 (p = 0.81) | 0.28 (p = 0.26) | – | – |

Accessions sequenced as part of the 19 genomes project were analysed (**Gan et al., 2011**) and heterozygosity calculated as the sum of SNPs and indel lengths divided by the length of region (kb). Correlations were between heterozygosity within the interval measured and F$_1$ cM measurements.

## Reciprocal crossover remodeling across heterozygosity/homozygosity junctions requires interference

The effect of heterozygosity/homozygosity juxtaposition on crossovers extends over megabase distances, which is similar to the scale of crossover interference in *Arabidopsis* (**Copenhaver et al., 2002**; **Giraut et al., 2011**; **Salomé et al., 2012**). We therefore next used mutations in meiotic recombination pathways to analyse the genetic requirements of these effects. Specifically, we generated plants carrying the linked chromosome 3 fluorescent crossover reporters *420* and *CEN3* (*420-CEN3*), with varying Col/Ct genotype and that were wild type, *fancm* or *fancm zip4* (**Figure 6—Figure 6—figure supplement 1**). Crossover frequency in *420* and *CEN3* can be scored in the same individuals, as these intervals use fluorescent proteins expressed in seed and pollen respectively. In *fancm* DSBs that would normally be repaired as non-crossovers enter a non-interfering pathway leading to a substantial increase in crossovers, although the interfering pathway remains active (**Crismani et al., 2012**). In *fancm zip4* only non-interfering crossovers occur, due to mutation of the ZMM gene *ZIP4* (**Chelysheva et al., 2007**; **Crismani et al., 2012**). In wild type, both interfering and non-interfering pathways are active, but interfering crossovers predominate and constitute ~85% of total crossovers (**Copenhaver et al., 2002**; **Higgins et al., 2004**; **Mercier et al., 2005**). Therefore, by comparing genetic distances in wild type, *fancm* and *fancm zip4*, where the relative proportions of interfering and non-interfering repair vary dramatically, we can investigate the sensitivity of different recombination pathways to heterozygosity.

When chromosome 3 is Col/Col homozygous (HOM-HOM) genetic distance in the *420* interval significantly increased in *fancm* and *fancm zip4* mutants compared with wild type (both GLM p < 2.0 × 10$^{-16}$) (**Figure 6A**, **Figure 6—source data 1**), consistent with repair of the majority of DSBs via a non-interfering crossover pathway (**Crismani et al., 2012**). However, the *CEN3* interval experienced a smaller yet significant increase in genetic distance in *fancm* and decreased in *fancm zip4* (both GLM p < 2.0 × 10$^{-16}$), indicating that non-interfering crossover repair is less efficient in this region

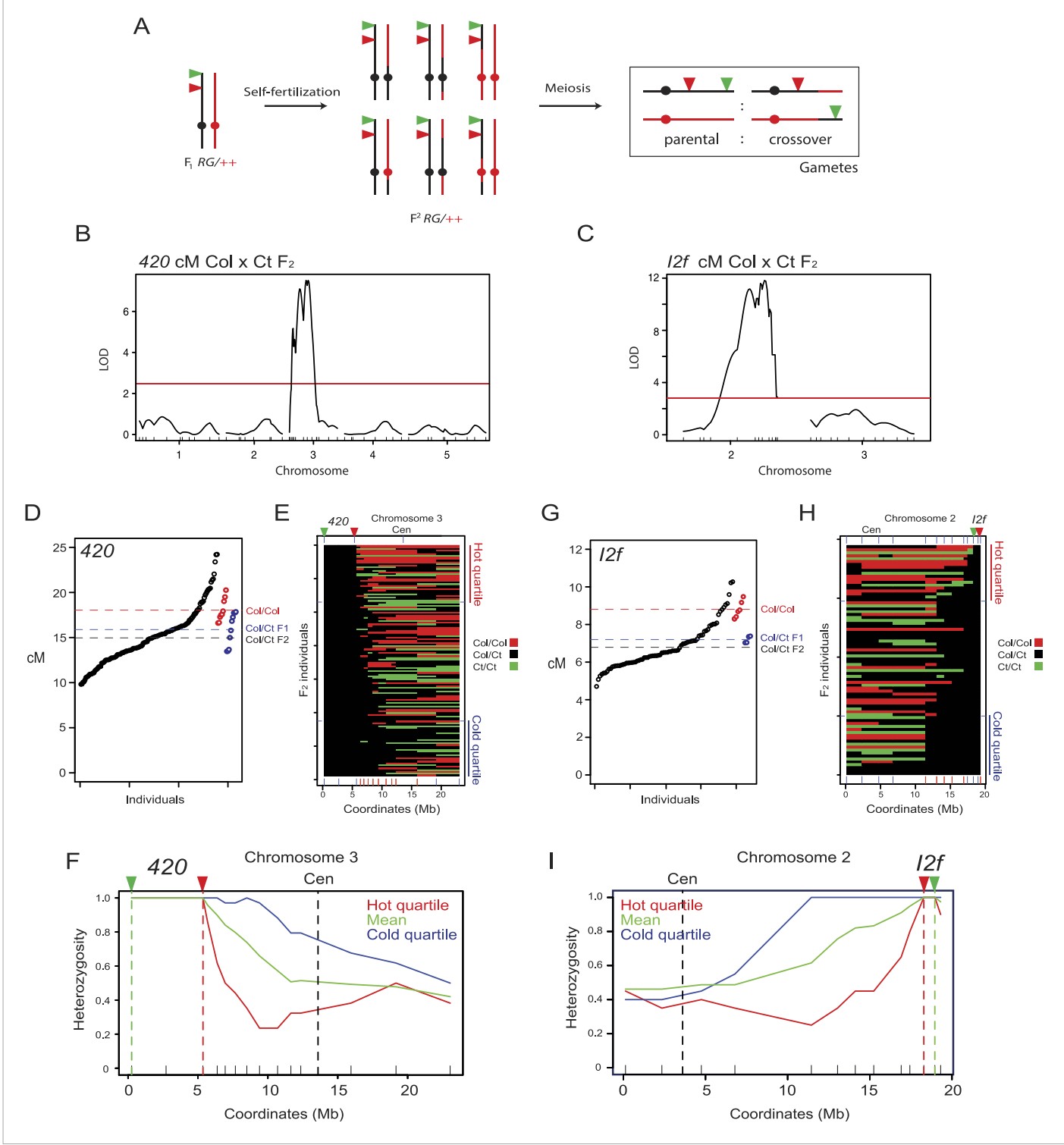

**Figure 4**. Modification of crossover frequency by juxtaposition of heterozygosity and homozygosity. (**A**) Diagram illustrating chromosome 3 genotypes (black = Col, red = Ct) in $RG/++$ $F_1$ individuals and their $F_2$ progeny. A single chromosome is shown for simplicity. Gametes or progeny are analysed for patterns of fluorescence following meiosis to measure genetic distance. (**B**) The program Rqtl was used to test for association between Col/Ct genotypes and $420$ cM in a $420/++$ $F_2$ population. The logarithm of odds (LOD) score is plotted along the 5 chromosomes with the positions of markers shown along the x-axis by ticks. The red horizontal line shows the 5% genome-wide significance threshold calculated with Hayley-Knott regression and by running 10,000 permutations. (**C**) As for (**B**) but analyzing Col/Ct markers on chromosomes 2 and 3 for an $I2f/++$ $F_2$ population. (**D**) $420$ cM measurements from Col/Ct $420/++$ $F_2$ (black), Col/Col homozygotes (red) and Col/Ct $F_1$ (blue) individuals. Mean values are indicated by horizontal dotted lines.

*Figure 4. continued on next page*

*Figure 4. Continued*

See *Figure 4—source data 1*. (**E**) Chromosome 3 genotypes shown for *420/++* F$_2$ individuals ranked by crossover frequency. Each horizontal row represents a single F$_2$ individual. X-axis ticks show marker positions, and which are coloured red when they showed significantly higher homozygosity in the hottest vs coldest quartiles (FDR-corrected chi square test). Fluorescent T-DNAs are indicated by triangles, in addition to the centromere (Cen). (**F**) Heterozygosity along chromosome 3 in the hottest (red), coldest (blue) *420* F$_2$ quartiles and the mean (green). The locations of reporter T-DNAs and the centromeres are indicated by vertical dashed lines. (**G–I**) As for (**D–F**) but for interval *I2f*. See *Figure 4—source data 2*.

The following source data and figure supplement are available for figure 4:

**Source data 1**. *420* Col/Ct F$_2$ fluorescent seed count data.

**Source data 2**. *I2f* Col/Ct F$_2$ fluorescent seed count data.

**Source data 3**. *CEN3* Col/Ct F$_2$ flow cytometry count data.

**Figure supplement 1**. Modification of crossover frequency by juxtaposition of heterozygosity and homozygosity.

(*Figure 6A*, *Figure 6—source data 2*). We next generated plants that were Col/Ct heterozygous (HET-HET) on chromosome 3 and observed that the previous increase in *420* crossovers was strongly suppressed in *fancm* and *fancm zip4* (GLM p = 1.24 × 10$^{-06}$ and p < 2.0 × 10$^{-16}$), whereas wild type Col/Ct were slightly but significantly higher than wild type Col/Col (GLM p = 0.0126) (*Figure 6A–B*). *CEN3* crossovers were also significantly suppressed by Col/Ct heterozygosity in *fancm* and nearly eliminated in *fancm zip4* compared to Col/Col (both GLM p < 2.0 × 10$^{-16}$) (*Figure 6A–B*). Together this indicates that the non-interfering crossover repair pathway that predominates in *fancm* and *fancm zip4* is less efficient in heterozygous regions and particularly within the centromeric region, which shows high polymorphism levels (*Table 2*).

We next tested the effect of juxtaposing heterozygous and homozygous regions in *fancm* and *fancm zip4* mutants. We first generated lines that were Col/Ct heterozygous within *420* and Col/Col homozygous outside (HET-HOM) (*Figure 6—figure supplement 1*). As expected, wild type HET-HOM lines show a significant increase in *420* and a reciprocal decrease in *CEN3* crossovers compared

**Table 5**. Chromosome 3 genotype counts from hot and cold quartile *420/++* Col/Ct F$_2$ individuals

| Marker coordinates (bp) | Hot quartile HET | Hot quartile HOM | Cold quartile HET | Cold quartile HOM | FDR p value |
|---|---|---|---|---|---|
| 259000 | 34 | 0 | 34 | 0 | 1 |
| 2718000 | 34 | 0 | 34 | 0 | 1 |
| 5352000 | 34 | 0 | 34 | 0 | 1 |
| 6375000 | 21 | 13 | 34 | 0 | 4.36 × 10$^{-04}$ |
| 6948000 | 17 | 17 | 33 | 1 | 1.05 × 10$^{-04}$ |
| 7674000 | 15 | 19 | 33 | 1 | 2.12 × 10$^{-05}$ |
| 8495000 | 12 | 22 | 34 | 0 | 3.65 × 10$^{-07}$ |
| 9404000 | 8 | 26 | 33 | 1 | 3.79 × 10$^{-08}$ |
| 10695000 | 8 | 26 | 30 | 4 | 1.36 × 10$^{-06}$ |
| 11649000 | 11 | 23 | 27 | 7 | 4.36 × 10$^{-04}$ |
| 12356000 | 11 | 23 | 27 | 7 | 4.36 × 10$^{-04}$ |
| 15949000 | 13 | 21 | 23 | 11 | 4.48 × 10$^{-02}$ |
| 19165000 | 17 | 17 | 21 | 13 | 0.591 |
| 23040000 | 13 | 21 | 17 | 17 | 0.591 |

The number of *420/++* Col/Ct F$_2$ individuals showing Col homozygosity (HOM) or Col/Ct heterozygosity (HET) for the indicated marker positions, in either the hottest or coldest F$_2$ quartile. The p value was obtained by performing a chi square test between homozygous and heterozygous marker genotype counts in the hottest and coldest quartiles (2x2 contingency table), followed by FDR correction for multiple testing.

**Table 6**. Chromosome 2 genotype counts from hot and cold quartile *I2f/++* Col/Ct F$_2$ individuals

| Marker coordinates (bp) | Hot quartile HET | Hot quartile HOM | Cold quartile HET | Cold quartile HOM | FDR p value |
|---|---|---|---|---|---|
| 132,000 | 9 | 11 | 8 | 12 | 1 |
| 2,346,000 | 7 | 13 | 8 | 12 | 1 |
| 4,748,000 | 8 | 12 | 9 | 11 | 1 |
| 6,789,000 | 7 | 13 | 11 | 9 | 0.63 |
| 11,443,000 | 5 | 15 | 20 | 0 | $6.26 \times 10^{-05}$ |
| 13,036,000 | 7 | 13 | 20 | 0 | $3.32 \times 10^{-04}$ |
| 14,117,000 | 9 | 11 | 20 | 0 | $1.30 \times 10^{-03}$ |
| 15,240,000 | 9 | 11 | 20 | 0 | $1.30 \times 10^{-03}$ |
| 16,909,000 | 13 | 7 | 20 | 0 | 0.0262 |
| 17,439,000 | 16 | 4 | 20 | 0 | 0.238 |
| 18,287,000 | 20 | 0 | 20 | 0 | 1 |
| 18,960,000 | 20 | 0 | 20 | 0 | 1 |
| 19,311,000 | 18 | 2 | 20 | 0 | 0.764 |

The number of *I2f/++* Col/Ct F$_2$ individuals showing Col homozygosity (HOM) or Col/Ct heterozygosity (HET) for the indicated markers, in either the hottest or coldest F$_2$ quartile. The p value was obtained by performing a chi square test between homozygous and heterozygous marker genotype counts in the hottest and coldest quartiles (2 × 2 contingency table), followed by FDR correction for multiple testing.

to wild type HOM-HOM (both GLM p < 2.0 × 10$^{-16}$) (*Figure 6A,C*), indicating compensatory changes between the two intervals in the HET-HOM lines. As the HET-HOM lines are heterozygous within *420*, this again inhibited crossovers in *fancm* compared to *fancm* HOM-HOM (GLM p = 2.38 × 10$^{-15}$) (*Figure 6A,C*). HET-HOM lines in *fancm zip4* showed lower *420* crossovers than wild type HOM-HOM (GLM p < 2.0 × 10$^{-16}$), which demonstrates that the interfering pathway is required for the heterozygosity-homozygosity juxtaposition effect (*Figure 6A,C*). We also generated HOM-HET lines that were homozygous within *420* and heterozygous outside, which significantly reduced *420* crossovers compared to wild type HOM-HOM as expected (GLM p = 7.60 × 10$^{-11}$) (*Figure 6A,D*). HOM-HET lines in *fancm* and *fancm zip4* showed high *420* crossovers comparable to HOM-HOM, as the non-interfering crossover repair active in these backgrounds is efficient in homozygous regions (*Figure 6A,D*). *CEN3* genetic distance was again strongly suppressed in *fancm* and *fancm zip4* HOM-HET lines compared with HOM-HOM (both GLM p < 2.0 × 10$^{-16}$), consistent with heterozygosity inhibiting non-interfering crossover repair (*Figure 6A,D*). Together these data demonstrate that juxtaposition of heterozygous and homozygous regions causes reciprocal changes in crossover frequency via interference.

## Total chiasmata are maintained when heterozygosity is varied

As we observed regional changes in crossover frequency with varying patterns of heterozygosity, we next sought to test whether total recombination events were different. When homologous chromosomes align on the metaphase-I plate, crossovers can be cytologically visualized as chiasmata (*Sanchez-Moran et al., 2002*). To estimate the number of crossovers per meiotic nucleus we performed chromosome spreads of pollen mother cells (PMCs), followed by fluorescence in situ hybridization using a *45S* rDNA probe (*Figure 7*, *Figure 7—source data 1*). We counted total chiasmata in metaphase-I nuclei in Col/Col homozygotes, Ct/Ct homozygotes and Col/Ct F$_1$ heterozygotes. In addition, we counted chiasmata in recombinant *420-CEN3* lines showing high (HET-HOM, 27.96 cM) and low (HOM-HET, 13.83 cM) *420* crossover frequency (*Figure 7C,D*). Adjacent chiasmata count categories were combined to give a minimum expected value of five for the purposes of a chi-square test with 8° of freedom. This test gave no significant differences in chiasmata between the genotypes (p = 0.3365) (*Figure 7*). Together this is consistent with homeostatic maintenance of crossover numbers, despite local crossover changes caused by juxtaposition of heterozygous and homozygous regions.

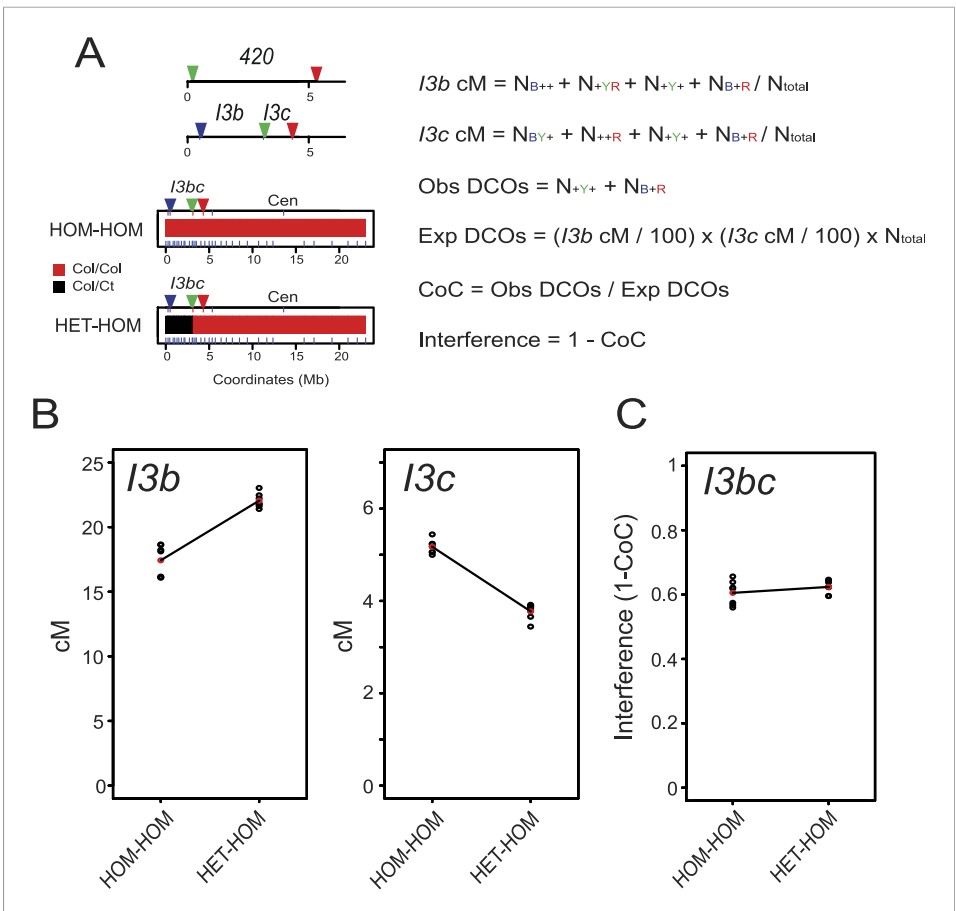

**Figure 5**. Juxtaposition of heterozygous and homozygous regions triggers reciprocal crossover remodelling. (**A**) Schematic diagram illustrating the physical location of *420* and *I3bc* transgenes expressing fluorescent proteins in seed and pollen. Beneath are diagrams illustrating the locations of Col/Col homozygous (red) and Col/Ct heterozygous (black) regions along chromosome 3. Positions of Col/Ct genotyping markers are indicated by blue ticks along the axis of the chromosome. Printed alongside are formulae for the calculation of genetic distance (cM) and crossover interference using *I3bc*. Counts of pollen with different combinations of fluorescence are indicated. For example, $N_{BYR}$ indicates the number of pollen with blue, yellow and red fluorescence. (**B**) *I3b* and *I3c* genetic distance (cM) measured in HOM-HOM and HET-HOM plants as illustrated in (**A**). See *Figure 5—source data 1*. (**C**) As for (**B**) but showing calculation of crossover interference (1-CoC). See *Figure 5—source data 2*.

The following source data and figure supplement are available for figure 5:

**Source data 1**. Three colour *I3bc* FTL flow cytometry count data.

**Source data 2**. Three colour *I3bc* FTL flow cytometry count data–measurement of crossover interference.

**Figure supplement 1**. Analysis of *I3bc* recombination using three-colour flow cytometry.

## Crossover interference increases in heterozygous regions

Our analysis of *420-CEN3* recombination rates implicated interference as driving crossover changes across homozygosity/heterozygosity junctions. We therefore sought to directly measure interference in lines with varying heterozygosity. We generated *I3bc* lines that varied in Col/Ct genotype and that were wild type, *fancm*, *zip4* or *fancm zip4* (*Figure 8—figure supplement 1*). We first compared *I3bc* plants that were Col/Col homozygous (HOM-HOM) with Col/Ct heterozygotes (HET-HET). In wild type, genetic distances did not significantly change between HOM-HOM and HET-HET (GLM p = 0.352 and p = 0.666), but crossover interference significantly increased (GLM p < 2.0 × 10⁻¹⁶)

**Table 7**. Chromosome 3 genotype counts from hot and cold quartile *CEN3*/++ Col/Ct F₂ individuals

| Marker coordinates (bp) | Hot quartile HET | Hot quartile HOM | Cold quartile HET | Cold quartile HOM | FDR *P* |
|---|---|---|---|---|---|
| 259000 | 16 | 14 | 17 | 13 | 1 |
| 2718000 | 16 | 14 | 18 | 12 | 1 |
| 5352000 | 19 | 11 | 17 | 13 | 1 |
| 7674000 | 20 | 10 | 12 | 18 | 0.129 |
| 8495000 | 23 | 7 | 13 | 17 | 0.0389 |
| 9404000 | 26 | 4 | 16 | 14 | 0.0308 |
| 11115724 | 30 | 0 | 30 | 0 | 1 |
| 16520560 | 30 | 0 | 30 | 0 | 1 |
| 21008000 | 27 | 3 | 14 | 16 | 0.00477 |
| 22076000 | 23 | 7 | 12 | 18 | 0.0308 |
| 23040000 | 24 | 6 | 10 | 20 | 0.00477 |

The number of *CEN3*/++ Col/Ct F₂ individuals showing Col homozygosity (HOM) or Col/Ct heterozygosity (HET) for the indicated markers, in either the hottest or coldest quartile. The p value was obtained by performing a chi square test between homozygous and heterozygous marker genotype counts in the hottest and coldest quartiles (2 × 2 contingency table), followed by FDR correction for multiple testing.

(*Figure 8A,B*, *Figure 8—source data 1*). Consistent with previous observations, *fancm* and *fancm zip4* showed a significant reduction and an absence of interference respectively, in a HOM-HOM background (GLM p < 2.0 × 10⁻¹⁶ and p = 4.94 × 10⁻¹⁶) (*Figure 8A*, *Figure 8—source data 2*) (*Crismani et al., 2012*; *Yelina et al., 2013*). In HET-HET plants the crossover frequency increases seen in *fancm* and *fancm zip4* were again greatly suppressed, or eliminated, relative to HOM-HOM, as observed previously for *420-CEN3* (GLM both p < 2.0 × 10⁻¹⁶) (*Figure 8B*). Unexpectedly, interference measurements significantly increased in both *fancm* and *fancm zip4* mutants in a HET-HET background compared to HOM-HOM (GLM p < 2.0 × 10⁻¹⁶ and p = 4.94 × 10⁻¹⁶) (*Figure 8B*). We propose that in the absence of the ZMM pathway alternative repair pathways exist which are differentially sensitive to polymorphism and interference. Multiple, redundant repair pathways are consistent with the residual crossovers observed in *msh4 mus81* double mutants (*Higgins et al., 2008b*). Finally, we measured *I3bc* cM in *zip4* mutants alone (HOM-HOM) and observed significantly decreased crossovers compared with wild type HOM-HOM (GLM p < 2.0 × 10⁻¹⁶) (*Figure 8E*, *Figure 8—source data 1*). Importantly, *zip4* genetic distances were further significantly reduced when comparing HOM-HOM to HET-HET backgrounds (GLM p = 1.79 × 10⁻¹⁰ and p = 1.53 × 10⁻⁹) (*Figure 8E*). This provides additional evidence that the non-interfering repair pathway remaining in *zip4* is inefficient in heterozygous regions. Interference measurements using *I3bc* are reliant on the relatively rare double crossover classes (N₋ᵧ₋ + N_B-R) (*Figure 5A*). Due to low *zip4* fertility it was difficult to obtain sufficient DCO counts to make reliable interference measurements, although the observed counts are consistent with an absence of interference in this mutant (*Figure 8—source data 4*).

To test the effects of heterozygosity/homozygosity juxtaposition we next generated lines that were Col/Ct heterozygous within *I3bc* and Col/Col homozygous outside (HET-HOM). As expected, wild type *I3b* and *I3c* genetic distances both significantly increase in HET-HOM lines relative to HOM-HOM (GLM both p < 2.0 × 10⁻¹⁶), consistent with our previous *420* experiments, and this was associated with a significant increase in crossover interference (GLM p < 2.0 × 10⁻¹⁶) (*Figure 8A,C*). As shown earlier, we observed that Col/Ct (HET-HOM) heterozygosity suppressed the crossover increases seen in *fancm* and *fancm zip4* (GLM p < 2.0 × 10⁻¹⁶), with the same significant increases in crossover interference strength (GLM p < 2.0 × 10⁻¹⁶ and p = 4.94 × 10⁻¹⁶) (*Figure 8A,C*). The reciprocal situation was observed in HOM-HET plants where *I3bc* was Col/Col homozygous and the rest of the chromosome Col/Ct heterozygous. *I3b* and *I3c* genetic distances were significantly decreased in wild type HOM-HET compared with wild type HOM-HOM plants (GLM both p < 2.0 × 10⁻¹⁶) (*Figure 8A,D*). HOM-HET *fancm* and *fancm zip4* plants showed high crossovers, as the non-interfering pathway is efficient in the homozygous *I3bc* interval (*Figure 8A,D*). We also generated HET-HOM *zip4* lines, which

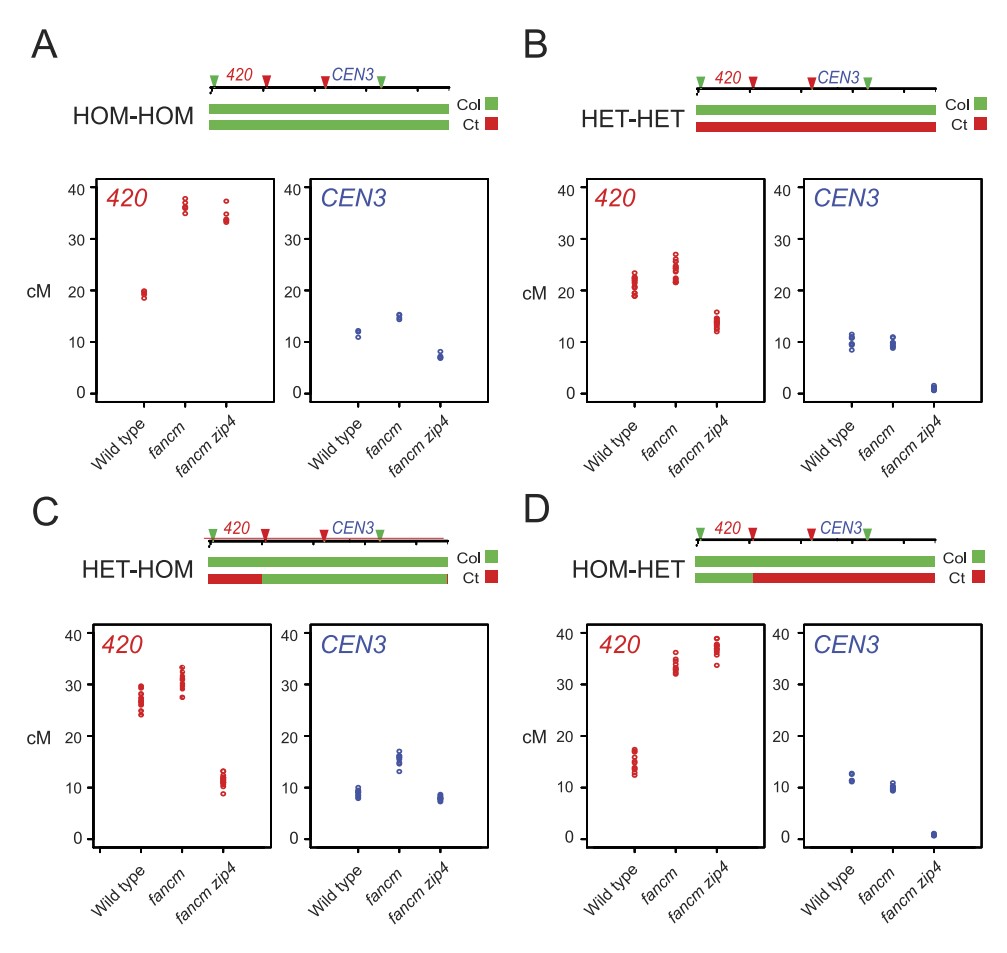

**Figure 6**. Genetic requirements of crossover remodelling via juxtaposition of heterozygous and homozygous regions. (**A–D**) Replicate measurements of *420* (red) and *CEN3* (blue) genetic distances (cM) are plotted in wild type, *fancm* and *fancm zip4*. See *Figure 6—source data 1, 2*. Chromosome 3 genotypes of the plants analysed are indicated above the plots (green = Col and red = Ct), for example, HET-HOM indicates heterozygous within *420* and homozygous outside.

The following source data and figure supplement are available for figure 6:

**Source data 1**. *420* fluorescent seed count data from wild type, *fancm* and *fancm zip4* individuals with varying heterozygosity.

**Source data 2**. *CEN3* flow cytometry count data from wild type, *fancm* and *fancm zip4* individuals with varying heterozygosity.

**Figure supplement 1**. Generation of wild type, *fancm* or *fancm zip4 420-CEN3* individuals with varying patterns of Col/Ct heterozygosity.

unlike wild type showed significantly lower *I3b* and *I3c* cM than HOM-HOM *zip4* (GLM both *P*= p < 2.0 × $10^{-16}$) (*Figure 8E*). This again demonstrates that crossover remodelling at heterozygosity/homozygosity junctions requires interference and that non-interfering repair is inefficient in heterozygous regions.

As an independent test of the effect of heterozygosity on crossover interference we analysed four three-colour FTL intervals distributed throughout the genome (*Figure 1A* and *Table 2*). We measured crossover frequency and interference in Col/Col homozygotes vs Col/Ler $F_1$ heterozygotes using meiotic pollen tetrads (*Tables 8, 9*). This approach is possible as the FTL crossover reporter system was generated in the *qrt1-2* mutant background, where sister pollen grains remain physically attached

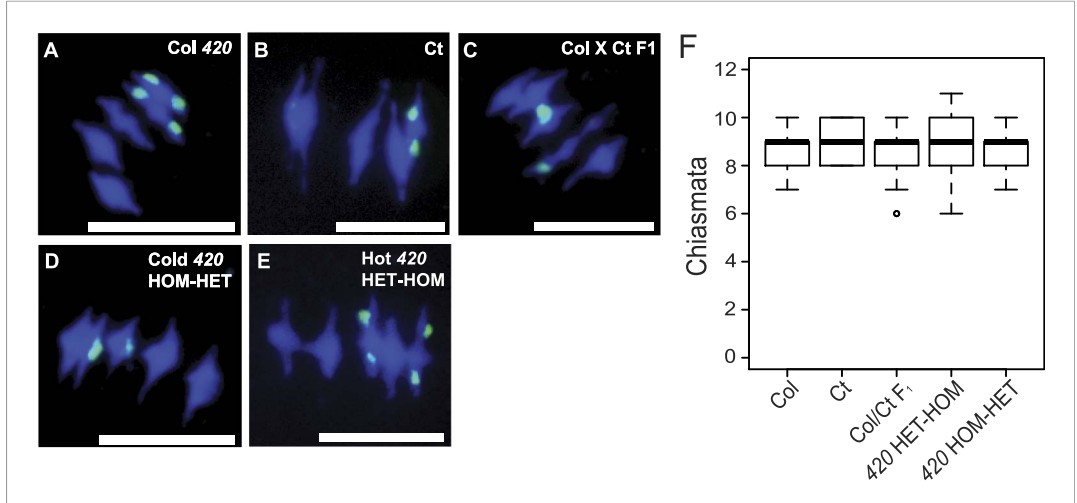

**Figure 7**. Total chiasmata frequencies are stable between Col, Ct and recombinant lines. (**A**–**E**) Metaphase-I chromosome spreads from anthers from (**A**) Col/Col *420*, (**B**) Ct/Ct, (**C**) Col × Ct F₁, (**D**) a Col × Ct *420* (HOM-HET) cold recombinant line and (**E**) a Col × Ct *420* hot (HET-HOM) recombinant line. DNA is stained with DAPI (blue) and labelled with a *45S* rDNA probe (green). Scale bars = 10 µM. (**F**) Boxplot showing total number of chiasmata per nucleus for each genotype. See **Figure 7—source data 1**.

The following source data is available for figure 7:

**Source data 1**. Chiasmata count data.

as meiotic tetrads (*Berchowitz and Copenhaver, 2008*). We scored a total of 49,801 tetrads for Col/Col (an average of 6225 per interval) and 42,422 tetrads for Col/Ler (an average of 5302 per interval) (*Tables 8, 9*). Compared to Col/Col, genetic distance significantly decreased in Col/Ler for six of the eight intervals measured and the remaining two intervals were not significantly changed (*Table 8*). To calculate interference strength we compared cM values in each interval from tetrads that had a crossover in the adjacent interval, to the same intervals in tetrads lacking a crossover in the adjacent interval, and detected significant positive interference in all cases (*Table 9*) (*Berchowitz and Copenhaver, 2008*). The resulting interference ratios were then compared between Col/Col and Col/Ler using Fisher's combined probability test, which revealed a significant increase in interference strength in Col/Ler ($\chi^2_{.001[16]} = 39.26$) (*Table 9*). Therefore, the effect of heterozygosity increasing the interference strength is evident in both Col × Ct and Col × Ler crosses.

## Discussion

We demonstrate reciprocal crossover increases and decreases when heterozygous and homozygous regions are juxtaposed and further demonstrate that this process requires crossover interference. The mechanism of interference is presently unclear, but a Beam-Film model has been developed where crossovers are patterned via forces similar to mechanical stress and which predicts experimental data (*Kleckner et al., 2004*; *Zhang et al., 2014a*, *2014b*). In this model each chromosome begins with an array of precursor interhomolog strand invasion events, one of which becomes crossover designated via a stress-related force (Designation Driving Force DDF). This causes a local reduction and redistribution of stress in both directions that dissipates with increasing distance (*Kleckner et al., 2004*; *Zhang et al., 2014a*, *2014b*). At the point where stress increases sufficiently precursor events can again become crossover designated. Any remaining precursors then mature into other fates including non-crossovers and non-interfering crossovers (*Kleckner et al., 2004*; *Zhang et al., 2014a*, *2014b*).

We considered the effect of juxtaposition of heterozygous/homozygous regions in the context of the Beam-Film model (*Kleckner et al., 2004*; *Zhang et al., 2014a*, *2014b*). Detection of heterozygosity most likely occurs downstream of interhomolog strand invasion and the formation of base pair mismatches. Therefore, we assume that the initial distribution of meiotic DSBs is

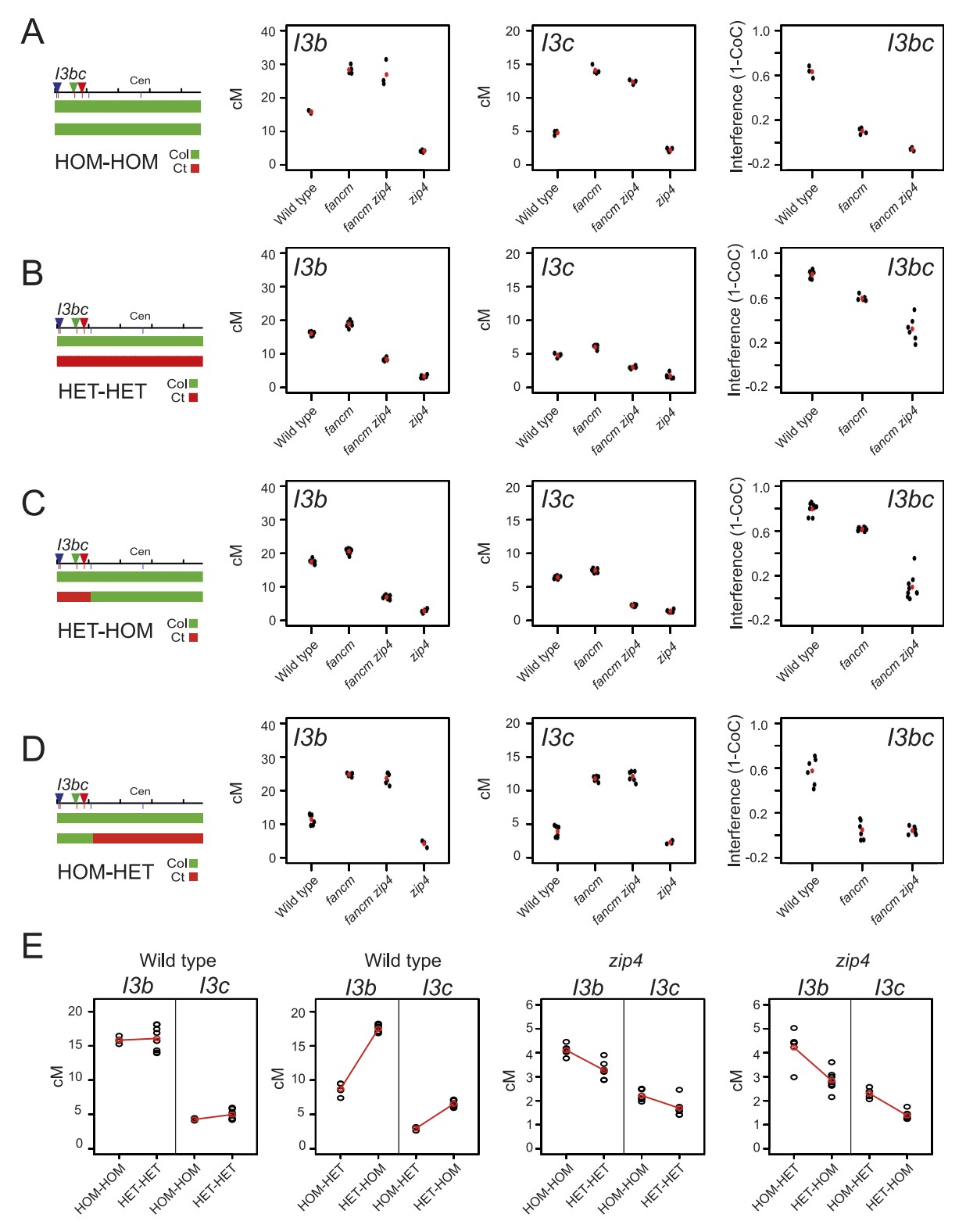

**Figure 8**. Crossover interference increases when heterozygous and homozygous regions are juxtaposed. (**A**–**D**) Replicate measurements of *I3b* and *I3c* genetic distances (cM), and *I3bc* crossover interference are plotted in wild type, *fancm*, *fancm zip4* and *zip4*. Black dots represent replicate measurements with mean values indicated by red dots. Chromosome 3 genotypes of the plants analysed are indicated above the plots (green = Col and red = Ct), for example, HET-HOM indicates heterozygous within *I3bc* and homozygous outside. See *Figure 8—source data 1, 2*. (**E**) *I3b* and *I3c* genetic distances (cM)

*Figure 8. Continued*

are plotted in wild type and *zip4* mutants with varying patterns of heterozygosity, labelled as for (**A**–**D**). Mean values between samples are connected with red lines. See *Figure 8—source data 3, 4*.

The following source data and figure supplement are available for figure 8:

**Source data 1**. *I3bc* fluorescent seed count data from wild type, *fancm* and *fancm zip4* individuals with varying heterozygosity.

**Source data 2**. Calculation of *I3bc* interference from wild type, *fancm* and *fancm zip4* individuals with varying heterozygosity.

**Source data 3**. *I3bc* fluorescent seed count data from wild type and *zip4* individuals with varying heterozygosity.

**Source data 4**. Calculation of *I3bc* interference in wild type and *zip4*.

**Figure supplement 1**. Generation of wild type, *fancm*, *zip4* or *fancm zip4 I3bc/++* plants with varying patterns of Col/Ct heterozygosity.

unchanged in homozygous or heterozygous states. Mismatches are observed to have a local inhibitory effect on meiotic crossovers (*Dooner, 1986*; *Borts and Haber, 1987*; *Jeffreys and Neumann, 2005*; *Baudat and de Massy, 2007*; *Cole et al., 2010*). Therefore, one possibility is that mismatched precursors in heterozygous regions are slowed in maturation and trigger feedback mechanisms that cause further DSBs, for example via ATM/ATR kinase signalling (*Carballo et al., 2008*; *Lange et al., 2011*; *Zhang et al., 2011*; *Kurzbauer et al., 2012*; *Garcia et al., 2015*). As a consequence, heterozygous regions would receive more 'late' DSBs, leading to more precursors and a higher chance of receiving a crossover designation event. An increased chance of crossover designation would lead to spreading of interference into adjacent homozygous regions causing reciprocal crossover decreases. An alternative model is that mismatched precursors are more sensitive to crossover designation and thus heterozygous regions have a higher chance of an interfering crossover, leading to similar effects. These potential models could be distinguished by measurement of non-crossover (NCO) levels, which should increase in heterozygous regions if more DSBs occur. Our data also demonstrate that non-interfering repair is less efficient in heterozygous regions, which will further contribute to the changes we see across homozygosity/heterozygosity junctions.

Sequence polymorphism has been observed to suppress crossover recombination at the hotspot (kilobase) scale in diverse eukaryotes (*Dooner, 1986*; *Borts and Haber, 1987*; *Jeffreys and Neumann, 2005*; *Baudat and de Massy, 2007*; *Cole et al., 2010*). For example, at the mouse *A3* hotspot an indel polymorphism within an inverted repeat overlaps a crossover refractory zone

**Table 8**. Tetrad FTL cM data in Col/Col and Col/Ler backgrounds

| Interval | Col/Col | | | | Col/Ler | | | |
|---|---|---|---|---|---|---|---|---|
| | PD | NPD | T | cM* | PD | NPD | T | cM* |
| 1b | 3976 | 3 | 742 | 8.05 ± 0.29 | 4395 | 2 | 652 | 6.58 ± 0.25† |
| 1c | 3022 | 11 | 1695 | 18.62 ± 0.04 | 3156 | 18 | 1891 | 19.73 ± 0.04 |
| 2a | 6787 | 2 | 430 | 3.06 ± 0.15 | 5920 | 0 | 283 | 2.28 ± 0.13† |
| 2b | 6582 | 2 | 635 | 4.48 ± 0.18 | 5796 | 0 | 407 | 3.28 ± 0.16† |
| 3b | 4363 | 22 | 2557 | 19.37 ± 0.35 | 2758 | 2 | 1056 | 13.99 ± 0.38† |
| 3c | 6185 | 5 | 736 | 5.53 ± 0.21 | 3576 | 2 | 238 | 3.28 ± 0.22† |
| 5c | 5356 | 1 | 666 | 5.58 ± 0.21 | 5458 | 0 | 676 | 5.51 ± 0.20 |
| 5d | 5358 | 1 | 664 | 5.56 ± 0.21 | 5540 | 2 | 594 | 4.94 ± 0.20† |

*Map distance in cM (±S.E.).
†Significant difference in map distance in the heterozygous Col/Ler background compared to the same interval in the Col/Col homozygous background.

**Table 9.** Tetrad FTL crossover interference data in Col/Col and Col/Ler backgrounds

| Interval | Col/Col | | | Col/Ler | | |
|---|---|---|---|---|---|---|
| | W/o adj. CO* | w/ adj. CO* | R1† | W/o adj. CO* | w/ adj. CO* | R2† |
| 1b | 10.69 ± 0.40 | 3.31 ± 0.30‡ | 3.23 | 9.78 ± 0.37 | 1.22 ± 0.18‡ | 8.04§ |
| 1c | 20.61 ± 0.45 | 7.92 ± 0.76‡ | 2.6 | 22.13 ± 0.46 | 3.52 ± 0.50‡ | 6.29§ |
| 2a | 3.20 ± 0.16 | 1.18 ± 0.30‡ | 2.75 | 2.42 ± 0.14 | 0.37 ± 0.21‡ | 6.55 |
| 2b | 4.65 ± 0.19 | 1.74 ± 0.44‡ | 2.68 | 3.41 ± 0.16 | 0.53 ± 0.30‡ | 6.44 |
| 3b | 20.84 ± 0.37 | 6.95 ± 0.82‡ | 2.3 | 14.73 ± 0.40 | 2.92 ± 0.76‡ | 5.05 |
| 3c | 7.65 ± 0.30 | 1.90 ± 22‡ | 4.03 | 4.28 ± 0.30 | 0.66 ± 0.18‡ | 6.46 |
| 5c | 5.87 ± 0.23 | 3.23 ± 0.47‡ | 1.82 | 5.85 ± 0.22 | 2.35 ± 0.43‡ | 2.49 |
| 5d | 5.85 ± 0.23 | 3.22 ± 0.48‡ | 1.82 | 5.29 ± 0.22 | 2.07 ± 0.38‡ | 2.56 |

*Map distances in cM (±S.E.) for intervals with and without adjacent crossovers (CO).
†Ratios of map distances for intervals with and without adjacent crossovers in homozygous Col/Col (R1) and heterozygous Col/Ler (R2) backgrounds.
‡Significant difference in map distances in intervals when adjacent interval does or doesn't have a CO.
§Significant difference between R2 and R1.

(*Cole et al., 2010*). However, this zone forms significant numbers of non-crossovers, indicating that the repeat/indel does not inhibit DSB formation, but inhibits downstream progression to crossover recombination (*Cole et al., 2010*). In yeast addition of SNPs to the *MAT-URA3* hotspot decreased crossovers and increased the frequency of gene conversions, further indicating that polymorphism can inhibit crossovers at fine-scale (*Borts and Haber, 1987*). Finally, intragenic mapping of the maize Bronze hotspot demonstrated that transposon insertions suppress crossovers more strongly than single nucleotide changes (*Dooner, 1986*; *Fu et al., 2001*; *Dooner and He, 2008*), again consistent with progression to crossover repair being inhibited by local sequence polymorphisms. Several heteroduplex joint molecules with distinct properties form during meiosis, including displacement-loops and dHJs (*Keeney and Neale, 2006*). It is possible that these joint molecules and their interactions with recombinases are sensitive to base-pair mismatches. The mismatch repair protein MutS directly recognizes mismatched base-pairs and serves as a paradigm for this type of function (*Lamers et al., 2000*; *Obmolova et al., 2000*).

The reciprocal crossover changes we observe when heterozygous regions are juxtaposed with homozygous regions are reminiscent of other homeostatic effects characterized during meiosis (*Hillers and Villeneuve, 2003*; *Martini et al., 2006*; *Robine et al., 2007*; *Libuda et al., 2013*; *Thacker et al., 2014*). For example, multiple levels of interference have been detected in mice (*de Boer et al., 2006*; *Cole et al., 2012*), Zip3 foci with distinct timing and properties are observed in budding yeast (*Serrentino et al., 2013*), and 'upstream' DSB patterns are altered in 'downstream' ZMM mutants (*Thacker et al., 2014*). As plants, fungi and mammals share the presence of interfering and non-interfering crossover repair pathways similar effects over heterozygosity/homozygosity junctions may be generally important (*Stahl et al., 2004*). However, when assessing the significance of such effects it is also important to consider how outcrossing vs selfing will influence patterns of homozygosity and heterozygosity within different species. Together our data show how varying patterns of sequence polymorphism along chromosomes can have a significant effect on distributions of meiotic recombination.

## Materials and methods

### Measuring crossovers using two-colour fluorescence microscopy of seed and flow cytometry of pollen

Flow cytometry of pollen can be used to rapidly measure meiotic segregation of heterozygous transgenes encoding distinct colours of fluorescent protein (*Yelina et al., 2012*, *2013*). cM were calculated from flow cytometry data using the formula:

$$cM = 100 \times (R5/(R3 + R5)),$$

Where R5 is a number of green-alone fluorescent pollen grains and R3 is a number of green and red fluorescent pollen grains (*Yelina et al., 2012*, *2013*). We previously observed that the number of red-alone pollen exceeded that of green-alone pollen when lines heterozygous for both eYFP and dsRed (eYFPDsRed/++) were analysed (*Yelina et al., 2012*, *2013*). Using pulse-width/SSC (side scatter) analysis and back-gating we demonstrated that the excess counts come primarily from non-hydrated pollen (*Yelina et al., 2012*, *2013*). Therefore to avoid this artifact we multiply the green-alone counts by two to obtain the number of recombinant pollen.

To increase measurement throughput using fluorescent seed we adapted CellProfiler image analysis software (*Carpenter et al., 2006*) (*Figure 2*). This program identifies seed boundaries in micrographs and assigns a RFP and GFP fluorescence intensity to each seed object (*Figure 2A–B*). Three pictures of the seed are acquired at minimum magnification (×0.72) using a charge coupled device (CCD) camera; (i) brightfield, (ii) UV through a dsRed filter and (iii) UV through a GFP filter (*Figure 2A*). As seed are diploid, there are nine possible fluorescent genotypes when a *RFP-GFP/++* heterozygote is self-fertilized, in contrast to four possible states for haploid pollen (*Yelina et al., 2013*) (*Figure 2E*). Histograms of seed fluorescence can be used to classify fluorescent and non-fluorescent seed for each colour (*Figure 2C–D*). Although it is possible to distinguish seed with one vs two T-DNA copies, there is greater overlap between the groups (*Figure 2C–E*). Therefore, we use fluorescent vs non-fluorescent seed counts for crossover measurement. Using this method it is possible to score 2000–6000 meioses per self-fertilized individual. When plants have been self-fertilized, genetic distance is calculated using the formula:

$$cM = 100 \times \left( 1 - [1 - 2(N_G + N_R)/N_T]^{1/2} \right),$$

Where $N_G$ is a number of green-alone fluorescent seeds, $N_R$ is a number of red-alone fluorescent seed and $N_T$ is the total number of seeds counted. During generation of *420/++* F$_2$ populations we selected for individuals that are heterozygous for transgenes expressing red and green fluorescent proteins (*RFP-GFP/++*). The majority of these individuals receive a chromosome with linked *RFP* and *GFP* transgenes over a non-transgenic chromosome (*RFP-GFP/++*) (*Figure 2—figure supplement 1*). In a minority of cases F$_2$ plants receive recombined *RFP-+* and *+-GFP* chromosomes (*Figure 2—figure supplement 1*). In the progeny of these individuals the fluorescent seed classes representing parental and crossover genotypes are reversed (*Figure 2—figure supplement 1*). As R+/+G plants also have variable heterozygosity/homozygosity patterns within *420* depending on crossover positions we excluded these plants from further analysis.

To test whether recombinant and non-recombinant counts were significantly different between replicate groups we used a GLM. We assumed the count data is binomially distributed:

$$Y_i \sim B(n_i, p_i),$$

where $Y_i$ represents the recombinant counts, $n_i$ are the total counts, and we wish to model the proportions $Y_i/n_i$. Then:

$$E(Y_i/n_i) = p_i,$$

and

$$var(Y_i/n_i) = \frac{p_{i(1-p_i)}}{n_i}.$$

Thus, our variance function is:

$$V(\mu_i) = \mu_i(1 - \mu_i),$$

and our link function must map from $(0,1) \rightarrow (-\infty, \infty)$. We used a logistic link function which is:

$$g(\mu_i) = \text{logit}(\mu_i) = \log \frac{\mu_i}{1 - \mu_i} = \beta X + \varepsilon_i,$$

where $\varepsilon_i \sim N(0, \sigma^2)$. Both replicates and genotypes are treated as independent variables ($X$) in our model. We considered p values less than 0.05 as significant.

## Measuring crossovers and interference using three-colour flow cytometry of pollen

Measurements of interference within the *I3bc* interval were carried out as described previously with minor modifications (Yelina et al., 2013). Inflorescences were collected in polypropylene tubes and pollen was extracted by vigorous shaking in 30 ml of freshly prepared pollen sorting buffer (PSB: 10 mM CaCl$_2$, 1 mM KCl, 2 mM MES, 5% wt/vol sucrose, 0.01% Triton X-100, pH 6.5). The pollen suspension was filtered through a 70 µM cell strainer to a fresh 50 ml polypropylene tube and centrifuged at 450×*g* for 3 min. The supernatant was removed and the pollen pellet washed once with 20 ml of PSB without Triton. The pollen suspension was centrifuged at 450×*g* for 3 min and the supernatant discarded and the pollen pellet resuspended in 500 µl of PSB without Triton. A CyAn ADP Analyser (Beckman Coulter, California, USA) equipped with 405 nm and 488 nm lasers and 530/40 nm, 575/25 nm and 450/50 nm band-pass filters was used to analyse the samples. Polygons were used for gating pollen populations and for each sample eight pollen class counts were obtained (*Figure 5—figure supplement 1*). *I3b* and *I3c* genetic distances were calculated using the following formula:

$$N_{total} = (N_{-Y-} + N_{B-R} + N_{-YR} + N_{B--} + N_{BY-} + N_{--R} + N_{BYR} + N_{---})$$

$$I3b\ cM = (N_{-Y-} + N_{B-R} + N_{-YR} + N_{B--})/N_{total}$$

$$I3c\ cM = (N_{-Y-} + N_{B-R} + N_{BY-} + N_{--R})/N_{total},$$

where $N_{-Y-}$, $N_{B-R}$, $N_{-YR}$, $N_{B--}$, $N_{BY-}$, $N_{--R}$, $N_{BYR}$, and $N_{---}$ are pollen grain counts in each of the eight populations (*Figure 5—figure supplement 1*). For example, $N_{BYR}$ is the number of pollen that were blue, yellow and red fluorescent.

Crossover interference was calculated using the following formulas:

$$Observed\ DCOs = (N_{-Y-} + N_{B-R}),$$

$$Expected\ DCOs = (I3b\ cM/100) \times (I3c\ cM/100) \times N_{total},$$

$$Coefficient\ of\ Coincidence = Observed\ DCOs/Expected\ DCOs,$$

$$Interference = 1 - CoC.$$

At least three biological replicates, constituting 3–5 individual plants were analysed for each sample (Yelina et al., 2013). Statistical tests for genetic distances were performed as described above using a GLM. To test for significant differences in interference we compared observed and expected double crossovers using the same approach.

## Generation of *fancm* and *fancm zip4* Col/Ct mapping populations with varying heterozygosity

Col-0 *420* and Ct-1 lines were crossed to *fancm-1 zip4-2* double mutant lines in the Col-0 background (Crismani et al., 2012) (*Figure 6—figure supplement 1*). The resulting F$_1$ plants were crossed together and progeny identified that were *fancm zip4* heterozygous, and *420/++* Col/Ct heterozygous on chromosome 3. Chromosome 3 genotypes were tested in all cases using 13 Col/Ct indel markers (*Supplementary file 1*). These plants were self-fertilized and *420* homozygous individuals identified (all seed were red and green fluorescent) that were also Ct homozygous outside of *420* and that were *fancm zip4* heterozygous (*Figure 6—figure supplement 1* and *Figure 8—figure supplement 1*). These plants were then crossed to *CEN3* or *I3bc* in wild type, *fancm* and *fancm zip4* mutants to obtain scorable progeny with a HOM-HET genotype (*Figure 6—figure supplement 1*). The selfed progeny of *420/++* Col/Ct *fancm zip4* heterozygous plants were also selected for plants with no fluorescent T-DNAs and either chromosome 3 in a Ct homozygous state, or with Ct homozygosity within *420* and Col homozygosity outside (*Figure 6—figure supplement 1*). These plants were crossed with doubly marked *420-CEN3* or *I3bc* lines in either wild type, *fancm* or *fancm zip4* mutant backgrounds to obtain HET-HET and HET-HOM scorable plants respectively (*Figure 6—figure supplement 1* and *Figure 8—figure supplement 1*). Equivalent genetic crosses were performed during analysis of *I3bc* (*Figure 8—figure supplement 1*). At least three independent lines were generated and analysed for each combination, apart from HOM-HET *420-CEN3* where two were analysed.

To genotype *zip4-2* (Salk_068052) the following primers were used:

zip4-2-F 5′-TTGCTACCTTGGGCTCTCTC-3′
zip4-2-R 5′-ATTCTGTTCTCGCTTTCCAG-3′
LBb.3 5′-ATTTTGCCGATTTCGGAAC-3′

The resulting PCR products were ~680 bp for wild type (zip4-2-F + zip4-2-R) and ~340 bp for *zip4-2* mutant (zip4-2-F + Lbb1.3) (*Crismani et al., 2012*).

To genotype the *fancm* mutation we amplified using the following primers:
fancm1dCAPsF1 5′-ACAATATATGTTTCGTGCAGGTAAGACATTGGAAG-3′
fancm1dCAPsR1 5′-CACCAATAGATGTTGCGACAAT-3′

The resulting PCR product was digested with *Mbo*II, which yields a ~215 bp product for wild type and ~180 bp for *fancm* (*Crismani et al., 2012*).

## Chiasmata counting

Chiasmata counting was performed as previously described (*Sanchez-Moran et al., 2002*).

## Acknowledgements

Research was supported by a Royal Society University Research Fellowship and Gatsby Charitable Foundation Grant 2962 to IRH, and United States National Science Foundation grant MCB-1121563 to GPC. We thank Raphaël Mercier for providing *fancm* and *zip4* mutations and genotyping information and Avi Levy for the *420* line. PAZ was supported by Polish Mobility Plus Fellowship 605/MOB/2011/0. We thank the editor and reviewers for insightful comments.

## Additional information

### Funding

| Funder | Grant reference | Author |
|---|---|---|
| Royal Society | University Research Fellowship | Ian R Henderson |
| Gatsby Charitable Foundation | 2962 | Ian R Henderson |
| National Science Foundation (NSF) | MCB-1121563 | Gregory P Copenhaver |
| Ministry of Science and Higher Education, Republic of Poland | 605/MOB/2011/0 | Piotr A Ziolkowski |

The funders had no role in study design, data collection and interpretation, or the decision to submit the work for publication.

### Author contributions

PAZ, Designed and performed all experiments and analysed data, except Col x Ler tetrad and chiasmata counting, and participated in preparation of the manuscript; LEB, GPC, Designed, performed and analysed Col x Ler tetrad experiments, read and approved the final manuscript; CL, ES-M, CF, Performed the chiasmata experiments, read and approved the final manuscript; NEY, Participated in data acquisition and analysis of *CEN3* $F_1$ data, read and approved the final manuscript; IRH, Designed experiments, collected $F_1$ data, analysed data and wrote the manuscript; XZ, KAK, Performed statistical analyses, read and approved the final manuscript; KC, Contributed to data analysis, read and approved the final manuscript; LZ, VJ, Aided PAZ in data acquisition, read and approved the final manuscript

### Author ORCIDs

Gregory P Copenhaver, http://orcid.org/0000-0002-7962-3862

## Additional files

### Supplementary file

• Supplementary file 1. Oligonucleotides used to genotype Col-0/Ct-1 polymorphisms.

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
