## [Decision Letter]

[Editors’ note: this article was originally rejected after discussions between the reviewers, but the authors were invited to resubmit after an appeal against the decision.]

Your full submission, “Heterozygosity promotes interfering and inhibits non-interfering crossovers during *Arabidopsis* meiotic recombination”, has been evaluated by Detlef Weigel (Senior editor), a Reviewing editor, and two peer reviewers, and the decision was reached after discussions between the reviewers. We regret to inform you that your work will not be considered further for publication.

There were two types of concerns. First, the reviewers were not convinced that heterozygosity generally increases crossovers (independently of genetic background and physical location in the genome), and that crossovers were recruited from homozygous regions of the genome to adjacent heterozygous regions. Similarly, the reviewers found that the mutant analysis was not fully developed. Second, there was agreement that, while your findings are provocative, they are not appropriately framed vis à vis alternative explanations for the observed relationships between recombination rates and diversity along genomes.

*Reviewer #1*:

The authors argue that recombination may be recruited from homozygous regions to nearby heterozygous regions. Overall I found the results interesting but I have some significant concerns about the presentation and interpretation of the results.

One of my major concerns about the article is that the authors frame it as a possible explanation for the positive correlation between crossover frequency and diversity that is seen across a number of the species examined to date. However, there is relatively good support, in population genetics literature, for the idea that this pattern reflects the fact that linked selection (e.g. hitchhiking and background selection) have effects over relatively large physical regions in regions of low crossing over (see [32]). The authors do not seem to mention this dominant, and relatively well supported, group of hypotheses and instead seem to frame the positive correlation as an empirical observation in search of a mechanistic hypothesis (that they provide). Obviously these hypotheses are not mutually exclusive, and so this does not rule out the idea that the authors' mechanistic explanation could also play a role. The authors need to be much clearer about the background to these observations, if they are to contribute to the larger debate about determinants of levels of polymorphism.

Along these lines it is not totally clear to me whether the observations could possibly explain a bulk of the correlation between recombination and polymorphism in many species. Much of the broad variation in recombination rates, in many systems, is explained by proximity to centromeres and telomeres (presumably a mechanistic constraint). Thus a lot of the variation in broad-scale recombination rates is not tied to polymorphism, but rather to large-scale chromosomal architecture. Once again the authors' proposed mechanism could contribute to the strength of the correlation, but this argument does suggest that this contribution may be somewhat minor.

The authors state, in the Abstract, that “using recombinant populations we show that heterozygous regions attract crossovers from homozygous regions on the same chromosome at the megabase scale”, and in one of the subsection of the Results that “Heterozygosity recruits crossovers over homozygous regions”.

I apologize if I've missed something, but I do not think the authors’ results actually demonstrate this. To my reading of these statements it sounds like the authors are saying that the extra recombination events in heterozygous regions come at the expense of fewer recombination events in the homozygous regions. But I do not see any results supporting for the second part of this statement, i.e. measuring recombination rates in homozygous regions. The authors' statements are one possible explanation of the results, but another is that the homozygous regions promote recombination with no “cost” to themselves. The authors may have evidence in favor of their hypothesis, perhaps this is the argument being made in the subsection headed “Differential sensitivity of interfering and non-interfering crossover repair to polymorphism”, but I found it hard to follow.

One concern I had was that recombination may be increased over broad chromosomal regions due to homozygosity in a specific region, because the individual would have reduced heterozygosity for structural rearrangements. This increase in recombination would not be associated with either parental allele, but with homozygosity itself. Presumably the authors have thought this through, and perhaps these lines are known to have no structural variation of suitable size in these regions, but it is worth clarifying this point.

At the end of the subsection headed “Modification of crossover frequency by *Arabidopsis* natural variation”, the authors state: “therefore, in many cases heterozygosity promotes *Arabidopsis* recombination relative to homozygosity, which is inconsistent with a purely suppressive effect of polymorphism.” I don't think this is correct as stated. The authors have shown that when Col is crossed to other lines the F_1_s sometimes have higher recombination rates than Col homozygotes. However, this could be because the other lines crossed to harbor recombination modifiers that increase recombination rates. For the authors claim that heterozygosity promotes recombination to be true they would need to show that the F_1_s often have higher recombination rates than either of the parental lines.

*Reviewer #2*:

This paper presents evidence for three conclusions.

1) In one well-studied test interval, in a particular heterozygous state, the frequency of crossovers is higher when the region adjacent to the test interval is homozygous rather than heterozygous. This is a “*cis*” effect. This is shown in Figure 4 and in the non-mutant background of Figure 7 (compare panels A and C).

There are some limitations to this observation. First, it is shown for only one interval. Second, interval is sub-telomeric and thus likely not to be representative of most of the genome. Third, it is shown for only one pair of lines (Col vs. Ct).

It is also important to note that in this test interval, there is no difference in crossover level between the homozygous and full F_1_ hybrid strains (Figure 7 compare panels A and B in the non-mutant case). Thus, there is no general effect of heterozygosity to increase the number of crossovers. This is also seen in the overall evaluation of Col/Col vs Col/Ct F_1_'s. Thus, the identified phenomenon is some type of unusual cis interaction which may or may not be widely general.

The basis for this cis interaction proposed by the authors does not seem to make sense to me. This is in part because it seems to invoke result (2) below in an inappropriate way and in part because the statement is made that crossovers are “recruited” from homozygous regions. But this conclusion cannot be drawn. There was no analysis of the effects of a heterozygous region on a flanking homozygous region.

It seems that this conclusion of some type of “competition” between homozygous and heterozygous regions is drawn largely from Figure 6. But the results in this Figure do not support this conclusion. What this figure shows is that when the test region is homozygous and the flanking region is heterozygous, the level of crossovers is reduced in the test region as compared to the case where the test region is heterozygous and the flanking region is homozygous. This does not imply that there is a reciprocal effect of heterozygosity to reduce crossovers in the homozygous region. There is no control to show what the frequency of crossovers is in a fully homozygous case where there is no heterozygous region adjacent to the test region. There is also no control to show what happens if the Col/Col homozygous test region is flanked by the same homozygous region as when the test region is heterozygous (i.e. Ct/Ct).

Finally, the authors suggest that the cis phenomenon is general because some F_1_ hybrid strains show higher levels of recombination than one particular reference homozygous strain. However, this finding is just as easily explained by general genetic background effects on the recombination process rather than anything to do with heterozygosity at the DNA level, as the authors admit. Thus, this is not really supporting evidence.

2) The authors show that heterozygosity is accompanied by a change in the sensitivity of crossovers to *fancm* and *zip4* mutations. The analysis is not rigorously correct because there was no test of a *zip4* mutation by itself; the only test was in a *fancm zip4* background.

The most important point is that this is a general characteristic of heterozygosity: it is observe in a pure F_1_ hybrid. Thus there is no reason to link this phenomenon to the *cis* effect, as the authors seem to do.

A second important point is that the total frequency of crossovers in the F_1_ hybrid is the same as in the homozygous reference strain. Thus, there is no “recruitment of crossovers” as the authors also seem to suggest.

Finally, comparison implies a likely situation is that there is a change in the proportion of recombination events that are “interfering crossovers” and versus crossovers that arise as a minority population from the “non-crossover” pathway. To put it another way, in the wild type case, there is an increase in crossovers and a decrease in non-crossovers (more of which turn into crossovers in *fancm* than in wild type). The basis for this effect is unclear. However, it is strongly reminiscent of crossover homeostasis where a reduction in DSBs leads to a differential loss of non-crossovers as compared to crossovers. Since it is unlikely that heterozygosity will be recognized at the DSB level (although this is not totally excluded given diverse indications of *trans* effects on DSBs in yeast), it would instead imply that there is a reduced chance that a DSB actually giving a recombination intermediate that could lead to a crossover or non-crossover. This is entirely possible since heterozygosity could be sensed at the time of establishment of such an interaction or concomitant with crossover/non-crossover decision making.

3) The authors show that an F_1_ hybrid appears to have increased crossover interference as compared to the homozygous reference strain. There are two problems with this result and linking it to other results. First, this is a different hybrid from the one tested for all other phenomena. Second, there is no evidence that this effect is the result of heterozygosity at the DNA level rather than some *trans*-acting strain background effect. Third, in this hybrid the total crossover level is lower than in the reference homozygote; thus, this is not related to the finding that F_1_ hybrids can exhibit higher levels of recombination.

That being said: there is a possible way that this result could be relevant to the sub-telomeric effect described above. In *Arabidopsis*, as in several other organisms, interference distance goes with physical chromosome length. Thus, in the hybrid, increased interference per Mb (which is what is assayed) could result from a decrease in physical chromosome length. Furthermore, the *cis* effect described in the results in prior sections could be explained if the longer length characteristic of the homozygous region can spread into the adjacent sub-telomeric region, thus increasing physical length, decreasing interference per Mb and thus increasing crossovers per Mb.

Given the above considerations, the statements in the Summary need some amendment, as follows:

A) “We… found hybrids with higher recombination than homozygotes, demonstrating that polymorphism can promote crossovers”. If this is a reference to the finding of F_1_ hybrids with higher recombination, it is not accurate. There is no basis for the conclusion that the higher recombination is due to DNA polymorphism rather than *trans*-acting genetic effects. The *cis* effect is not evidence. Moreover, in the Col/Ct F_1_ hybrid used for most of the analysis, recombination frequencies are generally not higher than in Col/Col and in the Col/Ler hybrid used for interference analysis, recombination frequencies are lower than in the Col/Col homozygote.

B) “Using recombinant populations we show that heterozygous regions attract crossovers from homozygous regions on the same chromosome at a megabase scale”. This is definitively not shown by these data (see discussion in point 1 above).

C) “We demonstrate that this polymorphism *cis*-effect is dually mediated by promotion of interfering crossovers and inhibition of non-interfering crossovers in heterozygous regions”. This is not correct. As discussed in detail for point (2) above, the changes in interfering and non-interfering crossovers are not specific to the *cis* effect: they are a general feature of the heterozygous cases analyzed (even assuming no general genetic background issues). Furthermore, the observation does not imply two mechanistically distinct effects as the above statement suggests. Rather, there is a change in the distribution of undifferentiated recombination intermediates into crossover versus non-crossover outcomes.

D) “This reveals an unanticipated mechanism whereby DNA polymorphism can recruit crossovers, contribute to positive correlations between recombination and diversity and influence the action of selection.” For this reviewer, there is no positive correlation between recombination and diversity shown in this paper, as described in detail above. The F_1_ hybrid data are not evidence. Hybrids can have higher or lower levels than a particular reference homozygote. A F_1_ hybrid strain shows a higher proportion of interfering crossovers among total crossovers but no difference in total crossover levels. And the one case in which crossover levels are increased in a heterozygous region is not shown to involve “recruitment” of crossovers to the heterozygous region from a homozygous region. There is a *cis* effect in which presence of an adjacent homozygous region increases crossovers in the heterozygous test region, but this is an increase above the level seen in the heterozygous region in the pure F_1_ hybrid, so it does not represent a simple effect of DNA polymorphism. Finally, there is no mechanism revealed. There is the finding that F_1_ hybrids have altered crossover/non-crossover ratios, but this does not increase the overall level of crossovers, so the “mechanism” alluded to does not increase crossing-over.

E) The Discussion culminates with the following point:

“We propose that detection of sequence mismatches occurs during strand invasion or heteroduplex/dHJ formation and differentially inflluences the activity of interfering and non-interfering recombination proteins coincident with crossover/non-crossover repair choice. Therefore, as interfering and non-interfering repair pathways compete for DSBs, their activities are simultaneously modulated by heterozygosity, causing the *cis* effect.”

There are several problems, touched on above, which converge here. (i) There is not really a competition between interfering and non-interfering pathways. There is a crossover designation process and the leftovers become mostly non-crossovers but occasionally become crossovers, and the level of those latter crossovers are increased by *fancm* (as the authors of the *fancm* study state). (ii) Most importantly, the relevant effect does not cause the *cis* effect—it is seen in a pure F_1_ hybrid. The *cis* effect must come from something else. (iii) There is no reason to think that the crossover/non-crossover choice is made during dJH formation; all that can happen at this stage is that the process can be degraded to give fewer crossovers and more non-crossovers, which is the opposite of the effect observed here.

[Editors’ note: what now follows is the decision letter after the authors submitted for further consideration.]

Thank you for resubmitting your work entitled “Heterozygosity promotes interfering and inhibits non-interfering crossovers during *Arabidopsis* meiotic recombination” for further consideration at *eLife*. Your revised article has been favorably evaluated by Detlef Weigel (Senior editor) and three reviewers. The manuscript is improved, but there are still concerns whether your interpretation of the data goes too far. Therefore, we are asking you to rewrite the manuscript as much as possible to be a fair description of the unsuspected phenomena, without making too many claims regarding crossovers being attracted from one kind of region to another. In other words: please accommodate the reviewers' comments as much as possible. We realize that there are differences in interpretation of the data between you and specifically reviewer 2, but we felt that the phenomenon is important enough that it deserves prominent publication. One of our board members also stated that the “phenomenon is interesting, but the point is not to find a model (it does not add much to invoke the beam film model, since there is no way to know if it is late DSBs and/or mismatches, and the authors are certainly far from understanding the molecular mechanism) but to validate the general principle.”

Detlef Weigel has made specific comments in the manuscript, as attached.

*Reviewer #1*:

Over all I found the manuscript to be much improved in terms of it presentation of its results, and the addition of the new analyses made the findings a more general statement of the effect of heterozygosity on recombination patterns. I note, however, that the other reviewer's original concerns were much more substantive than mine. As such, I view their opinion as carrying more weight than mine in this appeal.

The last paragraph of the Discussion is problematic:

“We propose that the biological function of the heterozygosity *cis*-effect is to recruit crossovers to variable regions of the chromosomes, acting as a feed-forward mechanism to increase diversity.”

“… meiotic recombination has been selected to promote…”

Arguments about the evolution of recombination modulators are very slippery (as recombination unlinks the fate of the modifier and the recombinant haplotype it creates). As such these proposals are unsupported speculation, perhaps changing “propose” to “speculate” would help. *Arabidopsis thaliana* has not been a selfing lineage for very long (like most selfers), and the authors' argument does not seem super convincing evolutionary mechanism for outcrossers (as homozygosity runs will be broken up across generations, by segregation). So their explanation seems somewhat shaky.

In outbred organisms very long blocks of homozygosity are rare. The authors should caution that they know little about how long a block of homozygosity is needed to promote this effect, so the importance of this effect in other systems (e.g. mammals) is unclear.

In general the authors have done a better job of acknowledging other likely contributors to the recombination/heterozygosity relationships. Except for:

“However, the *cis*-effect is unlikely to explain all of this relationship and genetic hitchhiking, background selection and recombination associated mutagenesis may play important roles”.

The contribution of these other effects has been subject to quantitative investigation for over a decade. While the authors’ findings are very interesting, it is a disservice to use “all” and “may” in this way. I'd say that the contribution of linked selection is much more established than their mechanism, and should be acknowledged as such.

*Reviewer #2*:

Suggestion:

Title: Adjacent homozygous and heterozygous regions reciprocally enhance and suppress crossing-over in an interference-mediated process.

Summary: Analysis of meiosis in mosaically-hybrid *Arabidopsis* lines reveals that a heterozygous region suppresses crossing-over in an adjacent homozogous region while, reciprocally, the homozygous region increases crossing-over in the adjacent heterozygous region. This interplay requires crossover interference: it is absent in a *fancm zip4* background where crossovers occur but interference is absent. Two new features specific to recombination in heterozygous regions are also revealed: an effective increase in crossover interference and a decreased effect of a *fancm* mutation, which normally increases crossovers that do not exhibit interference. Potential mechanisms and evolutionary implications are discussed.

Part I. Summary. The authors have identified an interesting phenomenon that takes place at the junctions between homozygous and heterozygous regions, particularly when one of the involved regions is sub-telomeric: the frequency of recombination (crossing-over) in the homozygous region goes down while the frequency of recombination in the heterozygous region goes up (relative to the fully homozygous and fully heterozygous cases, respectively). This phenomenon implies reciprocal interplay between the two regions. This phenomenon applies specifically to junction regions. Fully heterozygous regions exhibit almost the same recombination frequency as fully homozygous regions, at least in the situation examined here. This could be of genetic/evolutionary significance, although that would depend, particularly since the role of crossing-over for evolution is hotly debated. It is not 100% clear whether the phenomenon applies generally throughout the genome and/or why it is particularly prominent in sub-telomeric regions.

The authors go on to suggest a specific mechanism for this phenomenon. One point is clear: this phenomenon requires crossover interference. In a mutant situation where there are crossovers, but no interference, the phenomenon is absent. This is interesting and sensible because crossover interference is, by its nature, a process in which adjacent regions communicate with one another.

Beyond this point, however, the authors’ assertions regarding mechanism are not supported by the data.

1) The authors say that non-interfering COs are suppressed in heterozygous regions. This features prominently in the Title and Summary. This is not shown in this paper. What is shown is that the COs which occur in a *fancm* background (which do not exhibit interference) are reduced in heterozygous regions. There is no evidence that these COs are occurring by the same molecular mechanism as the “canonical non-interfering COs” that arise in wild type meiosis.

2) The authors show that interference, as classically defined, appears to be stronger in heterozygous regions than in homozygous regions. This is also an interesting observation, which is documented not only for the specific situation analyzed in detail, but more broadly. This is the first time that interference has been examined in heterozygous situations, as far as I know.

However, further consideration of the implications of this finding have some problems: (a) the phenomenon of increased interference may not mean what the authors think it means; and (b) the authors wish to say that this increased interference is responsible for the interplay between heterozygous and homozygous regions. This is a possible model. But I think it is wrong and there is actually evidence in the paper against it.

The strongest direct argument against the authors’ model is that the reciprocal interplay between heterozygous and homozygous regions is observed in a situation where crossovers in the two regions exhibit the level of interference characteristic of the homozygous region (Figures 6 and 8; details below).

More generally: it is clear that, at junctions, crossovers go down in homozygous regions and go up in heterozygous regions, relative to the pure homozygous or pure heterozygous cases, respectively, and that interference is required (above). The question is: what is the basis for this asymmetry? In my opinion, the underlying effect could be that there are more DSBs (or total inter-homolog interactions) in heterozygous regions. More DSBs means more COs, which means more interference emanating from that region across the border to the homozygous region which means fewer COs in the homozygous region than in the pure homozygous case. Oppositely, fewer DSBs in homozygous regions means fewer COs which means less interference emanating from than region across the border to the heterozygous region which means fewer COs in the heterozygous region than would be observed in the pure heterozygous case.

This model is further supported by two interrelated considerations. (1) Fully heterozygous regions show the same level of COs as fully homozygous regions (Figures 7 and 8 wild type cases) even though there is more interference. Clearly there must be some other effect that counterbalances the apparent increase in interference. (2) The apparent increase in interference could, in principle, reflect either of two effects (Zhang et al., 2014 PLoS Genetics and PNAS): (a) an increase in the distance over which interference spreads; or (b) a decrease in the “strength” of CO-designation. The first would imply a more robust process in the heterozygous case, which seems peculiar a priori; the second would imply a less robust process in the heterozygous case, which makes more sense. (3) If phenomenologically increased CO interference reflects model (b), then the level of COs can be restored by increasing the number of DSBs.

By my alternative model, the phenomenological increase in interference in heterozygous regions is actually irrelevant (as shown by the data mentioned above). The real effect would be more DSBs.

Another problem with the authors' model is that it requires that the “increased interference in a heterozygous region” spreads across the boundary into the homozygous region and that the “decreased interference in a homozygous region” spreads across the boundary into the heterozygous region. This is a priori unlikely if the basis for the altered interference is heterozygosity per se.

It would actually not be so difficult to provide at least some evidence for more DSBs/interactions in heterozygous regions (e.g. by Dmc1 focus analysis). However, this opens up another can of worms and is beyond the scope of the current presentation.

The basic point is that it is not proven that there is a role for increased interference in heterozygous regions in the observed junction phenomenon and there seems to be evidence against it.

Importantly also: the documentation, description and presentation in this paper requires significant improvement as described below. There are problems of logic, vocabulary, data, controls, explanation and interpretation. This paper is not really readable by a general audience in its present form.

There is redundancy in the comments below, for which I apologize, but hopefully this is useful.

Part II. Specific Issues with Title and Summary.

The title says: “Heterozygosity promotes interfering and inhibits non-interfering crossovers during *Arabidopsis* meiotic recombination”. I do not think that either of these conclusions is warranted.

(i) The latter conclusion, that heterozygosity inhibits non-interfering crossovers, is wrong because it is based on the assumption that the extra COs that arise in a *fancm* mutant (and do not exhibit interference) are biochemically the same as the extra COs that arise in wild type (and are defined as “non-interfering COs”). Heterozygosity does decrease the COs seen in a *fancm* mutant (Figures 7 and 8), but this cannot be extrapolated to wild type.

(ii) Heterozygosity does, phenomenologically, increase CO interference. This is an interesting observation. But the interpretations placed on it are not proven and, in my opinion, are not correct. This finding could perhaps be interpreted as “promoting the interfering CO pathway” (as in the Title). However: (a) In comparisons between fully homozygous and fully heterozygous regions, the frequency of COs is essentially the same (Figures 7 and 8). This cannot be explained by an increase in CO interference alone. It implies some other effect. (b) The observation of increased CO interference by CoC analysis can be explained either by an increase in the “spreading distance” of the interference signal or by a decrease in the strength of CO-designation; both effects decrease the CoC at short inter-interval distances (See Zhang et al PLoS Genetics 2014 and PNAS 2014). If the relevant effect is to decrease the strength of CO-designation the observed effect is actually not promoting the interfering CO pathway but making it worse. (c) The fact that CO number does not change in a fully heterozygous region even though interference effectively increases (a, above) suggests/implies the existence of another effect, which could/should be upstream of any recombination fate decision. Most simply: if there is a decrease in the efficiency of CO-designation, the resultant decrease in COs can be overcome by an increase in the frequency of DSBs (or DSB-mediated pre-CO interactions). Perhaps mismatched chromatin structure has such an effect, maybe especially in sub-telomeric regions which might be influenced by pre-DSB homologous pairing interactions.

The end of the Summary says: “Using recombinant populations we show that heterozygous regions attract crossovers from homozygous regions on the same chromosome at the megabase scale. Interference inhibits formation of adjacent crossovers over similar physical scales, and we demonstrate that this polymorphism *cis*-effect is dually mediated by promotion of interfering crossovers and inhibition of non-interfering crossovers in heterozygous regions. This reveals an unanticipated mechanism whereby DNA polymorphism can recruit crossovers, contribute to positive correlations between recombination and diversity and influence the action of selection.”

I do not agree with any of these statements. Let us first define “polymorphism *cis* effect”. What was analyzed were constructs in which recombination was assayed in a sub-telomeric test region, which could be homozygous or heterozygous, and had an adjacent internal region that could be homozygous or heterozygous. Assuming experimental problems (below) can be ignored, what is observed (assayed region underlined) is, essentially, the following: (a) the frequency of COs is lower in HOM-HET than in HOM-HOM; and (b) the frequency of COs in HET-HOM is higher than in HET-HET. That is, an adjacent heterozygous region decreases COs in a homozygous region relative to what would have been seen in a fully homozygous situation while an adjacent homozygous region increases COs in a heterozygous region relative to what would have been seen in a fully heterozygous situation. *But*… HOM-HOM and HET-HET both show the same frequency of COs as one another (see Figures 6CD, 7 and 8 for the clearest examples). [There is one exception (Figure 8; see below) which I will simply ignore]. This is not: “heterozygous regions attract crossovers from homozygous regions on the same chromosome at the megabase scale”. There is no attraction; this is not a zero-sum game; the effect is reciprocal, not unidirectional; and stating the result in this way is not accurate and is prejudicial to thinking. “Recruitment” to heterozygous regions is simply wrong. So what is the explanation for the observed effects? The paper also shows (Figures 7 and 8) that this interplay is not observed in a *fancm zip4* double mutant where there are still COs, but of a variety that does not set up interference. Fine. Interference is required. And if everything were the same in HOM and HET regions, there would be no effect of having one next to the other, so just saying that there is interference is not enough. There has to be some asymmetry. The authors seem to think that the difference is that interference is “stronger” in HET regions and “weaker” in HOM regions. If this were true, and if that effect crossed the border between the two regions, COs in HET regions would tend to decrease COs in HOM regions relative to the effect from normal HOM interference; oppositely, HET regions will experience less interference from an adjacent HOM region than from an adjacent HET region. However: I do not think that this is correct. To reiterate some of the points made above:

(A) This model assumes that interference specific to HOM or HET can cross a HET/HOM border. This is a priori unlikely. Furthermore, there is data in the paper, which says that you can see the diagnostic reciprocal effects on CO levels in three-factor crosses where the HOM-type CO interference level is observed (Figures 6 and 8; see below).

(B) An alternative explanation is the same one required to explain interference patterns above (and thus is more likely): more DSBs in HET regions vs HOM regions. More DSBs in HET regions will imply more COs. More COs means more interference signals, which means fewer COs in the adjacent region. Oppositely, in HOM regions, fewer COs means fewer interference signals which implies more COs in the adjacent region.

To return to the Summary: “we demonstrate that this polymorphism *cis*-effect is dually mediated by promotion of interfering crossovers and inhibition of non-interfering crossovers in heterozygous regions. This reveals an unanticipated mechanism whereby DNA polymorphism can recruit crossovers, contribute to positive correlations between recombination and diversity and influence the action of selection.” Inhibition of non-interfering COs is not shown (above).

Enhanced interference in heterozygous regions (which is apparently what the authors mean by “promotion of interfering COs”) is not relevant. The likely relevant effect (more DSBs and thus more COs in HET regions) is not mentioned or discussed as a possibility.Thus: there is, therefore, no “unanticipated mechanism”, and no effect in which “DNA polymorphosms recruit COs and thus increasing genetic diversity in hybrid situations”.

Instead: what has been shown? There is a phenomenon that is observed when HOM and HET regions are side-by-side. A “junction” phenomenon, if you will. There is a reciprocal interplay in which COs are decreased in HOM regions and increased in HET regions and this interplay requires interference. This is likely correct, although there are a lot of technical issues pertaining to the data (below). NB that this is specific to adjacent HET/HOM regions, because HOM/HOM and HET/HET recombination are very similar. Thus, this is not a general effect of global heterozygosity, but an effect specific to juxtaposed HET/HOM regions (junctions). Thus any evolutionary implications must derive from this specific situation, not from heterozygosity in general.

The basis for this effect, about which the authors are quite specific in the title and the Summary, is probably not what the authors think it is, as described above. To repeat (again): There is no data on interplay between HET regions and “non-interfering COs” in wild type, only in a mutant situation that might be different. The observed increase in interference is real and per se interesting, but this does not imply a more robust interfering CO pathway; it could as easily and more probably imply a weaker pathway. And regardless of that point, the increase in interference in HET regions is not responsible for the “junction interplay”, as described above. This leaves the actual basis for the asymmetry between the two types of regions to be determined, but increased DSBs is an attractive possibility, which is not considered (described above).

It is also notable that most of the data come from analysis of a sub-telomeric interval(s) which could be special for any number of reasons. This maybe ok, but should be discussed more.

The above considerations address the paper on the assumption that all data are valid and conclusions fully supported by the observations. But there are quite a few issues that need to be addressed before one is really sure that these criteria are met. These are discussed below. Maybe I'm just slow. But for me, this paper was extremely difficult to read and understand. This is, in part, because the underlying effects and issue are complicated. But in addition, there are a variety of problems with presentation and analysis and interpretation and logic/assumptions. Important experimental details are absent or buried or written “in plant language”. There is a mixture of ideas and experiments. There is a historical/narrative presentation, rather than a consideration of the data per se irrespective of how the authors came to some ideas. This latter feature makes it extremely difficult to think about what could be going on.

Part III: More comments of various types.

1) It is a well-established fact that heterozygous lines may have higher or lower recombination rates than (more) homozygous lines. This is reiterated by the authors, in considerable detail. As the authors state, these could reflect differences in the natures of the diffusible molecules produced, which the authors call “*trans*” effects.

There is a major vocabulary problem. In opposition to *trans*, there is *cis*. There a major confusion with regard to the word “*cis*”, which is used in two different ways at different places in the paper. The most general way is in opposition to “*trans*” as defined above. That is: the effects of heterozygosity per se, irrespective of differences in diffusible factors. However, the word cis can/is also used to refer to the effect of the nature of one region on events in an adjacent region on the same bivalent, that is an effect that is in cis along a given chromosome. The two different uses make reading this paper really difficult. For example: “dual *cis* effects” probably uses the word “*cis*” in the first sense; and “*cis* effects of heterozygosity” probably uses the word in the second sense.

2) There is additionally what appears to be a fundamental experimental problem. The entire paper (after Figure 4) presents the “genotypes” of tester bivalents, and analyzes events on those bivalents, without ever discussing anywhere that I could find in the text what is known about the other chromosomes in those same cells and whether variations in *trans*-acting factors might be relevant or controlling for such variations in any way. The problem begins with Figure 4 and its relatives. It is stated that the pattern described in that figure, which involves the presence of homo/heterozygosity adjacent to a heterozygous test interval, is independent of the nature of the homozygous region. But this is not correct. Genotypes at the top of the pattern tend to have Col/Col (red) while genotypes at the bottom tend to have Ct/Ct (green). This is seen also in the other analyzed sub-telomeric interval (Figure 4). This should be specifically investigated. More generally, there has to be some attempt to deal with this as a general issue for all experiments. Perhaps this was actually done by analyzing multiple lines with the same critical genotype but, necessarily, different complements of information on other chromosomes. Are these the multiple data points in various figures? If so, it is just not obvious in the text of the paper or the figures or figure legends and the reader should not have to go searching for it. If not, this issue remains as an underlying problem throughout the paper. There are arguments against *trans* effects as a big issue in several places, but they are never made in the paper.

Summary: this issue has to be specifically addressed for Figure 4 in order to draw the conclusion in that figure and it needs to be discussed as an issue in all other cases.

3) The paper makes many comparisons among different regions without much emphasis on the fact that things are different or why. Of course part of the difference between the CEN interval and the 420 interval is the location; but also there is the fact that CEN is measured in male meiosis where in *Arabidopsis*, there are more COs, and they are more sub-telomeric. 420, which is sub-telomeric, is measured in female meiosis, where CO rates are generally lower and COs tend to be LESS sub-telomeric. It is frustrating that the authors did not discuss how these differences, particularly the M/F differences, might affect their results and/or interpretation. To the only partly initiated, it seems like apples and oranges.

4) At the most general level, this paper is among the first, in a plant system, to consider *cis* effects in either sense defined above. In the first category of cis, there is a long history of studies on the effects of basepair mismatches on recombination, both biochemically (they are sensed by RecA and RecA homologs, in combination with mismatch repair proteins, such that the ongoing strand exchange is rejected) and genetically, where recognition of mismatches by the mismatch repair system is known, in meiosis, to specifically eliminate crossovers (e.g. Hunter and Borts). I actually don't know whether anyone has examined DSB levels in heterozygous regions. I am not aware of any previous study that examines either interference or the effects of *fancm*-type or similar mutations in heterozygous regions. I am also not aware of any other studies that address *cis* effects in the second sense. So, as phenomena, the reported observations will be of interest to the meiosis field.

5) To set a baseline, it is easiest to begin with consideration of general properties of heterozygous regions. The authors make two new findings.

A) *Fancm* mutations have different effects on CO frequency in heterozygous versus homozygous regions. However: the interpretation of this finding in the paper is incorrect. The authors assume that the COs that emerge specifically in a *fancm* mutant are going by the same “non-interfering crossover pathway” that is argued to exist at a low level in wild type cells. If this were true, then effects seen in *fancm* mutant backgrounds would also be occurring in wild type. As stated above, this is possible, but it is not known and is not shown by this paper.

More specifically: is generally agreed (e.g. by Mercier and supported indirectly by yeast studies) that there is a set of DSB-initiated intermediates, among which a subset are specified to be “interfering COs”, ie *Zip4*-dependent COs that show classical interference. The remaining interactions are then matured to other fates, mostly non-crossovers and inter-sister events but also, at a low level to COs. These are the “non-interfering COs”. In a *fancm* mutant, the fates of the interactions that are not specified to be “interfering COs” are different, with a much larger proportion becoming COs than in the wild type case. The reason for this is unclear, but presumably is some modulation of biochemistry. What is shown in this paper is that, in a *fancm* mutant, the level of “extra” COs that arise in heterozygous regions is less than in homozygous regions. This does not speak to what might or might not be going on in wild type.

Another problem with this analysis is that wild type, *fancm* and *fancm zip4* strains are analyzed but *zip4* single mutant strains are not. This is not acceptable for a properly controlled analysis, although I understand that it did not seem “relevant” in the context of what the authors were trying to address. Moreover, the missing mutant might actually be directly informative about what is going on in a wild type background.

There is the always-underlying issue of possible *trans* variations; however, it can be argued that a consistent picture emerges in several types of constructs that should (apparently?) have different types of *trans* effects.

B) The authors find that heterozygous regions show higher levels of (phenomenological) CO interference than homozygous regions. This is shown by a three-factor cross for one sub-telomeric region for Col/Ct heterozygosity and by a different method for a broader range of heterozygous lines. The latter analysis suggests that it is general.

This is an interesting and novel finding. It implies that, as an observational fact, within a heterozygous region, if there is a CO in one region, the probability that a CO will occur nearby is reduced even more than in the normal homozygous case.

[As summarized above: there are two ways to think about what this means in reality. The “normal” way might be to think that the interference signal spreads out for a longer distance in the heterozygote. However, there is another way to think about it. The same result can be obtained if there is a reduced propensity for a recombinational interaction to be specified to be a crossover. This is shown by the “beam-film” model (or any scenario involving designation of a CO and spreading interference) by shifts in CoC curves to the left or to the right according to variations in CO-desigation probability, with a constant “interference distance”.] This is an intriguing finding per se. Its significance for other phenomena is more complicated, as discussed above.

6) The authors spend a lot of time discussing situations in which they believe that events in a heterozygous region are influenced by events in a homozygous region, and/or vice versa. First, comments on Figures 4, 5 and 6:

Figure 4. The left part shows that a pure homozygous line, heterozygous for the red and green tester markers, is produced by crossing a marked Col/Col line with an unmarked accession line (top) and then that when this line is selfed to give an F_2_, various types of progeny can result.

Comment. The authors show only cases in which the R and G markers are in cis, which implies that during selfing, there has been no crossing-over between them. This may be the majority of outcomes. However, the criterion for selection of F_2_ progeny to analyze was heterozygosity for R and G. By fluorescence, this implies that there is one copy of each. For all of the examples shown, R and G are “in cis” to one another. But 15% of meioses will give a crossover between the two markers; and if the appropriate products unite (R+G- with R-G+), the resulting F_2_ seed will ALSO have only one copy of each marker. This may be rare (15% x 15%). Nonetheless, such cases would be scored as having the highest recombination rates if they were not somehow identified and removed from further analyses. Do the authors know from genotyping whether their F_2_s include such cases and if so, which ones are they among the 139?

Other “missing information”: (i) Were the F_2_'s obtained from a single F_1_? Or from multiple F_1_'s with results pooled? (ii) Is it completely obvious that the fluorescence detection could distinguish RRG and RGG from RG genotypes? This is clearly a critical point and it does not seem to be addressed explicitly, though I might have missed it.

Minor presentation issue: the meaning of the box on the right is obscure without working at figuring it out. Apparently it means that when the F_2_s are taken through meiosis, they may generate recombinants, but this is hardly obvious in the cartoon or the legend. (Again assuming that they started out with the cis configuration.)

Figure 4. In this figure, the authors select a set of F_2_s that are heterozygous at the R and G markers. They then ask, at each of 51 other positions, whether the frequencies of the two parental alleles is different from what you expect from an absence of correlation with the selected “phenotype”. What is detected is a deviation from expectation for chromosome 3 (the chromosome on which phenotype is selected).

Comment: We can start with the fact that this analysis is essentially not described at all in the paper. The non-plant reader would have no idea what is going on from the half a sentence in the text.

If I have understood correctly, this analysis appears to have a fundamental flaw. If you select for F_2_'s that are heterozygous for R and G, most of these will have the cis configuration, meaning that they have not undergone a CO between R and G on that chromosome in the previous meiosis. Since there must be at least one CO, there will be a CO somewhere else on chromosome 3. And there will be a non-random tendency for marker disposition to occur just because there is already a selection that restricts localization on this chromosome specifically. More specifically: since the average per chromosome is just a little above one, and since the CO is not sub-telomeric at the marked end, it will tend to occur at non-random position (or positions if there are 2 COs) away from the marked end, e.g. towards the middle of the chromosome. This effect explains the peak in the graph, which is centered in the middle of chromosome 3: it reflects the non-random consequences of having the CO in a non-random position along the marked chromosome. If this interpretation is correct: this data do not provide evidence for a *cis* effect. There is only the fact that you have pre-selected a population that, by its nature, will have a non-random arrangement of markers along chromosome 3.

Two further points: (i) 139 is a very small number of F_2_s from which to draw a conclusion. (ii) There should be a control showing that if you look at the entire population, you do not see any peak. Of course, given the basis for the peak (above), the control will show that it is specific to the selected sub-population, which is not really any help, but still, if such analysis is to be shown, the control must be there.

Figure 4. Figure 4 shows the recombination frequencies in the 420 interval observed for the 139 F_2_'s. The conclusion is that the range is greater than for the Col/Col wild type strain or for a Col/Ct F_1_ full genome heterozygote. This is likely true, but it is hardly surprising under any model. It likely reflects *trans* effects as well as *cis* effects. It is unclear why this is valuable information, unless it sets the stage for later panels.

Minor point: it is not appropriate to compare the rank plot for the 139 F_2_s with averages and ranges for Col/Col and Col/Ct. The ranges for the two compared lines are a statistical measure; they do not give the same impression as the rank plots. Thus one is comparing “apples and oranges”. The rank plots for the two comparison lines should also be shown, and then the averages and ranges for all three sample sets should be compared.

Figure 4; Figure 5. Figure 4 displays the natures of the 139 F_2_s, shown from top to bottom in rank order of recombination frequency in the 420 interval, with respect to whether the markers indicated on the X-axis are heterozygous (black), homozygous Col, or homozygous Ct. This plot shows a gradient of “black”, emanating from the test interval towards the end of the chromosome, to greater and greater distances from bottom to top. The authors interpret this data as follows: “However, we detected a significant association on chromosome 3 itself, where the hottest F_2_ quartile had significantly higher homozygosity outside 420, compared to the coldest quartile. This *cis*-effect was observed when the rest of the chromosome was either Col or Ct homozygous, indicating that it is non-allele specific and caused by polymorphism per se, rather than by specific sequence variants.”

The restatement of the result above does not accurately describe the entire picture. The fact that the black portion emanates from the selected region is not mentioned (and is presumably significant). An accurate restatement of the result, which admits to all interpretations, is that F_2_s with higher 420 recombination tend to be homozygous for flanking material while F_2_s with lower 420 recombination tend to be heterozygous. Moreover, the “extra heterozygosity” towards the bottom of the rank plot tends to occur specifically adjacent to the 420 region. The term “*cis* effect” should not be used, as this is an interpretation. It has become lab jargon for the phenomenon the authors think is going on, but it is confusing given the various versions of this phenomenon (including its reciprocity), as well as the fact that there should be a formal description of the data, not an interpretive one. Better to call it a “junction phenomenon” or Hom-Het interplay.

More importantly, contrary to what the authors say, there is a clear tendency for more “red” at the top and more “green” at the bottom, not only in Figure 4 but in the corresponding figure for the other sub-telomeric region in Figure 4. This would likely appear in an appropriate plot, although a larger data set may be required to see it with very strong statistical significance. It suggests that having more Col information promotes a higher frequency of COs in general versus more Ct information. This is directly supported by the fact that, overall, Col/Col recombination rates are higher than Col/Ct recombination rates genome-wide (Figure 4). This has to be dealt with.

Also importantly: the authors cannot, in principle, say that they see a *cis* effect unless they do a control to show that the sample subset of 139 F_2_s have similar frequencies of recombination throughout the genome and that there are or are not any associations of recombination with homozygosity, and/or Col vs Ct content on a genome-wide basis. I appreciate that this is not trivial. But the authors' hoped-for conclusion is not trivial either. It cannot be made from this figure without more supporting (or contradicting) data.

Secondarily (perhaps): there are general uncertainties. (i) There is the possible problem that the hottest F_2_s might have their markers in *trans*, with unknown consequences. (ii) The top-ranked F_2_ in the 4D data set is very pecuIiar. It appears to be Col for the entire length of one chromosome and to have had a single CO exactly at the border of the marked region in the other. How is this possible? (iii) As in Figure 4, there could be unknown effects from selecting the subset of F_2_s that did not have a CO in the sub-telomeric region. Are there markers all the way to the end of the chromosomes or are sub-telomeric crossovers going to be missed (which is a problem for the analysis since many COs will be there specifically in the selected population, thus biasing the sample. There is actually a hint of such a problem in the fact that the observed “effect” is absent for the centromere (a central marker) and weak for an interstitial marker. One interpretation is that observation of an effect is a consequence of selecting a sub-telomeric region for analysis (rather than being an actual phenomenon is specific to such regions). The authors have only presented data saying whether each F_2_ is homozygous or heterozygous for the markers tested. But in principle, they could figure out the exact arrays of markers on the two chromosomes of the F_2_ and thus were crossovers occurred in the preceding F_1_ cross(es). While perhaps difficult, this might reveal additional informative information to further guide interpretation.

Overall, data in Figures 7 and 8 tend to ameliorate concerns as to the validity of the authors' conclusions from Figures 4 and 5. But the concerns are still real and valid for presentation of this data.

Figure 6. Panels A-C show that two lines that are heterozygous for 420 and Ct/Ct for all or most of rest of the chromosome exhibit higher 420-region recombination than a line that is homozygous Col/Col in 420 and heterozygous for the rest of the chromosome. The authors think that the relevant difference is that heterozygosity adjacent to homozygous 420 depresses recombination in 420. They use this data to argue that the presence of homozygosity adjacent to heterozygous 420 “increases” recombination, and thus “eliminates the *cis* effect”.

First: this is not “eliminating the *cis* effect”. It is a different type of *cis* effect. It is not clear that the authors appreciate that they are seeing reciprocal effects. They are totally focused on the effects of heterozygous regions on homozygous regions.

In any case, without controls, there are several hypotheses that are equally consistent with the data. What is the relevant difference(s)? Heterozygosity in 420? Col/Col in 420? Heterozygosity or Ct/Ct in the rest of the chromosome? Some combination of the two? There is no way to know without more lines with different genotypes. What is needed (minimally) is Heterozygous 420 plus homozygous Col/Col and fully homozygous Col/Col. (Homozygous Ct/Ct is desirable but difficult to construct). No conclusion can be drawn without controls. This is just not acceptable. To a first approximation, this data should be omitted.

And to repeat: it is assumed that the only relevant differences among the different lines is on the chromosome of interest. But all of the chromosomes are different in the different strains. Is there evidence that the variations are not due to *trans* effects.

Another thing that is very unclear is why these particular lines were picked for testing. Both of the HL lines chosen for 6 A/B seem to be the rare type in the top quartile in Figure 4; most of the lines in this quartile are homozogous Col/Col outside of 420. Why were these chosen rather than the more common type; or, better, why not test both types? And specifically, exactly which of the 139 F_2_s are HL1 and HL2? It is not obvious.

Figure 6. This figure shows a 3-factor cross in which there are two intervals that comprise most of the single 420 interval in (AB). (DE) shows that HOM adjacent to HET results in an increase COs in the HET region and that HET adjacent to HOM results in a decrease in COs in the HOM region. This points to the reciprocal effect shown further in Figures 7 and 8. (F) shows that, despite these changes, there is no change in interference between the two intervals. To anticipate later results, this implies that the nature of interference in the HOM region predominates (below).

Figures 7 and 8. Figures 7 and 8 have three components (1) They provide information about *fancm* effects and interplay of heterology with *fancm* recombination. These were discussed above. [The observed effects pertain specifically to heterozygous regions regardless of what is or is not adjacent to them. They do not say anything about what is going on in wild type non-interfering COs.] (2) They provide more data for wild type (*FANCM*+) strains in several situations. (3) They show that interference is required for the so-called “*cis* effects”. The exposition in the paper was hard to follow. Here is my own restatement of the situation.

Issue (2). Figure 7. For the 420 interval, there are four CO frequencies: HOM/HOM 18cM; HOM/HET 15cM; HET/HET 21cM; HET/HOM 28cM. It is unclear what the multiple data points represent. If these are multiple different lines (and thus different “*trans*” effects, then the data, are more meaningful). Just looking at these numbers, some conclusions emerge:

Adjacent HET reduces COs in HOM (in HOM/HET 15cm relative to HOM/HOM 18cm). This is the effect emphasized above.

Oppositely, however, adjacent HOM increases COs in HET (in HET/HOM 28cM vs HET HET 21cM). So the effects are reciprocal, which is not emphasized enough.

For the CEN interval, HOM/HOM 12cM; HOM/HET 12cM; HET/HET 10cM; HET/HOM 9cM. Clearly the centromere behaves differently, for whichever reasons, as suggested by Figure 5 data.

Figure 8 addresses the same to issues above, but for a three-factor cross specifically in the sub-telomeric interval and with the addition of interference analysis.

Left interval: HOM/HOM 16cM; HOM/HET 11cM; HET/HET 16cM; HET/HOM 18cM.

Right interval: HOM/HOM 5cM; HOM/HET 4cM; HET/HET 5cM; HET/HOM 6cM.

In this case:

The left interval behave as the interval in Figure 7, which is the sum of left and right in Figure 8.

HET reduces COs in HOM: HOM/HOM 16cM; HOM/HET 11cM; HOM increases COs in HET: HET/HOM 18cM; HET/HET 16cM.

The right interval behaves oppositely, but with small effects;

HET increases COs in HOM: HOM/HOM 5cM; HET/HOM 6cM;

HOM reduces COs in HET: HOM/HET 4cM; HET/HET 5cM.

The right interval is small. Apparently the “left side dominates”. I have no idea what this means.

Figure 8. Interference is also analyzed in the wild type background. NB: there is nowhere in this paper (except the Methods) a description of what the interference metric is, even on the figure. Axis must be labeled (1-CoC) and this must be explained in the text.

Results: HOM/HOM 0.6; HOM/HET 0.6cM; HET/HET 0.8; HET/HOM 0.8.

(a) HET/HET has “more interference” than HOM/HOM, the effect discussed above Importantly, this is despite the fact that the frequencies of COs in both intervals are the same in these two situations! This would not be expected if the only difference were in interference. Thus, in the pure heterozygous situation, there must be two effects, one that increases COs plus interference that decreases COs, e.g. more DSBs and counterbalancing interference increase. (b) The two reciprocal “mixed” cases are not the same. HOM/HET looks like HOM/HOM while HET/HOM looks like HET/HET. Formally, the nature of the left interval dominates, as also seen with respect to effects on CO levels (above).

Overall, it seems that the ability of a HET region to depress COs in a HOM region does not depend on having a HET-type CO interference process.

Figure 8: In the larger/left interval, where canonical reciprocal effects are observed: HET reduces COs in HOM: HOM/HOM 16cM; HOM/HET 11cM; and for interference, HOM/HET looks like HOM/HOM. HOM increases COs in HET: HET/HOM 18cM; HET/HET 16cM and for interference, HET/HOM looks like HET/HET, and the same is seen in Figure 6. This figure compared a HOM/HOM situation with a HET/HOM and found them to have the same (HOM) interference. And the “canonical” reciprocal effects are observed.

7) The last item is the interplay between *fancm* and CO interference, keeping in mind that the information does not tell us anything about what is going on in *FANCM*+ (and thus the general cases of heterozygosity seen in nature).

One expectation is that if more COs are coming from the *fancm*-revealed process, there will be less interference. This expectation is met in Figure 8. More or less, there is plenty of interference in all four strains in *FANCM*+. And more or less, in both HOM/HET and HOM/HOM, where the entire test region is homozygous and the *fancm* mutation is having its full effect, there is a big reduction in interference. Whereas, in both HET/HOM and HET/HET, where the entire test region is heterozygous and there are fewer extra COs in the *fancm* mutant, interference is less reduced. And the zip4 mutation, in the *fancm* background, has the expected effects in all cases, reducing interference because it decreases the fraction of “interfering” COs. These results provide more support for the nature of the *fancm* interplay with heterozygosity (discussed in 2 above).

The authors also ask about variations in the “interplay” of *fancm* with HET/HOM differences. The underlying idea (which I finally realized after many hours) is that in *fancm zip4* there are still COs but they do not show interference, so one can ask if the variations in 420 according to what is adjacent are still seen. If no, then interference is required.

The data for 420 in Figure 7 are:

HOM/HOM 35cM HET/HET 12cm HET/HOM 10cM HOM/HET 35cm. Most importantly, since HOM/HET = HOM/HOM, there is no transmission of information from HET to HOM if interference is absent, which implicates interference in the process (in accord with the suggestions above). The reciprocal case: HET/HET 12cm HET/HOM 10cM seems to run against all rules, with HET increasing COs in the adjacent HET, but again, maybe this is a small effect and/or can be ignored. The fact that HET/HET << HOM/HOM is explained by the fact that there are fewer COs from the *fancm* pathway in heterozygous regions (above).

Similarly in Figure 8: for left and right regions

HOM/HOM 25, 12cM; HET/HET 8, 3.5; HET/HOM 7.5, 2.5cM HOM/HET 22, 12cm. These data are clear and very useful in showing that interference is relevant.

8) A final point on statistics: p-values are given everywhere as Chi-square p-values, but it seems the authors are comparing means, not comparing a result against an expectation. Is this a typo? Is it a different kind of test, or did they really use a Chi-square? (If the latter, they need to justify and clarify what the expectation was, otherwise give p-values for something that compares distributions, e.g. a T-test).

Reviewer #3:

There is no question that the manuscript of Ziolkowski et al. is of interest as it is challenging current views of crossover control, not only in plants.

I also read the thoughtful comment and valuable suggestions of both reviewers with great interest. I think that many of the points raised by the reviewers the authors were able to address in their response, especially by the inclusion of new experimental data.

I have to say that the main conclusion that interfering crossovers are enhanced in heterozygous regions is for me as for the other reviewers counterintuitive. Also, the model the authors bring forward at the end, that recombination intermediates are stabilized by mismatch recognition is quite special. Therefore it is of course important to take other explanations into account.

There is one thing that worries me a bit and that is quality of the sequence data available for the different *Arabidopsis* cultivars. Do heterozygous region really attract crossovers from homozygous regions in hybrids? As the authors state themselves, these sequences were aligned by short read technology and so the occurrence of inversions and of duplication in the range from hundreds to hundred-thousands of bps might be dramatically underestimated. Thus, changes above the nucleotide level were not taken into account by the authors. Nevertheless these changes might drastically influence recombination patterns.

Nevertheless, all in all, the authors supply us in the revised version with enough hard data that in my opinion strengthen their hypothesis to the point that it should be considered (and challenged) by the community. Also, it might well be that such a mechanism is exceptional and restricted to self-fertilizing species like *Arabidopsis*. But this would be still interesting.

---

## [Author Response]

[Editors’ note: the author responses to the first round of peer review follow.]

*There were two types of concerns. First, the reviewers were not convinced that heterozygosity generally increases crossovers (independently of genetic background and physical location in the genome), and that crossovers were recruited from homozygous regions of the genome to adjacent heterozygous regions*.

We have added substantial new data to the paper to directly address these concerns. To address the question of generality of the heterozygosity *cis*-effect:

i) In Figure 4 we present new data showing the *cis*-effect on chromosome 2 (*I2fg* Col x Ct), demonstrating that it is not restricted to chromosome 3. We have also observed the *cis*-effect in both sub-telomeric (*420*, *I2fg*) and centromeric (*CEN3*) chromosome regions (Figures 4 and 5).

ii) We present new data showing the *cis*-effect in a Col x Ler cross, demonstrating that this phenomenon is not specific to Col x Ct (Figure 5).

To address the question of crossover recruitment from homozygous to heterozygous regions:

i) We have performed three-colour flow cytometry that measures crossovers in adjacent regions. This has directly shown compensatory crossover changes in adjacent intervals when heterozygosity is varied (Figure 6).

ii) Three-colour experiments have been performed with *fancm* and *fancm zip4* mutants to measure crossover interference (Figure 8). This demonstrates that the heterozygosity *cis-*effect is associated with an increase in crossover interference, as predicted by our model.

*Similarly, the reviewers found that the mutant analysis was not fully developed*.

We agree with Reviewer 2 that analysis of a *zip4* mutant alone would be of interest. We have attempted to analyse *zip4* using flow cytometry but saw variable and inconsistent effects. We believe this is an artefact caused by gamete inviability and a corresponding loss, or alteration, of pollen fluorescent properties ([20], PLoS Genet.). Therefore, we are reluctant to use these data to draw conclusions. The major advantage of analysis in *fancm* and *fancm zip4* is that fertility is restored due to a large increase in non-interfering crossovers, allowing us to collect high quality fluorescence data in backgrounds with dramatically altered levels of interfering and non-interfering crossover repair ([31], Science). Importantly, we see significant differences between *fancm* and *fancm zip4*, which allows us to make inferences about the contribution of the interfering crossover pathway to the *cis-* effect. Finally, our new *I3bc* three-colour data allows measurement of crossover interference, which has also been completed in *fancm* and *fancm zip4* mutants and supports the model that heterozygosity promotes interfering and inhibits non-interfering crossovers.

*Second, there was agreement that, while your findings are provocative, they are not appropriately framed vis à vis alternative explanations for the observed relationships between recombination rates and diversity along genomes*.

We appreciate the need to more fully discuss literature relevant to the relationship between recombination and sequence diversity. To strengthen this aspect of the paper we have added analysis of historical recombination and genetic diversity based on maps we published previously to Figure 1 ([24], Nat Genet). These maps were generated using SNP data from 80 Eurasian accessions sequenced by the Weigel laboratory as part of the 1,001 genomes project ([17], Nat Genet). Here we show that recombination and diversity are positively correlated at multiple scales and add new plots to show their striking relationship in the chromosome arms. In the Introduction and Discussion we have expanded on factors that could influence this relationship, including background selection and genetic hitchhiking, which will cause regions of low recombination under directional selection to have low diversity. We also discuss the potential for recombination-associated mutagenesis. We believe the heterozygosity *cis-*effect contributes to the positive relationship between recombination and sequence diversity, together with these other forces.

Reviewer #1:

*The authors argue that recombination may be recruited from homozygous regions to nearby heterozygous regions. Overall I found the results interesting but I have some significant concerns about the presentation and interpretation of the results*.

*One of my major concerns about the article is that the authors frame it as a possible explanation for the positive correlation between crossover frequency and diversity that is seen across a number of the species examined to date. However, there is relatively good support, in population genetics literature, for the idea that this pattern reflects the fact that linked selection (e.g. hitchhiking and background selection) have effects over relatively large physical regions in regions of low crossing over (see*
[32]*). The authors do not seem to mention this dominant, and relatively well supported, group of hypotheses and instead seem to frame the positive correlation as an empirical observation in search of a mechanistic hypothesis (that they provide). Obviously these hypotheses are not mutually exclusive, and so this does not rule out the idea that the authors' mechanistic explanation could also play a role. The authors need to be much clearer about the background to these observations, if they are to contribute to the larger debate about determinants of levels of polymorphism*.

We thank the reviewer for these comments, which helped place our data in a firmer context. To address these important points we have added new analysis showing a positive correlation between historical crossover and sequence diversity in *Arabidopsis* (Figure 1) and extensively rewritten the paper to include reference to the relevant literature, as described above.

*Along these lines it is not totally clear to me whether the observations could possibly explain a bulk of the correlation between recombination and polymorphism in many species. Much of the broad variation in recombination rates, in many systems, is explained by proximity to centromeres and telomeres (presumably a mechanistic constraint). Thus a lot of the variation in broad-scale recombination rates is not tied to polymorphism, but rather to large-scale chromosomal architecture*.

We strongly agree with the reviewer that chromosomal location plays a major role in determining recombination rates. For example, the sub-telomeric interval *I2fg* shows higher recombination (13.02 cM/Mb) than the centromeric *CEN3* interval (2.11 cM/Mb). *CEN3* is largely heterochromatic, which is known to be inert for recombination and we have previously shown that DNA methylation in this region is important for normal crossover patterns ([114], PLoS Genetics). The high rate of *I2fg* recombination also derives from the fact that the sub -telomeric regions are highly recombining during male meiosis and *I2fg* measures fluorescent pollen ([45], PLoS Genet). Therefore, chromosomal location exerts a significant effect on recombination rates when comparing between intervals. However, comparing variation within a single interval between crosses controls for these differences. We have further discussed differences in interval location and chromosome architecture in the F_1_ recombination Results section.

*Once again the authors' proposed mechanism could contribute to the strength of the correlation, but this argument does suggest that this contribution may be somewhat minor*.

We agree that the *cis*-effect is one of a number of forces that could influence diversity and its relationship to recombination. In the revised manuscript we have framed this question and the relevant literature more clearly.

*The authors state, in the Abstract, that “using recombinant populations we show that heterozygous regions attract crossovers from homozygous regions on the same chromosome at the megabase scale*”*, and in one of the subsection of the Results that “Heterozygosity recruits crossovers over homozygous regions*”.

*I apologize if I've missed something, but I do not think the authors’ results actually demonstrate this. To my reading of these statements it sounds like the authors are saying that the extra recombination events in heterozygous regions come at the expense of fewer recombination events in the homozygous regions. But I do not see any results supporting for the second part of this statement, i.e. measuring recombination rates in homozygous regions. The authors' statements are one possible explanation of the results, but another is that the homozygous regions promote recombination with no* “*cost*” *to themselves. The authors may have evidence in favor of their hypothesis, perhaps this is the argument being made in the subsection headed “Differential sensitivity of interfering and non-interfering crossover repair to polymorphism”, but I found it hard to follow*.

To provide direct evidence for this model we have performed three-colour fluorescent pollen analysis shown in Figure 6. Three-colour analysis allows simultaneous measurement of crossovers in adjacent intervals and calculation of crossover interference ([115], Nat. Prot.). For these experiments we constructed the *I3bc* interval, which overlaps with the fluorescent seed interval *420*, but consists of three FTL T -DNAs expressing blue, green and red fluorescent proteins. The *I3bc* system was generated in a Col background, and for these experiments we crossed with the Ct accession. Specifically, we generated plants that were Col/Ct in *I3b* and Col/Col over the rest of chromosome 3 including *I3c*, and compared to pure Col/Col homozygotes. We observed that when *I3b* was Col/Ct heterozygous genetic distance increased and decreased in the adjacent homozygous *I3c* interval. These compensatory crossover changes provide direct support for the model that heterozygous regions can promote crossovers at the expense of adjacent homozygous regions.

*One concern I had was that recombination may be increased over broad chromosomal regions due to homozygosity in a specific region, because the individual would have reduced heterozygosity for structural rearrangements. This increase in recombination would not be associated with either parental allele, but with homozygosity itself. Presumably the authors have thought this through, and perhaps these lines are known to have no structural variation of suitable size in these regions, but it is worth clarifying this point*.

We agree that structural variation between accessions could contribute to variation in F_1_ hybrid recombination rates. Indeed, we see clear evidence of such effects in the *420* Col/Sha F_1_ data. *420* overlaps a ∼2.5 Mb inversion previously mapped between Col and Sha ([91]; Heredity, [72], Theor. Appl. Genet; [98], Genetics). Consistent with this we observe that Col/Sha inversion heterozygosity strongly suppresses *420* cM in this cross. Importantly, a Col x Ct genetic map generated alongside the Col x Sha map did not show evidence of similar inversions ([98], Genetics). Therefore, in our *420* Col/Ct experiments we think that inversions of comparable effect do not exist. However, the true extent of inversions or other structural rearrangements are unknown between all of the crosses analysed. However, the fact that we see the same heterozygosity *cis-*effect in three crossover reporter intervals and two backgrounds reduces the chance that structural variation is confounding our analysis.

*At the end of the subsection headed “Modification of crossover frequency by* Arabidopsis *natural variation”, the authors state:* “*therefore, in many cases heterozygosity promotes* Arabidopsis *recombination relative to homozygosity, which is inconsistent with a purely suppressive effect of polymorphism.*” *I don't think this is correct as stated. The authors have shown that when Col is crossed to other lines the F*_*1*_*s sometimes have higher recombination rates than Col homozygotes. However, this could be because the other lines crossed to harbor recombination modifiers that increase recombination rates. For the authors claim that heterozygosity promotes recombination to be true they would need to show that the F*_*1*_*s often have higher recombination rates than either of the parental lines*.

We agree that the F_1_ observations do not demonstrate that heterozygosity attracts crossovers. Therefore, we have modified the text to state that variation between F_1_ recombination rates is likely caused via a combination of *cis* and *trans* modification by polymorphism.

Reviewer #2:

*This paper presents evidence for three conclusions*.

*1) In one well-studied test interval, in a particular heterozygous state, the frequency of crossovers is higher when the region adjacent to the test interval is homozygous rather than heterozygous. This is a* “cis” *effect. This is shown in*
Figure 4
*and in the non-mutant background of*
Figure 7
*(compare panels A and C)*.

*There are some limitations to this observation. First, it is shown for only one interval*.

To provide evidence for the generality of the heterozygosity *cis*-effect we have added new data showing the same phenomenon at a further sub-telomeric interval *I2fg* on chromosome 2, in a Col x Ct F_2_ population (Figure 4). This demonstrates that the *cis-*effect is not chromosome specific. For both *420* and *I2fg* the *cis*- effect acts over similar megabase distances. We have also observed the effect in a *CEN3* Col x Ct F_2_ further reinforcing our model (Figure 5). Although the strength of the effect was weaker for *CEN3* this is likely to be due to the *cis-*effect acting independently from both chromosome arms. Finally, we have added new data showing the *cis*-effect in a *420* Col x Ler F_2_ population (Figure 5). The effect was weaker in the Col x Ler cross relative to Col x Ct. However, we have unpublished data that strong *trans* recombination modifiers are segregating in Col x Ler, and not in Col x Ct (PAZ and IRH unpublished observations). We are in the process of mapping these loci, but this explains why it is harder to see the *cis*-effect in this cross. We have also added new three-colour *I3bc* data using Col x Ct populations (Figures 6 and 8). This interval overlaps *420* and provides an independent confirmation of the *cis*-effect in this region, and additionally shows that crossover interference increases in heterozygous regions.

*Second, interval is sub-telomeric and thus likely not to be representative of most of the genome*.

Although the strongest *cis*-effect was observed when measuring sub-telomeric intervals (*420* and *I2fg)*, the effect extended for most of the chromosome arm. We have also seen a significant *cis-*effect at *CEN3*, which demonstrates that different chromosomal regions, not just the sub-telomeres, are sensitive to the *cis-*effect (Figure 5).

*Third, it is shown for only one pair of lines (Col vs. Ct)*.

We have added new data from a *420* Col x Ler F_2_ population showing the *cis*-effect, thereby demonstrating that this is not specific to Col x Ct crosses, as discussed above (Figure 5).

*It is also important to note that in this test interval, there is no difference in crossover level between the homozygous and full F*_*1*_*hybrid strains (*Figure 7
*compare panels A and B in the non-mutant case). Thus, there is no general effect of heterozygosity to increase the number of crossovers. This is also seen in the overall evaluation of Col/Col vs Col/Ct F*_*1*_*'s. Thus, the identified phenomenon is some type of unusual cis interaction which may or may not be widely general*.

We agree that the F_1_ data does not demonstrate the heterozygosity *cis*-effect. We believe that variation in crossover rate between crosses is due to a combination of *cis* and *trans* modifying effects. The *cis*-effect is only evident though analysis of F_2_ populations. We have modified the manuscript to make this clear.

*The basis for this cis interaction proposed by the authors does not seem to make sense to me. This is in part because it seems to invoke result (2) below in an inappropriate way and in part because the statement is made that crossovers are* “*recruited*” *from homozygous regions. But this conclusion cannot be drawn. There was no analysis of the effects of a heterozygous region on a flanking homozygous region*.

*It seems that this conclusion of some type of* “*competition*” *between homozygous and heterozygous regions is drawn largely from*
Figure 6*. But the results in this Figure do not support this conclusion. What this figure shows is that when the test region is homozygous and the flanking region is heterozygous, the level of crossovers is reduced in the test region as compared to the case where the test region is heterozygous and the flanking region is homozygous. This does not imply that there is a reciprocal effect of heterozygosity to reduce crossovers in the homozygous region. There is no control to show what the frequency of crossovers is in a fully homozygous case where there is no heterozygous region adjacent to the test region*.

We have directly addressed this concern via analysis of the *I3bc* three-colour FTL interval. This interval overlaps *420* and allows measurement of crossover frequency in adjacent regions. When the *I3b* interval was Col/Ct heterozygous it showed increased crossover frequency at the expense of the adjacent *I3c* Col/Col homozygous region. This clearly demonstrates that the *cis*-effect involves compensatory interactions between heterozygous and homozygous regions (Figure 6).

*There is also no control to show what happens if the Col/Col homozygous test region is flanked by the same homozygous region as when the test region is heterozygous (i.e. Ct/Ct)*.

As the FTL T- DNAs were generated in the Col accession to perform a Ct/Ct analysis in the test region would require extensive introgression, most likely in excess of 8 generations, of the FTLs into a Ct background. Even after extensive backcrossing it would be extremely difficult to remove all linked Col/Col sequences in proximity to the FTLs.

*Finally, the authors suggest that the cis phenomenon is general because some F*_*1*_
*hybrid strains show higher levels of recombination than one particular reference homozygous strain. However, this finding is just as easily explained by general genetic background effects on the recombination process rather than anything to do with heterozygosity at the DNA level, as the authors admit. Thus, this is not really supporting evidence*.

As discussed earlier we have revised the text in reference to our F _1_ data to state that these data are most likely explained via a combination of *cis* and *trans* modification effects.

*2) The authors show that heterozygosity is accompanied by a change in the sensitivity of crossovers to* fancm *and* zip4 *mutations. The analysis is not rigorously correct because there was no test of a* zip4 *mutation by itself; the only test was in a* fancm zip4 *background*.

*The most important point is that this is a general characteristic of heterozygosity: it is observe in a pure F*_*1*_
*hybrid. Thus there is no reason to link this phenomenon to the* cis *effect, as the authors seem to do*.

As discussed earlier we agree that analysis of a *zip4* mutant would be of great interest. However, the major advantage of analysis in *fancm* and *fancm zip4* is that fertility is high due to a large increase in non-interfering crossovers, allowing us to collect high quality flow cytometry fluorescence data in backgrounds with dramatically altered levels of interfering and non-interfering crossover repair ([31], Science). Importantly, we see significant differences between *fancm* and *fancm zip4*, which allows us to make inferences about the contribution of the interfering pathway to the *cis*-effect.

*A second important point is that the total frequency of crossovers in the* F_1_
*hybrid is the same as in the homozygous reference strain. Thus, there is no* “*recruitment of crossovers*” *as the authors also seem to suggest*.

Total recombination at *I3bc* is similar between Col/Col and Col/Ct F_1_ heterozygotes. However, we show that crossover interference in fact increases in Col/Ct heterozygotes (Figure 8). This is further consistent with our Col x Ler F_1_ tetrad data, which reached the same conclusion based on analysis of 4 independent 3-colour intervals. Together these observations are consistent with our model that non-interfering crossover repair is inhibited by heterozygosity and interfering crossover repair is promoted by it. This causes the strength of interference to increase, as a greater number of events will be generated from the interfering pathway in heterozygous regions. We acknowledge that our F_1_ data alone does not directly support the conclusion that heterozygosity attracts recombination.

*Finally, comparison implies a likely situation is that there is a change in the proportion of recombination events that are* “*interfering crossovers*” *and versus crossovers that arise as a minority population from the* “*non-crossover*” *pathway. To put it another way, in the wild type case, there is an increase in crossovers and a decrease in non-crossovers (more of which turn into crossovers in* fancm *than in wild type)*.

*The basis for this effect is unclear. However, it is strongly reminiscent of crossover homeostasis where a reduction in DSBs leads to a differential loss of non-crossovers as compared to crossovers. Since it is unlikely that heterozygosity will be recognized at the DSB level (although this is not totally excluded given diverse indications of* trans *effects on DSBs in yeast), it would instead imply that there is a reduced chance that a DSB actually giving a recombination intermediate that could lead to a crossover or non-crossover. This is entirely possible since heterozygosity could be sensed at the time of establishment of such an interaction or concomitant with crossover/non-crossover decision making*.

We thank the reviewer for these insightful comments, which we agree with. Regarding whether polymorphism can influence DSB frequency, we think this is formally possible as CTT and A-rich motifs associate with hotspots ([24], Nat. Gen*.*; [110], *eLife*), and therefore mutation of these sequences may alter recruitment of the recombination machinery. Equally, as pointed out by the reviewer, *trans* variation could also act to globally alter DSB formation. Due to an absence of clear *trans* effects in our Col x Ct QTL analysis, we think *trans* modifiers are not a major factor in these populations. We favor a model where polymorphism is detected downstream of homologous strand invasion via base pair mismatches. We are currently unable to define where in the pathway this effect is occurring, though our mutant analysis indicates differences in the sensitivity of interfering and non-interfering repair to sequence polymorphism. This could occur, as the reviewer suggests, at the crossover/non-crossover decision point. We have elaborated on these ideas in the Discussion and added a model to Figure 9 to develop these ideas further.

*3) The authors show that an F*_*1*_
*hybrid appears to have increased crossover interference as compared to the homozygous reference strain. There are two problems with this result and linking it to other results. First, this is a different hybrid from the one tested for all other phenomena*.

Our new *I3bc* data shows the same phenomena for Col x Ct crosses (Figure 8), demonstrating this effect is shared between Ler and Ct crosses.

*Second, there is no evidence that this effect is the result of heterozygosity at the DNA level rather than some* trans*-acting strain background effect. Third, in this hybrid the total crossover level is lower than in the reference homozygote; thus, this is not related to the finding that F*_*1*_
*hybrids can exhibit higher levels of recombination*.

As discussed earlier we have revised the manuscript to ensure our interpretation of the F_1_ data is in line with the reviewer’s points.

*That being said: there is a possible way that this result could be relevant to the sub-telomeric effect described above. In* Arabidopsis*, as in several other organisms, interference distance goes with physical chromosome length. Thus, in the hybrid, increased interference per Mb (which is what is assayed) could result from a decrease in physical chromosome length. Furthermore, the* cis *effect described in the results in prior sections could be explained if the longer length characteristic of the homozygous region can spread into the adjacent sub-telomeric region, thus increasing physical length, decreasing interference per Mb and thus increasing crossovers per Mb*.

The reviewer raises a very interesting idea. Our current understanding of genome-wide patterns of polymorphism between accessions is largely generated from short-read technology, which can underestimate structural variation. It is very possible that the mechanism described by the reviewer could explain some of the effects we observe, however, without more accurate knowledge of structural polymorphisms between the accessions we are not able to draw reliable conclusions here. We have added these ideas to the Discussion.

*Given the above considerations, the statements in the Summary need some amendment, as follows*:

*A)* “*We… found hybrids with higher recombination than homozygotes, demonstrating that polymorphism can promote crossovers*”*. If this is a reference to the finding of F*_*1*_
*hybrids with higher recombination, it is not accurate. There is no basis for the conclusion that the higher recombination is due to DNA polymorphism rather than* trans*-acting genetic effects. The* cis *effect is not evidence. Moreover, in the Col/Ct F*_*1*_
*hybrid used for most of the analysis, recombination frequencies are generally not higher than in Col/Col and in the Col/Ler hybrid used for interference analysis, recombination frequencies are lower than in the Col/Col homozygote*.

As discussed we have moderated our statements on the F_1_ data and the contribution of *cis* and *trans* effects. We agree that the F_1_ data do show lower genetic distances in many cases relative to the homozygotes. However, our new *I3bc* experiments in Figure 6 clearly support our conclusion that heterozygosity can promote recombination relative to adjacent homozygous regions.

*B)* “*Using recombinant populations we show that heterozygous regions attract crossovers from homozygous regions on the same chromosome at a megabase scale*”*. This is definitively not shown by these data (see discussion in point 1 above)*.

As discussed earlier we have directly addressed this point with our new *I3bc* experiments.

*C)* “*We demonstrate that this polymorphism cis-effect is dually mediated by promotion of interfering crossovers and inhibition of non-interfering crossovers in heterozygous regions*”*. This is not correct. As discussed in detail for point (2) above, the changes in interfering and non-interfering crossovers are not specific to the* cis *effect: they are a general feature of the heterozygous cases analyzed (even assuming no general genetic background issues). Furthermore, the observation does not imply two mechanistically distinct effects as the above statement suggests. Rather, there is a change in the distribution of undifferentiated recombination intermediates into crossover versus non-crossover outcomes.*

These points are related to those discussed above. We agree with the reviewer that the *cis*-effect could act via the crossover/non-crossover decision mechanism and have added these ideas to the Discussion.

*D)* “*This reveals an unanticipated mechanism whereby DNA polymorphism can recruit crossovers, contribute to positive correlations between recombination and diversity and influence the action of selection.*” *For this reviewer, there is no positive correlation between recombination and diversity shown in this paper, as described in detail above*.

To address this we have added new data to Figure 1 showing a strong positive correlation between historical recombination and sequence diversity in *Arabidopsis*.

*The F*_*1*_
*hybrid data are not evidence. Hybrids can have higher or lower levels than a particular reference homozygote. A F*_*1*_
*hybrid strain shows a higher proportion of interfering crossovers among total crossovers but no difference in total crossover levels. And the one case in which crossover levels are increased in a heterozygous region is not shown to involve* “*recruitment*” *of crossovers to the heterozygous region from a homozygous region*.

We believe our *I3bc* analysis directly addresses this point and shows that heterozygous regions increase crossover frequency at the expense of linked homozygous regions (Figure 6).

*There is a* cis *effect in which presence of an adjacent homozygous region increases crossovers in the heterozygous test region, but this is an increase above the level seen in the heterozygous region in the pure F*_*1*_
*hybrid, so it does not represent a simple effect of DNA polymorphism. Finally, there is no mechanism revealed. There is the finding that F*_*1*_
*hybrids have altered crossover/non-crossover ratios, but this does not increase the overall level of crossovers, so the* “*mechanism*” *alluded to does not increase crossing-over*.

We have clearly demonstrated that crossover interference is strongly influenced by the presence of heterozygosity. We agree with the reviewer that the most likely mechanism would be for the *cis*-effect to act via crossover/non-crossover choice. However, further work will be required to prove which step of meiotic recombination is influenced.

*E) The Discussion culminates with the following point*:

*“We propose that detection of sequence mismatches occurs during strand invasion or heteroduplex/dHJ formation and differentially inflluences the activity of interfering and non-interfering recombination proteins coincident with crossover/non-crossover repair choice. Therefore, as interfering and non-interfering repair pathways compete for DSBs, their activities are simultaneously modulated by heterozygosity, causing the* cis *effect*.”

*There are several problems, touched on above, which converge here*.

*i) There is not really a competition between interfering and non-interfering pathways. There is a crossover designation process and the leftovers become mostly non-crossovers but occasionally become crossovers, and the level of those latter crossovers are increased by* fancm *(as the authors of the* fancm *study state)*.

We agree with this model and hope the manuscript and Discussion section now more accurately reflect these ideas.

*ii) Most importantly, the relevant effect does not cause the* cis *effect—it is seen in a pure* F_1_
*hybrid. The* cis *effect must come from something else*.

We agree that F_1_ crossover variation is likely to be caused by a combination of *cis* and *trans* effects. However, analysis of F_2_ and backcross populations has allowed us to demonstrate the *cis*-effect being caused by varying patterns of heterozygosity.

*iii) There is no reason to think that the crossover/non-crossover choice is made during dJH formation; all that can happen at this stage is that the process can be degraded to give fewer crossovers and more non-crossovers, which is the opposite of the effect observed here*.

We agree and favor a promotive effect of heterozygosity at an early meiotic stage, most likely coincident with ZMM designation. However, we also cannot rule out that mismatches in dHJs might also differentially influence resolvases. Therefore, to be inclusive of these ideas we would like to keep consideration of alternative mechanisms in the Discussion.

[Editors’ note: the author responses to the re-review follow.]

*Thank you for resubmitting your work entitled* “*Heterozygosity promotes interfering and inhibits non-interfering crossovers during* Arabidopsis *meiotic recombination*” *for further consideration at* eLife*. Your revised article has been favorably evaluated by Detlef Weigel (Senior editor) and three reviewers. The manuscript is improved, but there are still concerns whether your interpretation of the data goes too far. Therefore, we are asking you to rewrite the manuscript as much as possible to be a fair description of the unsuspected phenomena, without making too many claims regarding crossovers being attracted from one kind of region to another. In other words: please accommodate the reviewers' comments as much as possible. We realize that there are differences in interpretation of the data between you and specifically reviewer 2, but we felt that the phenomenon is important enough that it deserves prominent publication. One of our board members also stated that the* “*phenomenon is interesting, but the point is not to find a model (it does not add much to invoke the beam film model, since there is no way to know if it is late DSBs and/or mismatches, and the authors are certainly far from understanding the molecular mechanism) but to validate the general principle*.”

Thank you for this guidance. We included the model in response to peer-review evaluations from a prior version of the manuscript. We agree that presenting the model is not critical to the central thesis of the paper and have removed the figure.

*Detlef Weigel has made specific comments in the manuscript, as attached*.

We appreciate Detlef’s attention to detail and have made the changes that he suggested.

Reviewer #1:

*Over all I found the manuscript to be much improved in terms of it presentation of its results, and the addition of the new analyses made the findings a more general statement of the effect of heterozygosity on recombination patterns. I note, however, that the other reviewer's original concerns were much more substantive than mine. As such, I view their opinion as carrying more weight than mine in this appeal*.

*The last paragraph of the Discussion is problematic*:

“*We propose that the biological function of the heterozygosity cis-effect is to recruit crossovers to variable regions of the chromosomes, acting as a feed-forward mechanism to increase diversity*.”

“*… meiotic recombination has been selected to promote…*”

*Arguments about the evolution of recombination modulators are very slippery (as recombination unlinks the fate of the modifier and the recombinant haplotype it creates). As such these proposals are unsupported speculation, perhaps changing* “*propose” to* “*speculate*” *would help.* Arabidopsis thaliana *has not been a selfing lineage for very long (like most selfers), and the authors' argument does not seem super convincing evolutionary mechanism for outcrossers (as homozygosity runs will be broken up across generations, by segregation). So their explanation seems somewhat shaky*.

We acknowledge these uncertainties and have removed these points from the Discussion. As discussed below we have also included a statement on the relevance of different mating-systems for these phenomena.

*In outbred organisms very long blocks of homozygosity are rare. The authors should caution that they know little about how long a block of homozygosity is needed to promote this effect, so the importance of this effect in other systems (e.g. mammals) is unclear*.

This is an important point, which we have addressed by adding the following sentence to the Discussion where we consider the potential generality of our observations: “However, when assessing the significance of such effects it is also important to consider how outcrossing versus selfing will influence patterns of homozygosity and heterozygosity within species.”

*In general the authors have done a better job of acknowledging other likely contributors to the recombination/heterozygosity relationships. Except for*:

“*However, the cis-effect is unlikely to explain all of this relationship and genetic hitchhiking, background selection and recombination associated mutagenesis may play important roles*”.

*The contribution of these other effects has been subject to quantitative investigation for over a decade. While the authors’ findings are very interesting, it is a disservice to use* “*all*” *and* “*may*” *in this way. I'd say that the contribution of linked selection is much more established than their mechanism, and should be acknowledged as such*.

We have removed this sentence from the Discussion, though description of the important and well-established role linked selection plays in driving correlations between recombination and diversity remains in the Introduction, including references to several key publications.

Reviewer #2:

*Suggestion*:

*Title: Adjacent homozygous and heterozygous regions reciprocally enhance and suppress crossing-over in an interference-mediated process*.

*Summary: Analysis of meiosis in mosaically-hybrid* Arabidopsis *lines reveals that a heterozygous region suppresses crossing-over in an adjacent homozogous region while, reciprocally, the homozygous region increases crossing-over in the adjacent heterozygous region. This interplay requires crossover interference: it is absent in a* fancm zip4 *background where crossovers occur but interference is absent. Two new features specific to recombination in heterozygous regions are also revealed: an effective increase in crossover interference and a decreased effect of a* fancm *mutation, which normally increases crossovers that do not exhibit interference. Potential mechanisms and evolutionary implications are discussed*.

*Part I. Summary. The authors have identified an interesting phenomenon that takes place at the junctions between homozygous and heterozygous regions, particularly when one of the involved regions is sub-telomeric: the frequency of recombination (crossing-over) in the homozygous region goes down while the frequency of recombination in the heterozygous region goes up (relative to the fully homozygous and fully heterozygous cases, respectively). This phenomenon implies reciprocal interplay between the two regions. This phenomenon applies specifically to junction regions. Fully heterozygous regions exhibit almost the same recombination frequency as fully homozygous regions, at least in the situation examined here. This could be of genetic/evolutionary significance, although that would depend, particularly since the role of crossing-over for evolution is hotly debated. It is not 100% clear whether the phenomenon applies generally throughout the genome and/or why it is particularly prominent in sub-telomeric regions*.

We thank the reviewer for this suggestion and have changed the title to: “Juxtaposition of heterozygous and homozygous chromosomal regions during meiosis triggers reciprocal crossover remodeling via interference”. We have also extensively rewritten the manuscript to emphasize that we report a junction phenomenon caused by juxtaposition of heterozygous and homozygous regions. We believe sub-telomeric regions show this effect most prominently as interference is acting from one direction only, though we have also detected a weaker, yet significant, effect across the centromere of chromosome 3, where interference acts from both arms.

*The authors go on to suggest a specific mechanism for this phenomenon. One point is clear: this phenomenon requires crossover interference. In a mutant situation where there are crossovers, but no interference, the phenomenon is absent. This is interesting and sensible because crossover interference is, by its nature, a process in which adjacent regions communicate with one another*.

We are glad the reviewer agrees that our mutant analysis supports a role for crossover interference in mediating these heterozygosity/homozygosity junction effects. To provide further support for the role of interference we have added analysis of *I3bc* crossovers in a *zip4* (interference defective) single mutant with varying heterozygosity (Figure 8), which further supports a role for crossover interference in these phenomena.

*Beyond this point, however, the authors’ assertions regarding mechanism are not supported by the data*.

*1) The authors say that non-interfering COs are suppressed in heterozygous regions. This features prominently in the Title and Summary. This is not shown in this paper. What is shown is that the COs which occur in a* fancm *background (which do not exhibit interference) are reduced in heterozygous regions. There is no evidence that these COs are occurring by the same molecular mechanism as the* “*canonical non-interfering COs*” *that arise in wild type meiosis*.

We acknowledge that we do not formally know whether the non-interfering repair that predominates in *fancm* is biochemically equivalent to the non-interfering repair that occurs in wild type. We have now explicitly stated this uncertainty in the second paragraph of the Introduction. As discussed above we have also added analysis of *zip4* single mutants, where only non-interfering repair is operating. Our observations are again consistent with non-interfering repair being less effective in heterozygous regions. Specifically, *zip4* crossover frequency is significantly decreased by heterozygosity, which is not observed in wild type where interfering repair predominates (Figure 8).

*2) The authors show that interference, as classically defined, appears to be stronger in heterozygous regions than in homozygous regions. This is also an interesting observation, which is documented not only for the specific situation analyzed in detail, but more broadly. This is the first time that interference has been examined in heterozygous situations, as far as I know*.

We are glad the reviewer appreciates the novelty of these findings.

*However, further consideration of the implications of this finding have some problems: (a) the phenomenon of increased interference may not mean what the authors think it means; and (b) the authors wish to say that this increased interference is responsible for the interplay between heterozygous and homozygous regions. This is a possible model. But I think it is wrong and there is actually evidence in the paper against it*.

*The strongest direct argument against the authors’ model is that the reciprocal interplay between heterozygous and homozygous regions is observed in a situation where crossovers in the two regions exhibit the level of interference characteristic of the homozygous region (*Figures 6 and 8*; details below)*.

*More generally: it is clear that, at junctions, crossovers go down in homozygous regions and go up in heterozygous regions, relative to the pure homozygous or pure heterozygous cases, respectively, and that interference is required (above). The question is: what is the basis for this asymmetry? In my opinion, the underlying effect could be that there are more DSBs (or total inter-homolog interactions) in heterozygous regions. More DSBs means more COs, which means more interference emanating from that region across the border to the homozygous region which means fewer COs in the homozygous region than in the pure homozygous case. Oppositely, fewer DSBs in homozygous regions means fewer COs which means less interference emanating from than region across the border to the heterozygous region which means fewer COs in the heterozygous region than would be observed in the pure heterozygous case*.

*This model is further supported by two interrelated considerations. (1) Fully heterozygous regions show the same level of COs as fully homozygous regions (*Figures 7 and 8
*wild type cases) even though there is more interference. Clearly there must be some other effect that counterbalances the apparent increase in interference. (2) The apparent increase in interference could, in principle, reflect either of two effects (Zhang et al., 2014 PLoS Genetics and PNAS): (a) an increase in the distance over which interference spreads; or (b) a decrease in the* “*strength*” *of CO-designation. The first would imply a more robust process in the heterozygous case, which seems peculiar a priori; the second would imply a less robust process in the heterozygous case, which makes more sense. (3) If phenomenologically increased CO interference reflects model (b), then the level of COs can be restored by increasing the number of DSBs*.

*By my alternative model, the phenomenological increase in interference in heterozygous regions is actually irrelevant (as shown by the data mentioned above). The real effect would be more DSBs*.

*Another problem with the authors' model is that it requires that the* “*increased interference in a heterozygous region*” *spreads across the boundary into the homozygous region and that the* “*decreased interference in a homozygous region*” *spreads across the boundary into the heterozygous region. This is a priori unlikely if the basis for the altered interference is heterozygosity per se*.

*It would actually not be so difficult to provide at least some evidence for more DSBs/interactions in heterozygous regions (e.g. by Dmc1 focus analysis). However, this opens up another can of worms and is beyond the scope of the current presentation*.

*The basic point is that it is not proven that there is a role for increased interference in heterozygous regions in the observed junction phenomenon and there seems to be evidence against it*.

We thank the reviewer for these considerations of the potential mechanistic basis for the effects of heterozygosity-homozygosity juxtaposition. Although we have limited data to distinguish between these potential models we have added consideration of them to the Discussion, including the idea that heterozygous regions could receive higher DSB numbers via feedback signaling. Specifically, we have considered the phenomenon we observe in the context of the Beam-Film model for crossover interference as suggested (Zhang et al., 2014, PLoS Genetics).

*Importantly also: the documentation, description and presentation in this paper requires significant improvement as described below. There are problems of logic, vocabulary, data, controls, explanation and interpretation. This paper is not really readable by a general audience in its present form*.

*There is redundancy in the comments below, for which I apologize, but hopefully this is useful*.

We apologize for these issues, which we have attempted to remedy as described in detail below.

*Part II. Specific Issues with Title and Summary*.

*The title says:* “*Heterozygosity promotes interfering and inhibits non-interfering crossovers during* Arabidopsis *meiotic recombination*”*. I do not think that either of these conclusions is warranted*.

*(i) The latter conclusion, that heterozygosity inhibits non-interfering crossovers, is wrong because it is based on the assumption that the extra COs that arise in a* fancm *mutant (and do not exhibit interference) are biochemically the same as the extra COs that arise in wild type (and are defined as* “*non-interfering COs*”*). Heterozygosity does decrease the COs seen in a* fancm *mutant (*Figures 7 and 8*), but this cannot be extrapolated to wild type*.

*(ii) Heterozygosity does, phenomenologically, increase CO interference. This is an interesting observation. But the interpretations placed on it are not proven and, in my opinion, are not correct. This finding could perhaps be interpreted as* “*promoting the interfering CO pathway*” *(as in the Title). However: (a) In comparisons between fully homozygous and fully heterozygous regions, the frequency of COs is essentially the same (*Figures 7 and 8*). This cannot be explained by an increase in CO interference alone. It implies some other effect. (b) The observation of increased CO interference by CoC analysis can be explained either by an increase in the* “*spreading distance*” *of the interference signal or by a decrease in the strength of CO-designation; both effects decrease the CoC at short inter-interval distances (See Zhang et al PLoS Genetics 2014 and PNAS 2014). If the relevant effect is to decrease the strength of CO-designation the observed effect is actually not promoting the interfering CO pathway but making it worse. (c) The fact that CO number does not change in a fully heterozygous region even though interference effectively increases (a, above) suggests/implies the existence of another effect, which could/should be upstream of any recombination fate decision. Most simply: if there is a decrease in the efficiency of CO-designation, the resultant decrease in COs can be overcome by an increase in the frequency of DSBs (or DSB-mediated pre-CO interactions). Perhaps mismatched chromatin structure has such an effect, maybe especially in sub-telomeric regions which might be influenced by pre-DSB homologous pairing interactions*.

As described above we have changed the title of the paper and acknowledged the uncertainty concerning the biochemical basis of non-interfering repair in wild type versus mutant backgrounds. We have also elaborated on the potential mechanistic basis of the effects we observe in the Discussion.

*The end of the Summary says: “Using recombinant populations we show that heterozygous regions attract crossovers from homozygous regions on the same chromosome at the megabase scale. Interference inhibits formation of adjacent crossovers over similar physical scales, and we demonstrate that this polymorphism cis-effect is dually mediated by promotion of interfering crossovers and inhibition of non-interfering crossovers in heterozygous regions. This reveals an unanticipated mechanism whereby DNA polymorphism can recruit crossovers, contribute to positive correlations between recombination and diversity and influence the action of selection*.*”*

*I do not agree with any of these statements*.

We acknowledge this and have removed these statements from the Summary.

*Let us first define* “*polymorphism* cis *effect*”*. What was analyzed were constructs in which recombination was assayed in a sub-telomeric test region, which could be homozygous or heterozygous, and had an adjacent internal region that could be homozygous or heterozygous. Assuming experimental problems (below) can be ignored, what is observed (assayed region underlined) is, essentially, the following: (a) the frequency of COs is lower in HOM-HET than in HOM-HOM; and (b) the frequency of COs in HET-HOM is higher than in HET-HET. That is, an adjacent heterozygous region decreases COs in a homozygous region relative to what would have been seen in a fully homozygous situation while an adjacent homozygous region increases COs in a heterozygous region relative to what would have been seen in a fully heterozygous situation.* But*… HOM-HOM and HET-HET both show the same frequency of COs as one another (see Figures 6CD, 7 and 8 for the clearest examples). [There is one exception (*Figure 8*; see below) which I will simply ignore]. This is not:* “*heterozygous regions attract crossovers from homozygous regions on the same chromosome at the megabase scale*”*. There is no attraction; this is not a zero-sum game; the effect is reciprocal, not unidirectional; and stating the result in this way is not accurate and is prejudicial to thinking.* “*Recruitment*” *to heterozygous regions is simply wrong*.

*So what is the explanation for the observed effects? The paper also shows (*Figures 7 and 8*) that this interplay is not observed in a* fancm zip4 *double mutant where there are still COs, but of a variety that does not set up interference. Fine. Interference is required. And if everything were the same in HOM and HET regions, there would be no effect of having one next to the other, so just saying that there is interference is not enough. There has to be some asymmetry. The authors seem to think that the difference is that interference is* “*stronger*” *in HET regions and* “*weaker*” *in HOM regions. If this were true, and if that effect crossed the border between the two regions, COs in HET regions would tend to decrease COs in HOM regions relative to the effect from normal HOM interference; oppositely, HET regions will experience less interference from an adjacent HOM region than from an adjacent HET region. However: I do not think that this is correct. To reiterate some of the points made above*:

*(A) This model assumes that interference specific to HOM or HET can cross a HET/HOM border. This is a priori unlikely. Furthermore, there is data in the paper, which says that you can see the diagnostic reciprocal effects on CO levels in three-factor crosses where the HOM-type CO interference level is observed (*Figures 6 and 8*; see below)*.

(B) An alternative explanation is the same one required to explain interference patterns above (and thus is more likely): more DSBs in HET regions vs HOM regions. More DSBs in HET regions will imply more COs. More COs means more interference signals, which means fewer COs in the adjacent region. Oppositely, in HOM regions, fewer COs means fewer interference signals which implies more COs in the adjacent region.

*To return to the Summary:* “*we demonstrate that this polymorphism cis-effect is dually mediated by promotion of interfering crossovers and inhibition of non-interfering crossovers in heterozygous regions. This reveals an unanticipated mechanism whereby DNA polymorphism can recruit crossovers, contribute to positive correlations between recombination and diversity and influence the action of selection.*” *Inhibition of non-interfering COs is not shown (above). Enhanced interference in heterozygous regions (which is apparently what the authors mean by* “*promotion of interfering COs*”*) is not relevant. The likely relevant effect (more DSBs and thus more COs in HET regions) is not mentioned or discussed as a possibility. Thus: there is, therefore, no* “*unanticipated mechanism*”*, and no effect in which* “*DNA polymorphosms recruit COs and thus increasing genetic diversity in hybrid situations*”*.*

*Instead: what has been shown? There is a phenomenon that is observed when HOM and HET regions are side-by-side. A* “*junction*” *phenomenon, if you will. There is a reciprocal interplay in which COs are decreased in HOM regions and increased in HET regions and this interplay requires interference. This is likely correct, although there are a lot of technical issues pertaining to the data (below). NB that this is specific to adjacent HET/HOM regions, because HOM/HOM and HET/HET recombination are very similar. Thus, this is not a general effect of global heterozygosity, but an effect specific to juxtaposed HET/HOM regions (junctions). Thus any evolutionary implications must derive from this specific situation, not from heterozygosity in general.*

*The basis for this effect, about which the authors are quite specific in the title and the Summary, is probably not what the authors think it is, as described above. To repeat (again): There is no data on interplay between HET regions and* “*non-interfering COs*” *in wild type, only in a mutant situation that might be different. The observed increase in interference is real and per se interesting, but this does not imply a more robust interfering CO pathway; it could as easily and more probably imply a weaker pathway. And regardless of that point, the increase in interference in HET regions is not responsible for the* “*junction interplay*”*, as described above. This leaves the actual basis for the asymmetry between the two types of regions to be determined, but increased DSBs is an attractive possibility, which is not considered (described above).*

It is also notable that most of the data come from analysis of a sub-telomeric interval(s) which could be special for any number of reasons. This maybe ok, but should be discussed more.

We acknowledge the issues raised by the reviewer relating to the interpretation of our data. To address this we have removed mention of the words ‘attract’ and ‘recruit’ from the manuscript in reference to the effects we observe. We now express these ideas explicitly in terms of juxtaposition of heterozygous and homozygous regions and that the reciprocal changes in crossover frequency are mediated via interference. We further explore possible causes in the Discussion, including increases in DSBs in heterozygous regions.

*The above considerations address the paper on the assumption that all data are valid and conclusions fully supported by the observations. But there are quite a few issues that need to be addressed before one is really sure that these criteria are met. These are discussed below. Maybe I'm just slow. But for me, this paper was extremely difficult to read and understand. This is, in part, because the underlying effects and issue are complicated. But in addition, there are a variety of problems with presentation and analysis and interpretation and logic/assumptions. Important experimental details are absent or buried or written* “*in plant language*”*. There is a mixture of ideas and experiments. There is a historical/narrative presentation, rather than a consideration of the data per se irrespective of how the authors came to some ideas. This latter feature makes it extremely difficult to think about what could be going on.*

We again apologise for these issues in the clarity of our presentation, which we have addressed as described below.

Part III: More comments of various types.

*1) It is a well-established fact that heterozygous lines may have higher or lower recombination rates than (more) homozygous lines. This is reiterated by the authors, in considerable detail. As the authors state, these could reflect differences in the natures of the diffusible molecules produced, which the authors call* “trans” *effects.*

*There is a major vocabulary problem. In opposition to trans, there is cis. There a major confusion with regard to the word* “cis”*, which is used in two different ways at different places in the paper. The most general way is in opposition to* “*trans*” *as defined above. That is: the effects of heterozygosity per se, irrespective of differences in diffusible factors. However, the word cis can/is also used to refer to the effect of the nature of one region on events in an adjacent region on the same bivalent, that is an effect that is in cis along a given chromosome. The two different uses make reading this paper really difficult. For example:* “*dual* cis *effects*” *probably uses the word* “cis” *in the first sense; and* “cis *effects of heterozygosity*” *probably uses the word in the second sense.*

We acknowledge the confusion caused by our previous use of *‘cis’* and *‘trans’*. To clarify this we have explicitly stated our distinction between *cis* and *trans* modification of recombination in the Introduction. “We define *trans* modifiers as loci encoding diffusible molecules that control recombination on other chromosomes, and elsewhere on the same chromosome, as exemplified by mammalian *PRDM9* (5; 11; 39; 65; 83; 88; 93). We define *cis* modification as variation that influences recombination only on the same chromosome, for example, the inhibitory effects of high SNP density, inversions and translocations (6; 15; 27; 33; 60; 95; 106).” We therefore restrict later usage of these terms within the context of these definitions. We have also removed many previous uses of the word ‘*cis’* and instead described the observed phenomenon in terms of the juxtaposition of heterozygous and homozygous regions.

*2) There is additionally what appears to be a fundamental experimental problem. The entire paper (after*
Figure 4*) presents the* “*genotypes*” *of tester bivalents, and analyzes events on those bivalents, without ever discussing anywhere that I could find in the text what is known about the other chromosomes in those same cells and whether variations in* trans*-acting factors might be relevant or controlling for such variations in any way. The problem begins with*
Figure 4
*and its relatives. It is stated that the pattern described in that figure, which involves the presence of homo/heterozygosity adjacent to a heterozygous test interval, is independent of the nature of the homozygous region. But this is not correct. Genotypes at the top of the pattern tend to have Col/Col (red) while genotypes at the bottom tend to have Ct/Ct (green). This is seen also in the other analyzed sub-telomeric interval (*Figure 4*). This should be specifically investigated. More generally, there has to be some attempt to deal with this as a general issue for all experiments. Perhaps this was actually done by analyzing multiple lines with the same critical genotype but, necessarily, different complements of information on other chromosomes. Are these the multiple data points in various figures? If so, it is just not obvious in the text of the paper or the figures or figure legends and the reader should not have to go searching for it. If not, this issue remains as an underlying problem throughout the paper. There are arguments against* trans *effects as a big issue in several places, but they are never made in the paper.*

*Summary: this issue has to be specifically addressed for*
Figure 4
*in order to draw the conclusion in that figure and it needs to be discussed as an issue in all other cases. There is the always-underlying issue of possible* trans *variations; however, it can be argued that a consistent picture emerges in several types of constructs that should (apparently?) have different types of* trans *effects.*

This is an important point and we present several lines of evidence that argues against the presence of significant *trans* modifiers (as defined above) of recombination segregating in the Col x Ct recombinant populations. First in Figure 4 we present QTL analysis for *420* recombination rate in a Col x Ct F_2_ population that shows an absence of significant associations on the chromosomes other than the one being measured (chromosome 3). To extend this we performed additional QTL analysis for *I2f* recombination rate in an independent Col x Ct F_2_ population (Figure 4). Here again a significant association was only detected on chromosome 2 where recombination is measured, and not on chromosome 3. This clearly demonstrates that lack of reciprocal *trans* effects between chromosomes 2 and 3 in these crosses. To further mitigate against potential weak *trans* modification effects from other chromosomes in experiments where we derive recombinant genotypes for analysis, e.g. HET-HOM (Figures 5, 6 and 8), at least 3 independently derived lines were analysed for each condition (apart from HOM-HET *420-CEN3* where two were analysed). Generation of independent lines means that they will be Col/Ct mosaic throughout the genome in different ways in each case, minimizing the confounding effect of any potential *trans* modifiers. Further detail has been provided on generation of these lines in the Materials and methods and in Figure 7–figure supplement 1 and Figure 8—figure supplement 1 where crossing schemes are diagrammed.

*3) The paper makes many comparisons among different regions without much emphasis on the fact that things are different or why. Of course part of the difference between the CEN interval and the 420 interval is the location; but also there is the fact that CEN is measured in male meiosis where in* Arabidopsis*, there are more COs, and they are more sub-telomeric. 420, which is sub-telomeric, is measured in female meiosis, where CO rates are generally lower and COs tend to be LESS sub-telomeric. It is frustrating that the authors did not discuss how these differences, particularly the M/F differences, might affect their results and/or interpretation. To the only partly initiated, it seems like apples and oranges.*

The reviewer raises valid points concerning regional variation in crossover patterns along chromosomes, and also variation between male and female meiosis. A major difference between *420* and *CEN3* is that the former is located in gene-rich euchromatin and the later located in repeat-rich heterochromatin. This is important, as we have previously shown distinct effects in these intervals when epigenetic information (DNA methylation) is altered ([114], PLoS Genetics). This is also reflected in the lower recombination rate observed in *CEN3* relative to *420*, consistent with general recombination suppression observed in heterochromatic regions. As we observe that these regions show different responses to recombination mutants (*fancm* and *fancm zip4*, Figure 6) it is likely that this reflects an interaction with chromatin. These differences are discussed in the Results section ‘Heterozygosity extensively modifies crossover frequency in *Arabidopsis’*.

The reviewer also raises the pronounced differences observed in *Arabidopsis* sub-telomeric regions between male and female meiosis ([45], PLoS Genetics). They are correct that our analysis of the sub-telomeric interval *420* represents a mixture of male and female meiosis, due to collection of seeds from self-fertilized plants. However, we complement this with analysis of the sub-telomeric *I2f* and *I3bc* intervals, which are analysed in pollen and therefore measure exclusively male meiosis. Importantly, the conclusions reached from *420* versus *I2f/I3bc* are concordant, indicating that these sex differences are not a major problem for our analysis.

*4) At the most general level, this paper is among the first, in a plant system, to consider* cis *effects in either sense defined above. In the first category of cis, there is a long history of studies on the effects of basepair mismatches on recombination, both biochemically (they are sensed by RecA and RecA homologs, in combination with mismatch repair proteins, such that the ongoing strand exchange is rejected) and genetically, where recognition of mismatches by the mismatch repair system is known, in meiosis, to specifically eliminate crossovers (e.g. Hunter and Borts). I actually don't know whether anyone has examined DSB levels in heterozygous regions. I am not aware of any previous study that examines either interference or the effects of* fancm*-type or similar mutations in heterozygous regions. I am also not aware of any other studies that address* cis *effects in the second sense. So, as phenomena, the reported observations will be of interest to the meiosis field.*

We are pleased that the reviewer appreciates the novelty of our study and its interest to the meiosis field. We have added further discussion of the known interactions between recombination and base-pair mismatches to the Discussion.

5) To set a baseline, it is easiest to begin with consideration of general properties of heterozygous regions. The authors make two new findings.

*A)* Fancm *mutations have different effects on CO frequency in heterozygous versus homozygous regions. However: the interpretation of this finding in the paper is incorrect. The authors assume that the COs that emerge specifically in a* fancm *mutant are going by the same* “*non-interfering crossover pathway*” *that is argued to exist at a low level in wild type cells. If this were true, then effects seen in* fancm *mutant backgrounds would also be occurring in wild type. As stated above, this is possible, but it is not known and is not shown by this paper.*

*More specifically: is generally agreed (e.g. by Mercier and supported indirectly by yeast studies) that there is a set of DSB-initiated intermediates, among which a subset are specified to be* “*interfering COs*”*, ie* Zip4-*dependent COs that show classical interference. The remaining interactions are then matured to other fates, mostly non-crossovers and inter-sister events but also, at a low level to COs. These are the* “*non-interfering COs*”*. In a* fancm *mutant, the fates of the interactions that are not specified to be* “*interfering COs*” *are different, with a much larger proportion becoming COs than in the wild type case. The reason for this is unclear, but presumably is some modulation of biochemistry. What is shown in this paper is that, in a* fancm *mutant, the level of* “*extra*” *COs that arise in heterozygous regions is less than in homozygous regions. This does not speak to what might or might not be going on in wild type.*

*Another problem with this analysis is that wild type,* fancm *and* fancm zip4 *strains are analyzed but* zip4 *single mutant strains are not. This is not acceptable for a properly controlled analysis, although I understand that it did not seem* “*relevant*” *in the context of what the authors were trying to address. Moreover, the missing mutant might actually be directly informative about what is going on in a wild type background.*

As discussed earlier we have now acknowledged the uncertainty of whether non-interfering repair occurs via the same biochemical pathway in wild type and *fancm* backgrounds in the Introduction. We have also added *zip4* single mutant data, which is further consistent with our model (Figure 8). Specifically, that crossover frequency in *zip4* is further decreased by heterozygosity, indicating that non-interfering repair is less efficient in this situation.

B) The authors find that heterozygous regions show higher levels of (phenomenological) CO interference than homozygous regions. This is shown by a three-factor cross for one sub-telomeric region for Col/Ct heterozygosity and by a different method for a broader range of heterozygous lines. The latter analysis suggests that it is general.

This is an interesting and novel finding. It implies that, as an observational fact, within a heterozygous region, if there is a CO in one region, the probability that a CO will occur nearby is reduced even more than in the normal homozygous case.

*[As summarized above: there are two ways to think about what this means in reality. The* “*normal*” *way might be to think that the interference signal spreads out for a longer distance in the heterozygote. However, there is another way to think about it. The same result can be obtained if there is a reduced propensity for a recombinational interaction to be specified to be a crossover. This is shown by the* “*beam-film*” *model (or any scenario involving designation of a CO and spreading interference) by shifts in CoC curves to the left or to the right according to variations in CO-desigation probability, with a constant* “*interference distance*”*.] This is an intriguing finding per se. Its significance for other phenomena is more complicated, as discussed above.*

We are pleased the reviewer finds our results interesting and novel. We have added further detail to the Discussion to take into account the proposed ideas for the potential cause of these effects.

*6) The authors spend a lot of time discussing situations in which they believe that events in a heterozygous region are influenced by events in a homozygous region, and/or vice versa. First, comments on*
Figures 4, 5 and 6*:*

Figure 4*. The left part shows that a pure homozygous line, heterozygous for the red and green tester markers, is produced by crossing a marked Col/Col line with an unmarked accession line (top) and then that when this line is selfed to give an F*_*2*_*, various types of progeny can result.*

*Comment. The authors show only cases in which the R and G markers are in cis, which implies that during selfing, there has been no crossing-over between them. This may be the majority of outcomes. However, the criterion for selection of F*_*2*_
*progeny to analyze was heterozygosity for R and G. By fluorescence, this implies that there is one copy of each. For all of the examples shown, R and G are* “*in cis*” *to one another. But 15% of meioses will give a crossover between the two markers; and if the appropriate products unite (R+G- with R-G+), the resulting F*_*2*_
*seed will ALSO have only one copy of each marker. This may be rare (15% x 15%). Nonetheless, such cases would be scored as having the highest recombination rates if they were not somehow identified and removed from further analyses. Do the authors know from genotyping whether their F*_*2*_*s include such cases and if so, which ones are they among the 139?*

The reviewer is correct that ‘*trans’ R+/+G 420* lines arise at the expected frequencies. In the seed of these individuals the fluorescent classes representing parental and crossover genotypes are reversed. As *R+*/+G plants also have variable heterozygosity/homozygosity patterns within *420* depending on crossover positions in the previous generation we excluded these plants from further analysis. To clarify this point we have added additional text to Materials and Methods and we present analysis of *RG/++* and *R+/+G* seed in Figure 2—figure supplement 1. As this shows it is straightforward to identify these individuals and exclude them from further analysis.

*Other* “*missing information*”*: (i) Were the F*_*2*_*'s obtained from a single F*_*1*_*? Or from multiple F*_*1*_*'s with results pooled? (ii) Is it completely obvious that the fluorescence detection could distinguish RRG and RGG from RG genotypes? This is clearly a critical point and it does not seem to be addressed explicitly, though I might have missed it.*

i) As discussed earlier ∼3 independent lines were analysed for recombinant genotypes to minimize potential confounding effects from other chromosomes, although it is important to restate that our QTL analysis did not detect such loci in Col x Ct populations. ii) Although it is possible to distinguish seed with one versus two copies of fluorescent transgenes, there is a greater degree of overlap between classes than between fluorescent and non-fluorescent seed (Figure 2). Therefore we only use divisions between fluorescent and non-fluorescent seed to calculate genetic distance. This is possible using the formula: cM = 100 × {1 – [1-2(*N*_*G*_+*N*_*R*_)/*N*_*T*_] ^½^}. Where *N*_*G*_ is a number of green only fluorescent seeds, *N*_*R*_ is a number of red only fluorescent seed and *N*_*T*_ is the total number of seeds counted. This information is included in the Materials and methods section.

*Minor presentation issue: the meaning of the box on the right is obscure without working at figuring it out. Apparently it means that when the F*_*2*_*s are taken through meiosis, they may generate recombinants, but this is hardly obvious in the cartoon or the legend. (Again assuming that they started out with the cis configuration.)*

We have modified this diagram to make its meaning easier to comprehend (Figure 4).

Figure 4*. In this figure, the authors select a set of F*_*2*_*s that are heterozygous at the R and G markers. They then ask, at each of 51 other positions, whether the frequencies of the two parental alleles is different from what you expect from an absence of correlation with the selected* “*phenotype*”*. What is detected is a deviation from expectation for chromosome 3 (the chromosome on which phenotype is selected).*

Comment: We can start with the fact that this analysis is essentially not described at all in the paper. The non-plant reader would have no idea what is going on from the half a sentence in the text.

We have added the following description to the text to explain the statistical test we used here: “For each marker we used the heterozygous and homozygous counts in the hottest quartile versus the coldest quartile to construct 2x2 contingency tables and performed chi-square tests, followed by FDR correction for multiple testing (Table 5).”

*If I have understood correctly, this analysis appears to have a fundamental flaw. If you select for F*_*2*_*'s that are heterozygous for R and G, most of these will have the cis configuration, meaning that they have not undergone a CO between R and G on that chromosome in the previous meiosis. Since there must be at least one CO, there will be a CO somewhere else on chromosome 3. And there will be a non-random tendency for marker disposition to occur just because there is already a selection that restricts localization on this chromosome specifically. More specifically: since the average per chromosome is just a little above one, and since the CO is not sub-telomeric at the marked end, it will tend to occur at non-random position (or positions if there are 2 COs) away from the marked end, e.g. towards the middle of the chromosome. This effect explains the peak in the graph which is centered in the middle of chromosome 3 - it reflects the non-random consequences of having the CO in a non-random position along the marked chromosome. If this interpretation is correct: this data do not provide evidence for a* cis *effect. There is only the fact that you have pre-selected a population that, by its nature, will have a non-random arrangement of markers along chromosome 3.*

*Two further points: (i) 139 is a very small number of F*_*2*_*s from which to draw a conclusion. (ii) There should be a control showing that if you look at the entire population, you do not see any peak. Of course, given the basis for the peak (above), the control will show that it is specific to the selected sub-population, which is not really any help, but still, if such analysis is to be shown, the control must be there.*

We agree with the reviewer that our selection scheme imposes a significant bias on the location of crossovers within the F_2_ population on the selected chromosome. Due to our selection strategy for ‘*cis’* configuration *RG/++* plants, crossovers are excluded within the measured interval. As a consequence of this there will be a bias in the location of additional crossovers elsewhere on the selected chromosome, both due to this selection and due to inherent variation in the chance of recombination between chromosomal regions. However, this alone cannot explain the observed association between patterns of heterozygosity on the selected chromosome and recombination rate within the measured interval. Furthermore, we independently confirm this effect using independent experiments in Figures 5, 6 and 8 where we use defined recombinant lines instead of F_2_ populations.

Figure 4*.*
Figure 4
*shows the recombination frequencies in the 420 interval observed for the 139 F*_*2*_*'s. The conclusion is that the range is greater than for the Col/Col wild type strain or for a Col/Ct F*_*1*_
*full genome heterozygote. This is likely true, but it is hardly surprising under any model. It likely reflects* trans *effects as well as* cis *effects. It is unclear why this is valuable information, unless it sets the stage for later panels.*

As discussed above we did not detect significant *trans* effects via QTL analysis in these populations and our use of independently derived recombinant lines in later experiments will mitigate against any undetected weak effects from other chromosomes.

*Minor point: it is not appropriate to compare the rank plot for the 139 F*_*2*_*s with averages and ranges for Col/Col and Col/Ct. The ranges for the two compared lines are a statistical measure; they do not give the same impression as the rank plots. Thus one is comparing* “*apples and oranges*”*. The rank plots for the two comparison lines should also be shown, and then the averages and ranges for all three sample sets should be compared.*

We have now presented the data for the F_2_ and control Col/Col and Col/Ct lines identically in these figures.

Figure 4*;*
Figure 5*.*
Figure 4
*displays the natures of the 139 F*_*2*_*s, shown from top to bottom in rank order of recombination frequency in the 420 interval, with respect to whether the markers indicated on the X-axis are heterozygous (black), homozygous Col, or homozygous Ct. This plot shows a gradient of* “*black*”*, emanating from the test interval towards the end of the chromosome, to greater and greater distances from bottom to top. The authors interpret this data as follows:* “*However, we detected a significant association on chromosome 3 itself, where the hottest F*_*2*_
*quartile had significantly higher homozygosity outside 420, compared to the coldest quartile. This cis-effect was observed when the rest of the chromosome was either Col or Ct homozygous, indicating that it is non-allele specific and caused by polymorphism per se, rather than by specific sequence variants.*”

*The restatement of the result above does not accurately describe the entire picture. The fact that the black portion emanates from the selected region is not mentioned (and is presumably significant). An accurate restatement of the result, which admits to all interpretations, is that F*_*2*_*s with higher 420 recombination tend to be homozygous for flanking material while F*_*2*_*s with lower 420 recombination tend to be heterozygous. Moreover, the* “*extra heterozygosity*” *towards the bottom of the rank plot tends to occur specifically adjacent to the 420 region. The term* “cis *effect*” *should not be used, as this is an interpretation. It has become lab jargon for the phenomenon the authors think is going on, but it is confusing given the various versions of this phenomenon (including its reciprocity), as well as the fact that there should be a formal description of the data, not an interpretive one. Better to call it a* “*junction phenomenon*” *or Hom-Het interplay.*

As described above we have now more clearly described the statistical test used to detect this association in the Results section. We have also rewritten the text to describe the observed phenomenon in terms of heterozygosity/homozygosity junction effects and avoided describing this as a ‘*cis* effect’.

*More importantly, contrary to what the authors say, there is a clear tendency for more* “*red*” *at the top and more* “*green*” *at the bottom, not only in*
Figure 4
*but in the corresponding figure for the other sub-telomeric region in*
Figure 4*. This would likely appear in an appropriate plot, although a larger data set may be required to see it with very strong statistical significance. It suggests that having more Col information promotes a higher frequency of COs in general versus more Ct information. This is directly supported by the fact that, overall, Col/Col recombination rates are higher than Col/Ct recombination rates genome-wide (*Figure 4*). This has to be dealt with.*

Although this association on chromosome 3 would be consistent with the presence of a weak ‘*trans’* modifier in this region, when we performed a QTL analysis for the *I2f* interval on chromosome 2 using an F_2_ population derived from the same parents no significant effect was detected on chromosome 3 (Figure 4). This provides evidence that this weak association is not caused by the presence of a general *trans* modifier on chromosome 3.

*Also importantly: the authors cannot, in principle, say that they see a* cis *effect unless they do a control to show that the sample subset of 139 F*_*2*_*s have similar frequencies of recombination throughout the genome and that there are or are not any associations of recombination with homozygosity, and/or Col vs Ct content on a genome-wide basis. I appreciate that this is not trivial. But the authors' hoped-for conclusion is not trivial either. It cannot be made from this figure without more supporting (or contradicting) data.*

This is addressed directly by our QTL analysis in Figure 4, which did not detect significant *trans* modification effects.

Secondarily (perhaps): there are general uncertainties.

*i) There is the possible problem that the hottest F*_*2*_*s might have their markers in trans, with unknown consequences.*

We have addressed this point earlier (see Figure 2—figure supplement 1).

*ii) The top-ranked F*_*2*_
*in the 4D data set is very pecuIiar. It appears to be Col for the entire length of one chromosome and to have had a single CO exactly at the border of the marked region in the other. How is this possible?*

As this is a gene-rich euchromatic region it shows relatively high levels of recombination (e.g. [91], Heredity) and therefore it is not unlikely that we would recover crossovers in this interval.

*iii) As in*
Figure 4*, there could be unknown effects from selecting the subset of F*_*2*_*s that did not have a CO in the sub-telomeric region. Are there markers all the way to the end of the chromosomes or are sub-telomeric crossovers going to be missed (which is a problem for the analysis since many COs will be there specifically in the selected population, thus biasing the sample. There is actually a hint of such a problem in the fact that the observed* “*effect*” *is absent for the centromere (a central marker) and weak for an interstitial marker. One interpretation is that observation of an effect is a consequence of selecting a sub-telomeric region for analysis (rather than being an actual phenomenon is specific to such regions). The authors have only presented data saying whether each F*_*2*_
*is homozygous or heterozygous for the markers tested. But in principle, they could figure out the exact arrays of markers on the two chromosomes of the F*_*2*_
*and thus were crossovers occurred in the preceding F*_*1*_
*cross(es). While perhaps difficult, this might reveal additional informative information to further guide interpretation.*

*Overall, data in*
Figures 7 and 8
*tend to ameliorate concerns as to the validity of the authors' conclusions from*
Figures 4 and 5*. But the concerns are still real and valid for presentation of this data.*

For *420* the first T-DNA is located 256,516 bp from the telomere, for *I3bc* the first T-DNA is located 498,916 bp from the telomere and for *I2f* the last T-DNA is located 741,196 bp from the chromosome end. Based on chromosome average recombination rates these terminal intervals would be estimated to have genetic distances of ∼0.95, ∼1.85 and ∼2.41 cM. Hence, crossovers would be expected in these regions but at relatively low overall frequency. For the largest interval between *I2f* and the end of the chromosome we included an additional marker in our analysis at 19,311,521 bp and observed a small number of crossovers (2) (Figure 4 and Table 6). Importantly, these crossovers resulted in F_2_ individuals with heterozygosity between *I2f* and the telomere that showed high crossover frequency, further consistent with juxtaposition of heterozygosity and homozygosity increasing recombination.

Figure 6*. Panels A-C show that two lines that are heterozygous for 420 and Ct/Ct for all or most of rest of the chromosome exhibit higher 420-region recombination than a line that is homozygous Col/Col in 420 and heterozygous for the rest of the chromosome. The authors think that the relevant difference is that heterozygosity adjacent to homozygous 420 depresses recombination in 420. They use this data to argue that the presence of homozygosity adjacent to heterozygous 420* “*increases*” *recombination, and thus* “*eliminates the* cis *effect*”*.*

*First: this is not* “*eliminating the* cis *effect*”*. It is a different type of* cis *effect. It is not clear that the authors appreciate that they are seeing reciprocal effects. They are totally focused on the effects of heterozygous regions on homozygous regions.*

In any case, without controls, there are several hypotheses that are equally consistent with the data. What is the relevant difference(s)? Heterozygosity in 420? Col/Col in 420? Heterozygosity or Ct/Ct in the rest of the chromosome? Some combination of the two? There is no way to know without more lines with different genotypes. What is needed (minimally) is Heterozygous 420 plus homozygous Col/Col and fully homozygous Col/Col. (Homozygous Ct/Ct is desirable but difficult to construct). No conclusion can be drawn without controls. This is just not acceptable. To a first approximation, this data should be omitted.

*And to repeat: it is assumed that the only relevant differences among the different lines is on the chromosome of interest. But all of the chromosomes are different in the different strains. Is there evidence that the variations are not due to* trans *effects.*

*Another thing that is very unclear is why these particular lines were picked for testing. Both of the HL lines chosen for 6 A/B seem to be the rare type in the top quartile in*
Figure 4*; most of the lines in this quartile are homozogous Col/Col outside of 420. Why were these chosen rather than the more common type; or, better, why not test both types? And specifically, exactly which of the 139 F*_*2*_*s are HL1 and HL2? It is not obvious.*

We acknowledge the complications of these backcrossing experiments and have therefore removed them from the final version of the manuscript. We have added additional *I3bc* experiments (Figure 5), which address these issues directly and demonstrate that juxtaposition of heterozygous and homozygous regions causes reciprocal increases in recombination in the heterozygous region and reduces them in the homozygous region (Figure 5). We further show this effect is mediated via interference via analysis in *zip4, fancm* and *fancm zip4* mutant backgrounds (Figures 6 and 8).

Figure 6*. This figure shows a 3-factor cross in which there are two intervals that comprise most of the single 420 interval in (AB). (DE) shows that HOM adjacent to HET results in an increase COs in the HET region and that HET adjacent to HOM results in a decrease in COs in the HOM region. This points to the reciprocal effect shown further in*
Figures 7 and 8*. (F) shows that, despite these changes, there is no change in interference between the two intervals. To anticipate later results, this implies that the nature of interference in the HOM region predominates (below).*

Figures 7 and 8*.*
Figures 7 and 8
*have three components (1) They provide information about* fancm *effects and interplay of heterology with* fancm *recombination. These were discussed above. [The observed effects pertain specifically to heterozygous regions regardless of what is or is not adjacent to them. They do not say anything about what is going on in wild type non-interfering COs.] (2) They provide more data for wild type (*FANCM*+) strains in several situations. (3) They show that interference is required for the so-called* “cis *effects*”*. The exposition in the paper was hard to follow. Here is my own restatement of the situation.*

*Issue (2).*
Figure 7*. For the 420 interval, there are four CO frequencies: HOM/HOM 18cM; HOM/HET 15cM; HET/HET 21cM; HET/HOM 28cM. It is unclear what the multiple data points represent. If these are multiple different lines (and thus different* “trans” *effects, then the data, are more meaningful). Just looking at these numbers, some conclusions emerge:*

Adjacent HET reduces COs in HOM (in HOM/HET 15cm relative to HOM/HOM 18cm). This is the effect emphasized above.

Oppositely, however, adjacent HOM increases COs in HET (in HET/HOM 28cM vs HET HET 21cM). So the effects are reciprocal, which is not emphasized enough.

As discussed the use of independently derived recombinant lines in each case should mitigate against any potential *trans* effects. Our QTL analysis additionally argues against the existence of such modifiers in Col x Ct crosses as described above (Figure 4). We have rewritten the text to emphasize the reciprocal nature of these effects.

*For the CEN interval, HOM/HOM 12cM; HOM/HET 12cM; HET/HET 10cM; HET/HOM 9cM. Clearly the centromere behaves differently, for whichever reasons, as suggested by*
Figure 5
*data.*

The *CEN3* interval is expected to behave differently by nature of it being highly heterochromatic, as discussed earlier.

Figure 8
*addresses the same to issues above, but for a three-factor cross specifically in the sub-telomeric interval and with the addition of interference analysis.*

Left interval: HOM/HOM 16cM; HOM/HET 11cM; HET/HET 16cM; HET/HOM 18cM.

Right interval: HOM/HOM 5cM; HOM/HET 4cM; HET/HET 5cM; HET/HOM 6cM.

In this case:

*The left interval behave as the interval in*
Figure 7*, which is the sum of left and right in*
Figure 8*.*

HET reduces COs in HOM: HOM/HOM 16cM; HOM/HET 11cM; HOM increases COs in HET: HET/HOM 18cM; HET/HET 16cM.

The right interval behaves oppositely, but with small effects;

HET increases COs in HOM: HOM/HOM 5cM; HET/HOM 6cM;

HOM reduces COs in HET: HOM/HET 4cM; HET/HET 5cM.

*The right interval is small. Apparently the* “*left side dominates*”*. I have no idea what this means.*

The dominance of the ‘left’ interval (*I3b*) is likely to reflect the telomeric gradient of increasing recombination in *Arabidopsis* male meiosis ([45] PLoS Genetics). As *I3b* is closer to the telomere, and we are measuring male meiosis, it shows a higher recombination rate than *I3c* (Table 2).

Figure 8*. Interference is also analyzed in the wild type background. NB: there is nowhere in this paper (except the Methods) a description of what the interference metric is, even on the figure. Axis must be labeled (1-CoC) and this must be explained in the text.*

Results: HOM/HOM 0.6; HOM/HET 0.6cM; HET/HET 0.8; HET/HOM 0.8.

*(a) HET/HET has* “*more interference*” *than HOM/HOM, the effect discussed above Importantly, this is despite the fact that the frequencies of COs in both intervals are the same in these two situations! This would not be expected if the only difference were in interference. Thus, in the pure heterozygous situation, there must be two effects, one that increases COs plus interference that decreases COs, e.g. more DSBs and counterbalancing interference increase. (b) The two reciprocal* “*mixed*” *cases are not the same. HOM/HET looks like HOM/HOM while HET/HOM looks like HET/HET. Formally, the nature of the left interval dominates, as also seen with respect to effects on CO levels (above).*

Overall, it seems that the ability of a HET region to depress COs in a HOM region does not depend on having a HET-type CO interference process.

Figure 8*: In the larger/left interval, where canonical reciprocal effects are observed: HET reduces COs in HOM: HOM/HOM 16cM; HOM/HET 11cM; and for interference, HOM/HET looks like HOM/HOM. HOM increases COs in HET: HET/HOM 18cM; HET/HET 16cM and for interference, HET/HOM looks like HET/HET, and the same is seen in*
Figure 6*. This figure compared a HOM/HOM situation with a HET/HOM and found them to have the same (HOM) interference. And the* “*canonical*” *reciprocal effects are observed.*

We have added a more detailed explanation of our interference calculations to the section where we first describe *I3bc*. We have also changed the axis labels to ‘Interference (1-CoC)’. Additionally we have added the formulae used to calculate interference to Figure 5. We agree that our observations are consistent with the model of increased DSBs being recruited to heterozygous regions and have added these ideas to the Discussion.

*7) The last item is the interplay between* fancm *and CO interference, keeping in mind that the information does not tell us anything about what is going on in* FANCM+ *(and thus the general cases of heterozygosity seen in nature).*

We have rewritten the manuscript to make it clear that the non-interfering crossovers observed in *fancm* may not be biochemically the same as those observed in wild type.

*One expectation is that if more COs are coming from the* fancm*-revealed process, there will be less interference. This expectation is met in*
Figure 8*. More or less, there is plenty of interference in all four strains in* FANCM+. *And more or less, in both HOM/HET and HOM/HOM, where the entire test region is homozygous and the* fancm *mutation is having its full effect, there is a big reduction in interference. Whereas, in both HET/HOM and HET/HET, where the entire test region is heterozygous and there are fewer extra COs in the* fancm *mutant, interference is less reduced. And the* zip4 *mutation, in the* fancm *background, has the expected effects in all cases, reducing interference because it decreases the fraction of* “*interfering*” *COs. These results provide more support for the nature of the* fancm *interplay with heterozygosity (discussed in 2 above).*

*The authors also ask about variations in the* “*interplay*” *of* fancm *with HET/HOM differences. The underlying idea (which I finally realized after many hours) is that in* fancm zip4 *there are still COs but they do not show interference, so one can ask if the variations in 420 according to what is adjacent are still seen. If no, then interference is required.*

*The data for 420 in*
Figure 7
*are:*

*HOM/HOM 35cM HET/HET 12cm HET/HOM 10cM HOM/HET 35cm. Most importantly, since HOM/HET = HOM/HOM, there is no transmission of information from HET to HOM if interference is absent, which implicates interference in the process (in accord with the suggestions above). The reciprocal case: HET/HET 12cm HET/HOM 10cM seems to run against all rules, with HET increasing COs in the adjacent HET, but again, maybe this is a small effect and/or can be ignored. The fact that HET/HET << HOM/HOM is explained by the fact that there are fewer COs from the* fancm *pathway in heterozygous regions (above).*

*Similarly in*
Figure 8*: for left and right regions*

HOM/HOM 25, 12cM; HET/HET 8, 3.5; HET/HOM 7.5, 2.5cM HOM/HET 22, 12cm. These data are clear and very useful in showing that interference is relevant.

We are glad the reviewer agrees that these data support a role for crossover interference in the observed phenomena.

8) A final point on statistics: p-values are given everywhere as Chi-square p-values, but it seems the authors are comparing means, not comparing a result against an expectation. Is this a typo? Is it a different kind of test, or did they really use a Chi-square? (If the latter, they need to justify and clarify what the expectation was, otherwise give p-values for something that compares distributions, e.g. a T-test).

As our recombinant data is count-based we considered a t-test to be inappropriate. We have added further description of our statistical testing to the Materials and methods section as follows. To test whether recombinant and non-recombinant counts were significantly different between groups of replicates we used a generalized linear model (GLM) and assumed that the count data is binomially distributed:Yi∼B(ni,pi)

where *Y*_*i*_ represents the recombinant counts, ni are the total counts, and we wish to model the proportions *Y*_*i*_/*n*_*i*_. Then:

E(Yi/ni)=pi, and var(Yi/ni)=pi(1−pi)ni.

Thus, our variance function is: V(μi)=μi(1−μi), and our link function must map from (0,1) -> (-∞, ∞). We used a logistic link function which is: g(μi)=logit(μi)=logμi1−μi=βX+εi,

where ει∼N(0,σ2). Both replicates and genotypes are treated as independent variables ((*X*)) in our model. We considered *P* values less than 0.05 as significant.

Reviewer #3:

There is no question that the manuscript of Ziolkowski et al. is of interest as it is challenging current views of crossover control, not only in plants.

I also read the thoughtful comment and valuable suggestions of both reviewers with great interest. I think that many of the points raised by the reviewers the authors were able to address in their response, especially by the inclusion of new experimental data.

We are pleased the reviewer found our findings of interest and that our new experimental data has addressed previous concerns.

I have to say that the main conclusion that interfering crossovers are enhanced in heterozygous regions is for me as for the other reviewers counterintuitive. Also, the model the authors bring forward at the end, that recombination intermediates are stabilized by mismatch recognition is quite special. Therefore it is of course important to take other explanations into account.

We agree that currently there are several possibilities for the mechanistic basis of our observations. We have attempted to include those that we consider most likely in the Discussion, including several of those raised by the reviewers.

*There is one thing that worries me a bit and that is quality of the sequence data available for the different* Arabidopsis *cultivars. Do heterozygous region really attract crossovers from homozygous regions in hybrids? As the authors state themselves, these sequences were aligned by short read technology and so the occurrence of inversions and of duplication in the range from hundreds to hundred-thousands of bps might be dramatically underestimated. Thus, changes above the nucleotide level were not taken into account by the authors. Nevertheless these changes might drastically influence recombination patterns.*

The reviewer raises a valid point concerning our knowledge of structural polymorphisms between the accessions analysed. We provide clear evidence of their effect in the suppression of *420* crossovers in Col/Sha F_1_ hybrids, as Sha is known to contain a large inversion overlapping this region (Figure 3). However, the extent of similar inversions or other structural polymorphisms likely remains underestimated. To acknowledge this fact we have added the following sentence to the Results section ‘Heterozygosity extensively modifies crossover frequency in *Arabidopsis*’: “Hence the contribution of unknown structural polymorphisms to variation in recombination rates could be significant.”

*Nevertheless, all in all, the authors supply us in the revised version with enough hard data that in my opinion strengthen their hypothesis to the point that it should be considered (and challenged) by the community. Also, it might well be that such a mechanism is exceptional and restricted to self-fertilizing species like* Arabidopsis*. But this would be still interesting.*

We have added a sentence to the Discussion acknowledging the importance of considering mating system (selfing versus outcrossing) in applying these findings to other systems: ‘However, when assessing the significance of such effects it is also important to consider how outcrossing versus selfing will influence patterns of homozygosity and heterozygosity within different species.’